# Detrimental proarrhythmogenic interaction of Ca$^{2+}$/calmodulin-dependent protein kinase II and Na$_V$1.8 in heart failure

Philipp Bengel [1,2,7], Nataliya Dybkova[1,2,7], Petros Tirilomis[1,2,7], Shakil Ahmad[1,2,3], Nico Hartmann[1,2], Belal A. Mohamed[1,2], Miriam Celine Krekeler[1,2], Wiebke Maurer[1,2], Steffen Pabel[3], Maximilian Trum[3], Julian Mustroph[3], Jan Gummert[4], Hendrik Milting[4], Stefan Wagner[3], Senka Ljubojevic-Holzer [5], Karl Toischer[1,2], Lars S. Maier[3], Gerd Hasenfuss[1,2], Katrin Streckfuss-Bömeke [1,2,6,7] & Samuel Sossalla [1,2,3,7 ✉]

An interplay between Ca$^{2+}$/calmodulin-dependent protein kinase IIδc (CaMKIIδc) and late Na$^+$ current (I$_{NaL}$) is known to induce arrhythmias in the failing heart. Here, we elucidate the role of the sodium channel isoform Na$_V$1.8 for CaMKIIδc-dependent proarrhythmia. In a CRISPR-Cas9-generated human iPSC-cardiomyocyte homozygous knock-out of Na$_V$1.8, we demonstrate that Na$_V$1.8 contributes to I$_{NaL}$ formation. In addition, we reveal a direct interaction between Na$_V$1.8 and CaMKIIδc in cardiomyocytes isolated from patients with heart failure (HF). Using specific blockers of Na$_V$1.8 and CaMKIIδc, we show that Na$_V$1.8-driven I$_{NaL}$ is CaMKIIδc-dependent and that Na$_V$1.8-inhibtion reduces diastolic SR-Ca$^{2+}$ leak in human failing cardiomyocytes. Moreover, increased mortality of CaMKIIδc-overexpressing HF mice is reduced when a Na$_V$1.8 knock-out is introduced. Cellular and in vivo experiments reveal reduced ventricular arrhythmias without changes in HF progression. Our work therefore identifies a proarrhythmic CaMKIIδc downstream target which may constitute a prognostic and antiarrhythmic strategy.

[1] Clinic for Cardiology & Pneumology, Georg-August University Göttingen, Göttingen, Germany. [2] DZHK (German Centre for Cardiovascular Research), partner site Göttingen, Göttingen, Germany. [3] Clinic and Polyclinic for Internal Medicine II, University Medical Centre Regensburg, Regensburg, Germany. [4] Heart and Diabetes Centre North Rhine-Westphalia, Bad Oeynhausen, Germany. [5] Department of Cardiology, Medical University of Graz, Graz, Austria. [6] Institute of Pharmacology and Toxicology, University of Würzburg, Würzburg, Germany. [7] These authors contributed equally: Philipp Bengel, Nataliya Dybkova, Petros Tirilomis, Katrin Streckfuss-Bömeke, Samuel Sossalla. ✉email: Samuel.Sossalla@klinik.uni-regensburg.de

Voltage-gated sodium channels (Na$_V$) play a critical role in physiological cardiac conduction. Na$_V$ channels become inactive within a few milliseconds after activation under physiological conditions. However, in cardiac pathologies such as ischemia, hypoxia, oxidative stress, and heart failure (HF), some Na$_V$ remain persistently open or reopen, generating a small but persistent Na$^+$ current, referred to as the late Na$^+$ current (I$_{NaL}$)[1–5]. This current slows the repolarisation rate and thereby prolongs the action potential duration (APD). Augmented I$_{NaL}$ may additionally cause Na$^+$-dependent Ca$^{2+}$ overload in cardiomyocytes, thereby playing an essential role for arrhythmogenesis and diastolic dysfunction[1,2,6,7]. Furthermore, Na$^+$/Ca$^{2+}$ overload caused by augmented I$_{NaL}$ can give rise to early afterdepolarizations (EADs), delayed afterdepolarizations (DADs), and hence sustained triggered arrhythmias[8–11]. In the failing heart, increased I$_{NaL}$ induces an influx of Na$^+$ into the cardiomyocyte, which in turn stimulates Ca$^{2+}$ influx via the reverse mode of Na$^+$/Ca$^{2+}$ exchanger (NCX)[12,13]. Cytosolic Ca$^{2+}$ may now bind to calmodulin (CaM), forming a Ca$^{2+}$/CaM complex, which activates Ca$^{2+}$/calmodulin-dependent protein kinase IIδ (CaMKIIδ), a multifunctional serine/threonine protein kinase[8,10,14,15]. CaMKII is expressed in four isoforms α, β, γ, and δ. CaMKIIδ is the predominant isoform in heart[16] while the δc isoform is mainly located in the cytosol. Once CaMKIIδc is activated, it may cause hyperphosphorylation of the ryanodine receptor 2 (RyR2) residing within the sarcoplasmic reticulum (SR)-sarcolemma junction, leading to spontaneous proarrhythmogenic SR-Ca$^{2+}$ release events in HF[13,17–20]. Further, this augmented CaMKIIδc activity can also induce I$_{NaL}$ augmentation by phosphorylating Na$_V$ channels[10,14,21,22] leading to a vicious cycle between I$_{NaL}$ and CaMKIIδc.

Besides Na$_V$1.5 other Na$_V$ isoforms have been reported to be present in the heart. Theoretically, they could also generate I$_{NaL}$, induce APD prolongation, and spontaneous SR-Ca$^{2+}$ release. In the previous few years, different reports have been published on the existence of noncardiac Na$_V$ in the heart. Na$_V$1.8, a noncardiac tetrodotoxin-resistant Na$_V$ channel, is encoded by the SCN10A gene and was originally reported to be expressed in the dorsal root ganglion[23]. Certain genome-wide association studies reported an association of SCN10A with changes in ECG parameters but most importantly with cardiac arrhythmias such as atrial fibrillation and sudden cardiac death[24–27]. Later, the presence of Na$_V$1.8 was detected in atria and further evidence came from studies conducted in mouse and rabbit cardiomyocytes investigating its involvement in cardiac electrophysiology[28–30]. Moreover, we recently reported that Na$_V$1.8 mRNA and protein expression are upregulated in tissue from human hypertrophied and failing ventricles and that Na$_V$1.8 contributes to I$_{NaL}$ generation in the human heart under these pathological conditions[31,32]. However, Na$_V$1.8 regulation, a potential interplay with pathologically increased CaMKIIδc activity in HF, and the role of Na$_V$1.8 on HF progression and arrhythmias in vivo and in vitro remain elusive.

In this work, we describe a detrimental interaction of Na$_V$1.8 with CaMKIIδc in human and murine failing cardiomyocytes. Moreover, we investigate the contribution of Na$_V$1.8 to cellular electrophysiology in relation to enhanced CaMKIIδc activity and consequently show a reduction of arrhythmias which is paralleled by an improved survival due to Na$_V$1.8 deletion in CaMKIIδc transgenic HF mice.

## Results

**Na$_V$1.8 and CaMKIIδc interaction in human HF.** Since interaction between CaMKIIδc and Na$_V$1.5 was shown previously, we hypothesized that CaMKIIδc interacts also with Na$_V$1.8 and therefore performed co-immunoprecipitation using human ventricular tissue homogenates. Indeed, we found that CaMKIIδc associates with Na$_V$1.8 in human non-failing as well as HF myocardium (Fig. 1a). Using immunofluorescence stainings, CaMKIIδc and Na$_V$1.8 were found to be co-localized in isolated human cardiomyocytes (Fig. 1b). Of note, SCN10A mRNA expression in tissue from non-failing and HF hearts, as well as isolated cardiomyocytes from human HF hearts was found to be much lower than SCN5A. Further, RT-qPCR experiments revealed that a relevant part of SCN10A mRNA in the heart originates from cardiomyocytes (Supplementary information, Supplementary Figs. 1, 2).

**Na$_V$1.8 inhibition reduces I$_{NaL}$ in human failing, mouse CaMKIIδc transgenic cardiomyocytes, and in SCN10A knockout iPSC-cardiomyocytes.** In functional experiments, we could show that I$_{NaL}$, induced by increased CaMKIIδc activity, was significantly reduced following pharmacological inhibition and genetical knockout of Na$_V$1.8 in human and murine cardiomyocytes. In isolated murine CaMKIIδc$^{+/T}$ cardiomyocytes, I$_{NaL}$ was augmented to ~150% compared to wildtype (WT), whereas the specific Na$_V$1.8 blockers A-806734 or PF-01247324 reduced I$_{NaL}$ by ~40% in the same background (CaMKIIδc$^{+/T}$) (Fig. 2a, b). These results illustrate that CaMKIIδc-induced I$_{NaL}$ can be clearly ameliorated by inhibiting Na$_V$1.8. A current–voltage relationship of Na$_V$1.8-dependent I$_{NaL}$ in isolated ventricular myocytes from CaMKIIδc$^{+/T}$ mice is presented in the Supplementary information (Supplementary Fig. 3).

In the human failing heart, CaMKIIδc and I$_{NaL}$ are upregulated in parallel[2,33]. Therefore, we inhibited Na$_V$1.8 by using PF-01247324 and compared its ability to reduce I$_{NaL}$ to CaMKIIδc inhibition using autocamtide-2-related inhibitory peptide (AIP) in human failing ventricular cardiomyocytes. In addition, we blocked Na$_V$1.8 and CaMKIIδc in parallel by exposing human failing cells simultaneously to PF-01247324 and AIP. I$_{NaL}$ measurements demonstrated that Na$_V$1.8 inhibition alone (PF-01247324) led to a ~40% decrease and CaMKIIδc inhibition

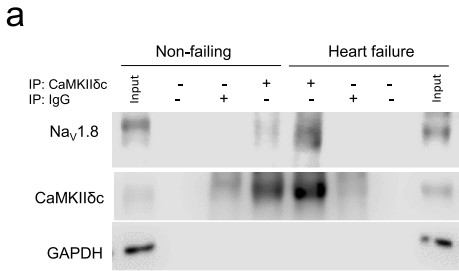
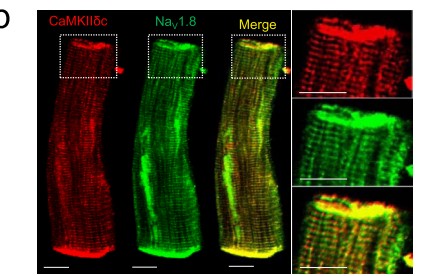

**Fig. 1 CaMKIIδc interacts with Na$_V$1.8 in human myocardium and isolated cardiomyocytes. a** Co-immunoprecipitation of CaMKIIδc and Na$_V$1.8 from left ventricular homogenates of human non-failing and failing hearts (NF: $n = 7$; HF: $n = 7$). **b** Co-localization of CaMKIIδc and Na$_V$1.8 in human failing cardiomyocytes with immunofluorescence staining. Scale bar: 10 μm (staining was performed in cardiomyocytes isolated from five heart failure patients).

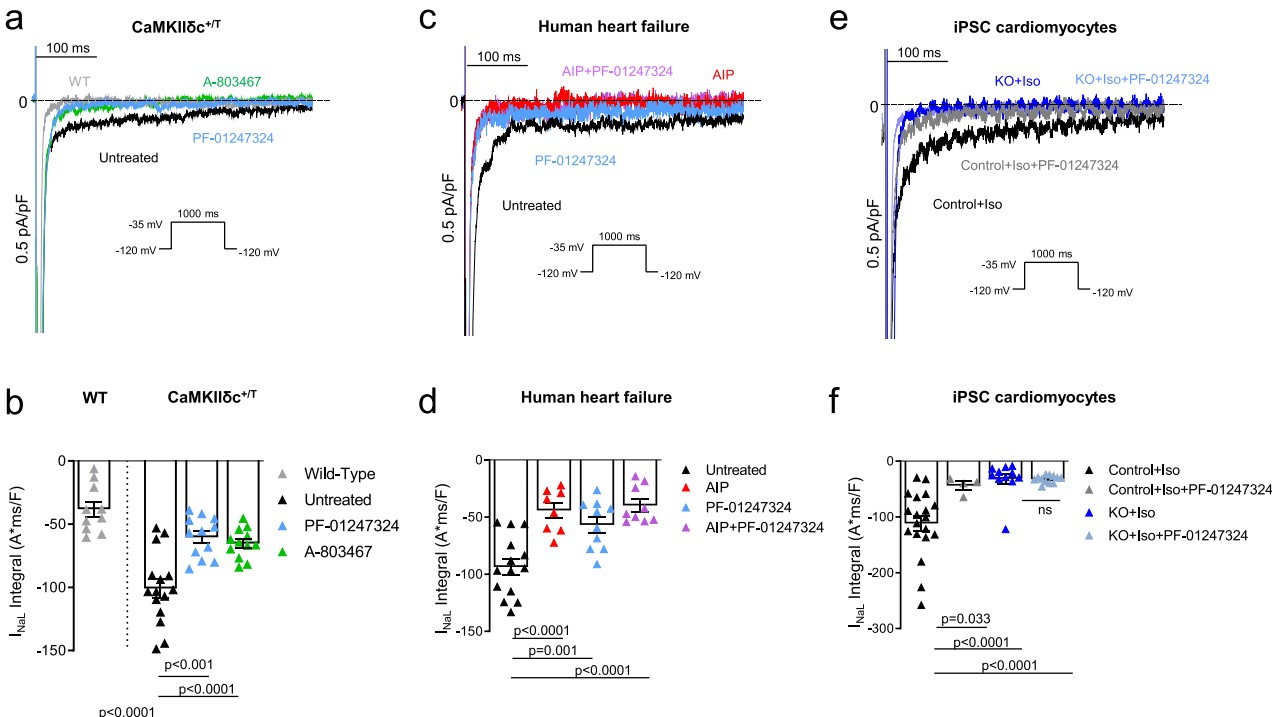

**Fig. 2 Reduced I_NaL upon Na_V1.8 inhibition in human failing and mouse CaMKIIδc transgenic cardiomyocytes, and in *SCN10A* knockout iPSC-cardiomyocytes. a** Original traces of I_NaL in WT and CaMKIIδc$^{+/T}$ mouse ventricular cardiomyocytes elicited using the protocol shown in the inset. **b** Mean data ± SEM along with individual values shown in the graph plotting WT ($n = 10$ cells/5 mice) and CaMKIIδc$^{+/T}$ (untreated: $n = 15$ cells/7 mice; A-806467: $n = 12$ cells/7 mice; PF-01247324: $n = 12$ cells/7 mice). Probability vs. untreated (One-way ANOVA with post hoc Bonferroni's correction). **c** Original traces of I_NaL from human failing ventricular cardiomyocytes elicited using the protocol shown in the inset. **d** Mean data ± SEM along with individual values shown in the graph plotting (untreated: $n = 14$ cells/8 patients; AIP: $n = 8$ cells/5 patients; PF-01247324: $n = 10$ cells/6 patients; AIP + PF-01247324 = 9 cells/5 patients). Probability vs. untreated (One-way ANOVA with post hoc Bonferroni's correction). **e** Original traces of I_NaL from human ventricular *SCN10A* knockout iPSC-cardiomyocytes elicited using the protocol shown in the inset. **f** Mean data ± SEM along with individual values shown in the graph plotting (control + Iso: $n = 19$ cells/3 cardiac differentiations; control + Iso + PF: $n = 4$ cells/3 differentiations; *SCN10A* knockout (KO) + Iso: $n = 11$ cells/3 differentiations; KO + Iso + PF-01247324 = 12 cells/3 differentiations). Probability vs. control + Iso (One-way ANOVA with post hoc Bonferroni's correction).

by AIP to a ~53% reduction of I_NaL in human failing cardiomyocytes (Fig. 2c, d). However, preincubation with AIP and PF-01247324 together decreased I_NaL comparable to AIP alone suggesting that CaMKIIδc inhibition might already suppress Na_V1.8-driven I_NaL. Further, peak I_Na measurements revealed, that I_NaL reduction due to Na_V1.8 inhibition is not caused by a reduction of overall Na$^+$ current (Supplementary information, Supplementary Fig. 4).

As the existence of Na_V1.8 and its role in cardiomyocytes is still a matter of debate, we generated homozygous Na_V1.8 knockout (KO) lines by using CRISPR-Cas9 in human induced pluripotent stem cells (iPSC) and differentiated these cells into 2-month-old cardiomyocytes. Sanger sequencing demonstrated frameshifts and premature stop codons on both alleles (Supplementary information, Supplementary Fig. 5). As the amplitude of I_NaL is relatively small under healthy conditions we used isoproterenol (Iso, 50 nmol/l) to enhance I_NaL. Control iPSC-cardiomyocytes treated simultaneously with Iso and PF-01247324 exhibited ~60% less I_NaL compared to Iso alone, (Fig. 2e, f). Most importantly, in Na_V1.8 KO iPSC-cardiomyocytes I_NaL was reduced by ~70% compared to the control iPSC-cardiomyocytes. PF-01247324 did not exert any further impact on I_NaL in Na_V1.8 KO iPSC-cardiomyocytes compared to untreated Na_V1.8 KO cells, underlining the specificity of the drug.

**Na_V1.8 modulates Ca$^{2+}$ homeostasis under enhanced CaMKIIδc activity.** In HF, enhanced I_NaL can potently induce

proarrhythmic SR-Ca$^{2+}$ release events[8,11]. Accordingly, we investigated whether inhibition of the Na_V1.8-mediated I_NaL could attenuate the increase of the proarrhythmogenic SR-Ca$^{2+}$ spark frequency (CaSpF) caused by overexpression of CaMKIIδc (Fig. 3a, b). We incubated CaMKIIδc$^{+/T}$ mouse cardiomyocytes with Na_V1.8 inhibitors and measured the CaSpF. A ~50% reduction of CaSpF was observed in both Na_V1.8 inhibitor groups compared to untreated cells (Fig. 3a, b). These results display that SR-Ca$^{2+}$ leak due to increased CaMKIIδc expression and activity can be reduced by inhibiting Na_V1.8.

It is well known that inhibition of CaMKIIδc can attenuate SR-Ca$^{2+}$ leak[18,34]. However, therapeutic general inhibition of CaMKIIδc in humans may not be suitable because of its pivotal involvement in different vital pathways such as learning processes[35]. We explored whether the Na_V1.8 inhibitor PF-01247324 exerts similar effects comparable to the inhibition of CaMKIIδc. Incubation of human failing cardiomyocytes with either the CaMKIIδc inhibitor AIP or the Na_V1.8 inhibitor PF-01247324 resulted in a similar reduction of CaSpF compared to untreated cells (Fig. 3c, d). Furthermore, blocking CaMKIIδc and Na_V1.8 in parallel resulted in a significant reduction of CaSpF, comparable to AIP or PF-01247324 alone in human failing cardiomyocytes (Fig. 3c, d).

We further investigated whether Na_V1.8 inhibition modulates the Ca$^{2+}$ transient amplitude and SR-Ca$^{2+}$ load in cardiomyocytes isolated from CaMKIIδc$^{+/T}$ mice. Na_V1.8 inhibition using PF-01247324 did not pose any effect on either the Ca$^{2+}$ transient

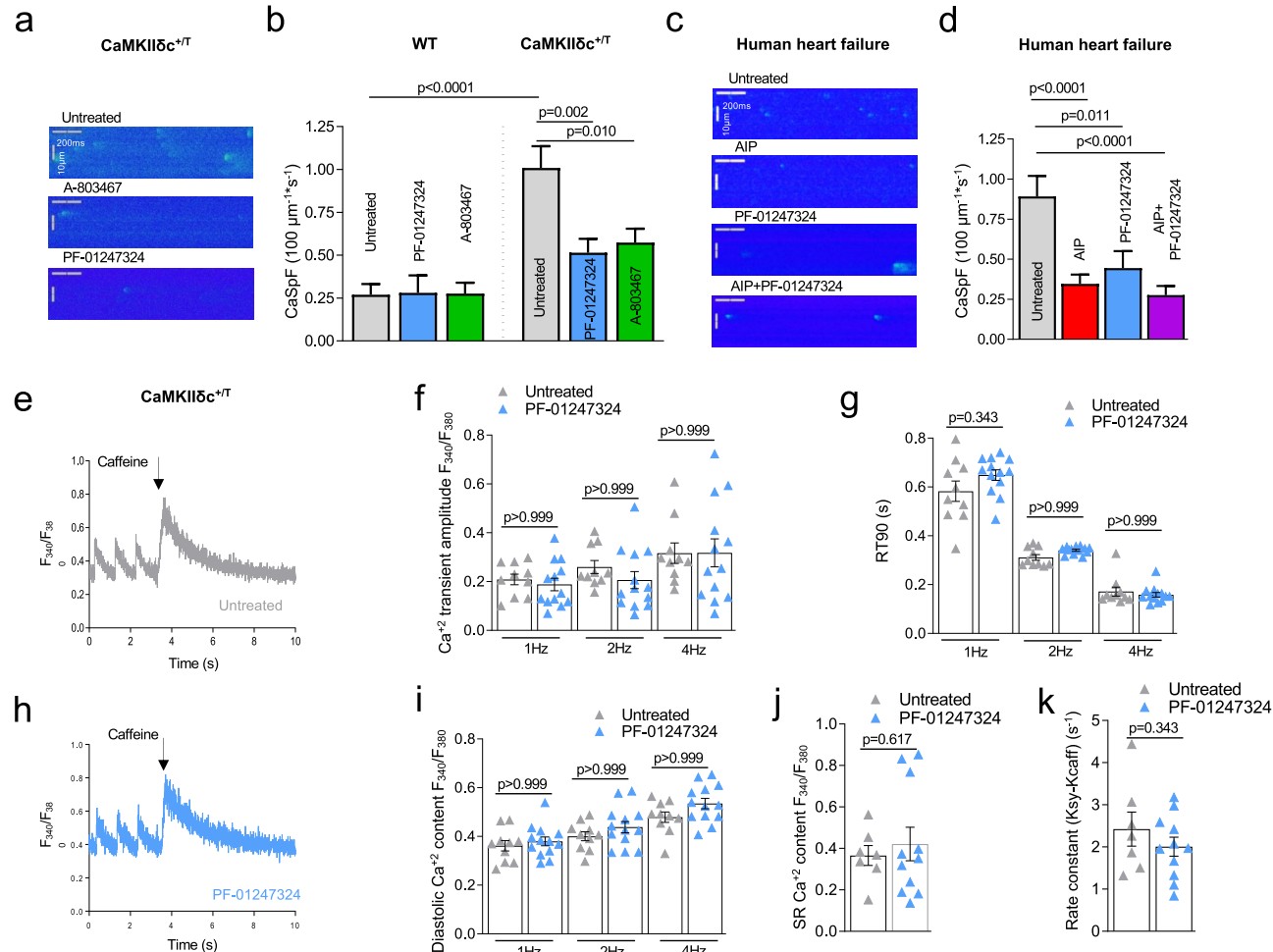

**Fig. 3 Effects Na$_V$1.8 inhibition on intracellular Ca$^{2+}$ handling. a** Representative line scan images of CaMKIIδc$^{+/T}$ ventricular cardiomyocytes. **b** CaSpF data shown as mean ± SEM for wildtype (WT) (untreated: $n = 58$ cells/4 mice; PF-01247324: $n = 41$ cells/4 mice; A-806467: $n = 41$ cells/4 mice) and CaMKIIδc$^{+/T}$ (untreated: $n = 122$ cells/8 mice; PF-01247324: $n = 105$ cells/7 mice; A-806467: $n = 101$ cells/8 mice). Data were analyzed by one-way ANOVA with post hoc Bonferroni's correction. **c** Representative line scan images of human failing ventricular cardiomyocytes. **d** Data shown as mean ± SEM (untreated: $n = 123$ cells/14 patients; Autocamtide-2-related inhibitory peptide (AIP): $n = 105$ cells/15 patients; PF-01247324: $n = 59$ cells/10 patients; AIP + PF-01247324 = 89 cells/13 patients). Data were analyzed by one-way ANOVA with post hoc Bonferroni's correction, Probability vs. untreated. **e** Representative Ca$^{2+}$ transients stimulated at 1 Hz and caffeine-induced Ca$^{2+}$ transients in ventricular cardiomyocytes from CaMKIIδc$^{+/T}$ under untreated conditions. **f** Mean data ± SEM show no effect of PF-01347324 treatment on Ca$^{2+}$ transient amplitude at 1.0, 2.0, and 4.0 Hz stimulation ($n = 13$ cells/5 mice) compared to untreated ($n = 10$ cells/5 mice). **g** Ca$^{2+}$-transient decay (90% of Ca$^{2+}$-removal) RT90 was unchanged in PF-01347324-treated cells ($n = 11$ cells/5 mice) compared to untreated ($n = 8$ cells/5 mice). Data were presented as mean values ± SEM. **h** Representative Ca$^{2+}$ transients stimulated at 1 Hz and caffeine-induced Ca$^{2+}$ transients in ventricular cardiomyocytes from CaMKIIδc$^{+/T}$ treated with PF-01247324. **i** Diastolic Ca$^{2+}$ after addition of PF-01347324 ($n = 13$ cells/5 mice) compared to untreated cells ($n = 10$ cells/5 mice) at different stimulation frequencies was unchanged (one-way ANOVA with post hoc Bonferroni's correction, Fig. 3f, g, i). Data were presented as mean values ± SEM. **j** Mean and individual values ± SEM of caffeine-induced Ca$^{2+}$ transients (untreated: $n = 7$ cells/4 mice, PF-01247324: $n = 11$ cells/5 mice) did not differ between the groups (Student's $t$-test). **k** Ca$^{2+}$-reuptake into the SR was not affected by inhibition of Na$_V$1.8 (untreated: $n = 7$ cells/4 mice, PF-01247324: $n = 11$ cells/5 mice), analyzed by Student's $t$-test. Data were presented as mean values ± SEM.

amplitude or Ca$^{2+}$ transient decay measured at different stimulation frequencies (Fig. 3e–h). Furthermore, diastolic Ca$^{2+}$, SR-Ca$^{2+}$ content, and SERCA2a activity were not affected by PF-01247324 (Fig. 3i–k). Similar effects of PF-01247324 on Ca$^{2+}$ transient kinetics and SR-Ca$^{2+}$ content were observed in cardiomyocytes isolated from WT mice (Supplementary information, Supplementary Fig. 6).

**Scn10a knockout improves survival in CaMKIIδc$^{+/T}$ mice in the absence of structural ventricular changes.** To study whether inhibition of Na$_V$1.8 influences the development of HF, arrhythmogenesis, or survival in CaMKIIδc$^{+/T}$ mice, we crossbred these mice with Na$_V$1.8 knockout mice. Interestingly,

SCN10A$^{-/-}$/CaMKIIδc$^{+/T}$ mice showed a significantly improved survival compared to CaMKIIδc$^{+/T}$ (median survival 98 vs. 72 days, 64 vs. 43 animals, Hazard Ratio 0.6) as assessed in blinded investigations. Specifically, CaMKIIδc$^{+/T}$ mice showed only a 37% survival at 12 weeks, whereas SCN10A$^{-/-}$/CaMKIIδc$^{+/T}$ died at a slower rate, with 67% survival at this age (Fig. 4a). These data illustrate that Na$_V$1.8 knockout is capable to counteract the lethal phenotype of CaMKIIδc overexpression to a relevant extent.

Detailed investigations of hearts from SCN10A$^{-/-}$/CaMKIIδc$^{+/T}$ double-mutant and CaMKIIδc$^{+/T}$ mice by the age of 12 weeks exhibited comparably enlarged heart chambers (Fig. 4b). Heart weight to tibia length ratio was similarly increased in double-mutant and CaMKIIδc$^{+/T}$ mice (Fig. 4c). To investigate whether Na$_V$1.8

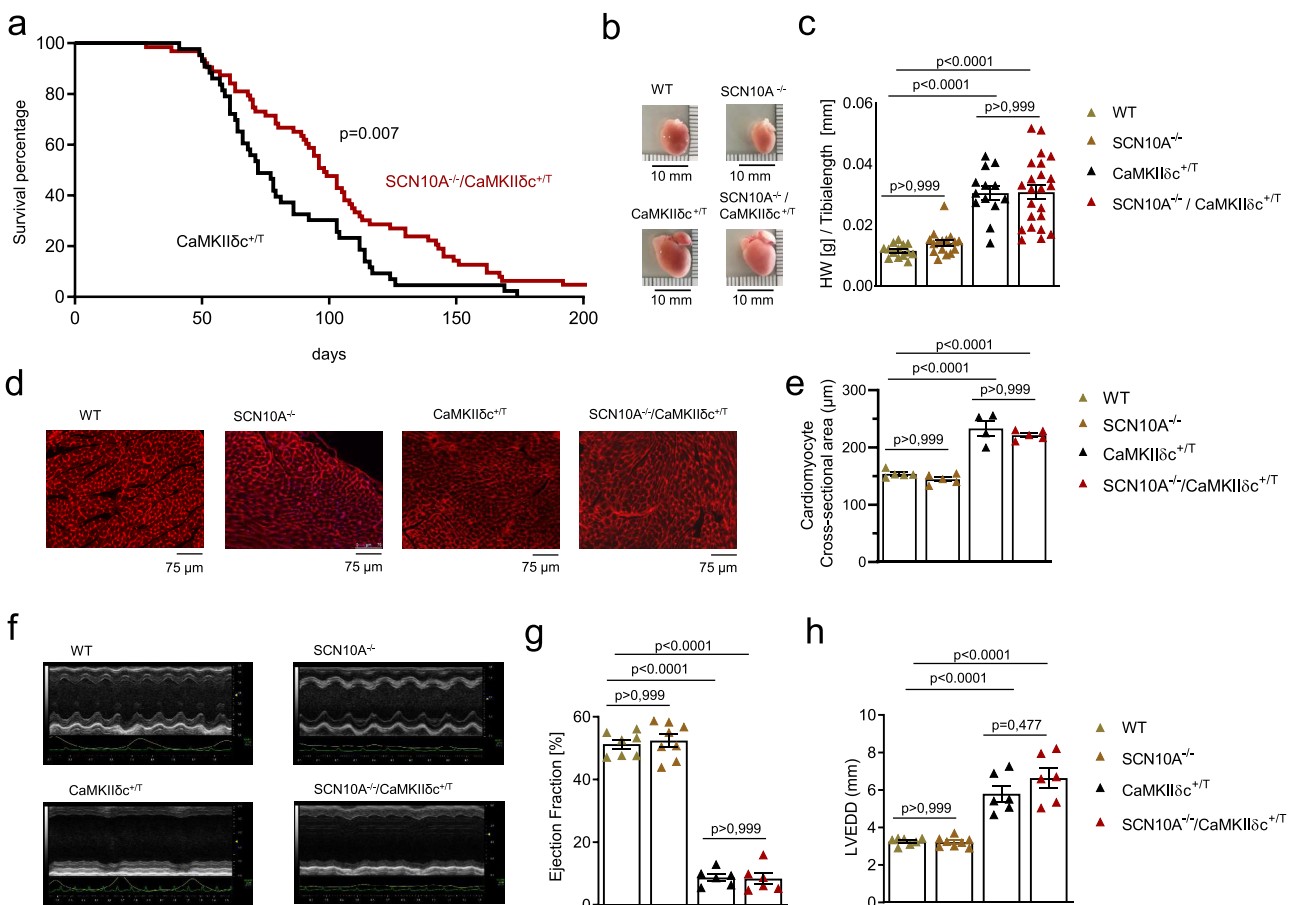

**Fig. 4 Knockout of the *Scn10a* (Na$_V$1.8) gene in CaMKIIδc$^{+/T}$ mice improves survival. a** Survival curve of CaMKIIδc$^{+/T}$ and SCN10A$^{-/-}$/CaMKIIδc$^{+/T}$ (43 vs. 64 animals, median survival 72 vs. 98 days, blinded analysis). Log-rank (Mantel–Cox test and Gehan–Breslow–Wilcoxon test (two-tailed analysis) were performed to calculate the survival percentage of mice. Probability vs CaMKIIδc$^{+/T}$. **b** Hearts from WT, SCN10A$^{-/-}$, CaMKIIδc$^{+/T}$, and SCN10A$^{-/-}$/CaMKIIδc$^{+/T}$ mice. **c** Ratio of heart weight to tibia length as a parameter of cardiac hypertrophy. CaMKIIδc$^{+/T}$ and SCN10A$^{-/-}$/CaMKIIδc$^{+/T}$ showed a significant increase in this ratio compared to WT and SCN10A$^{-/-}$ mice. Data were analyzed by one-way ANOVA with post hoc Bonferroni's correction. ($N$ = hearts studied, WT = 14, SCN10A$^{-/-}$ = 16, CaMKIIδc$^{+/T}$ = 13, and SCN10A$^{-/-}$/CaMKIIδc$^{+/T}$ = 23). Data were presented as mean values ± SEM. **d** Original histological wheat germ agglutinin staining from WT, SCN10A$^{-/-}$, CaMKIIδc$^{+/T}$, and SCN10A$^{-/-}$/CaMKIIδc$^{+/T}$ mice. Scale bars = 75 μm. Stainings were produced from different sections and three different regions (basal, mid-ventricular, and apical) of each heart studied. **e** Cardiomyocyte cross-sectional-area (CSA) as a parameter for cellular hypertrophy. CaMKIIδc$^{+/T}$ and SCN10A$^{-/-}$/CaMKIIδc$^{+/T}$ showed a significant increase in CSA compared to WT and SCN10A$^{-/-}$ mice. CSA in CaMKIIδc$^{+/T}$ and SCN10A$^{-/-}$/CaMKIIδc$^{+/T}$ mice did not significantly differ. Data were analyzed by one-way ANOVA with post hoc Bonferroni's correction. $N$ = hearts studied (>300 cardiomyocytes were studied per heart, from different sections and different regions (basal, mid-ventricular, apical), WT = 5 hearts, SCN10A$^{-/-}$ = 5 hearts, CaMKIIδc$^{+/T}$ = 4 hearts, SCN10A$^{-/-}$/CaMKIIδc$^{+/T}$ = 5 hearts. Data were presented as mean values ± SEM. **f** Original echocardiography recordings from WT, SCN10A$^{-/-}$, CaMKIIδc$^{+/T}$, and SCN10A$^{-/-}$/CaMKIIδc$^{+/T}$ at M-mode in 12–13- week-old mice. **g** Echocardiography recordings revealed a decrease in left ventricular ejection fraction (EF) in CaMKIIδc$^{+/T}$ (six mice) and SCN10A$^{-/-}$/CaMKIIδc$^{+/T}$ (six mice) compared to WT (seven mice) or SCN10A$^{-/-}$ (eight mice) ($p$ < 0.0001(one-way ANOVA with post hoc Bonferroni's correction). Data were presented as mean values ± SEM. **h** Echocardiography recordings revealed a significant increase in left ventricular end-diastolic diameter (LVEDD) in CaMKIIδc$^{+/T}$ (six mice) and SCN10A$^{-/-}$/CaMKIIδc$^{+/T}$ (six mice) compared to WT (seven mice) or SCN10A$^{-/-}$ (eight mice) ($p$ < 0.0001). LVEDD was not significantly different in WT vs SCN10A$^{-/-}$ or CaMKIIδc$^{+/T}$ vs SCN10A$^{-/-}$/CaMKIIδc$^{+/T}$ mice (one-way ANOVA with post hoc Bonferroni's correction). Data were presented as mean values ± SEM.

knockout influences CaMKIIδc-mediated hypertrophy we prepared histological heart stainings. CaMKIIδc overexpression led to an increase of cardiomyocyte cross-sectional area (CSA) compared to WT and SCN10A$^{-/-}$ in mouse left ventricles. However, in double-mutant mice CSA did not differ from CaMKIIδc$^{+/T}$ alone (Fig. 4d, e). In addition, *Scn10a* knockout did not change the expression of Na$_V$1.5 (*Scn5a*) and CaMKIIδc on protein and mRNA levels in hearts from WT and CaMKIIδc$^{+/T}$ mice (Supplementary information, Supplementary Figs. 7, 8).

Serial echocardiography revealed a severe HF phenotype in CaMKIIδc$^{+/T}$ and SCN10A$^{-/-}$/CaMKIIδc$^{+/T}$ with a significant reduction of left ventricular ejection fraction (EF) (Fig. 4f, g).

Moreover, significantly enlarged left ventricular end-diastolic diameters (LVEDD) were measured in CaMKIIδc$^{+/T}$ and SCN10A$^{-/-}$/CaMKIIδc$^{+/T}$ compared to WT or SCN10A$^{-/-}$ (Fig. 4f, h). There were no changes between WT and SCN10A$^{-/-}$ or CaMKIIδc$^{+/T}$ and SCN10A$^{-/-}$/CaMKIIδc$^{+/T}$ mice with respect to LVEDD and EF.

**Reduction of proarrhythmic activity in SCN10A$^{-/-}$/CaMKIIδc$^{+/T}$ cardiomyocytes.** To test whether enhanced I$_{NaL}$ in CaMKIIδc$^{+/T}$ mice can be ameliorated by genetic knockout of Na$_V$1.8, we measured I$_{NaL}$ in isolated mouse cardiomyocytes (Fig. 5a, b). While CaMKIIδc$^{+/T}$ cardiomyocytes showed significantly enhanced I$_{NaL}$,

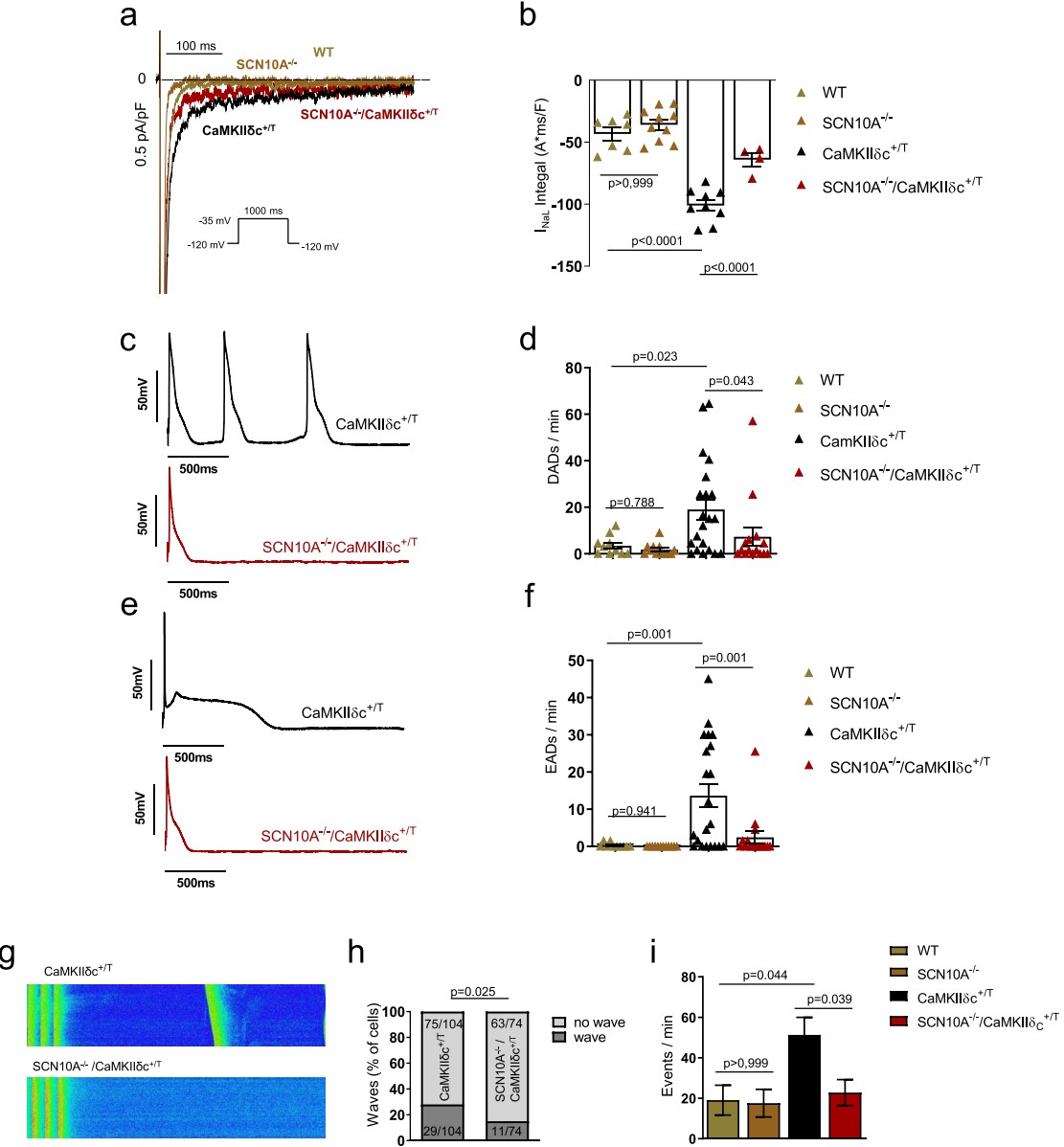

**Fig. 5 Knockout of *Scn10a* (Na$_V$1.8) in CaMKIIδc$^{+/T}$ mice (SCN10A$^{-/-}$/CaMKIIδc$^{+/T}$) significantly reduces I$_{NaL}$ and proarrhythmic triggers. a** Original traces of I$_{NaL}$ in WT, SCN10A$^{-/-}$, CaMKIIδc$^{+/T}$, and SCN10A$^{-/-}$/CaMKIIδc$^{+/T}$ mouse ventricular cardiomyocytes elicited using the protocol shown in the inset. **b** Mean data ± SEM along with individual values shown in the graph plotting (WT: $n = 7$ cells/4 mice, SCN10A$^{-/-}$ $n = 10$ cells/5 mice, CaMKIIδc$^{+/T}$: $n = 9$ cells/5 mice; SCN10A$^{-/-}$/CaMKIIδc$^{+/T}$: $n = 4$ cells/4 mice), there was a significantly reduced I$_{NaL}$ in SCN10A$^{-/-}$/CaMKIIδc$^{+/T}$ cardiomyocytes compared to CaMKIIδc$^{+/T}$. Data were analyzed by one-way ANOVA with post hoc Bonferroni's correction. **c** Original traces of action potentials showing triggered action potentials originating from delayed afterdepolarizations (DADs) in CaMKIIδc$^{+/T}$ and SCN10A$^{-/-}$/CaMKIIδc$^{+/T}$ cardiomyocytes. **d** Graph of mean data ± SEM along with individual values showing DADs per minute in WT ($n = 10$ cells/5 mice), SCN10A$^{-/-}$ ($n = 12$ cells/5 mice), CaMKIIδc$^{+/T}$ ($n = 21$ cells/5 mice) and SCN10A$^{-/-}$/CaMKIIδc$^{+/T}$ ($n = 15$ cells/5 mice) cardiomyocytes. There were significantly less events of afterdepolarizations in SCN10A$^{-/-}$/CaMKIIδc$^{+/T}$ compared to CaMKIIδc$^{+/T}$ cardiomyocytes. Data were analyzed by one-way ANOVA with the post hoc two-stage step-up method of Benjamini, Krieger, and Yekutieli. **e** Original traces of action potential showing early afterdepolarizations (EADs) in CaMKIIδc$^{+/T}$ and SCN10A$^{-/-}$/CaMKIIδc$^{+/T}$ cardiomyocytes. **f** Graph of mean data ± SEM along with individual values showing EADs per minute in WT ($n = 10$ cells/5 mice), SCN10A$^{-/-}$ ($n = 12$ cells/5 mice), CaMKIIδc$^{+/T}$ ($n = 21$ cells/5 mice) and SCN10A$^{-/-}$/CaMKIIδc$^{+/T}$ ($n = 16$ cells/5 mice) cardiomyocytes. There were significantly less events of afterdepolarizations in SCN10A$^{-/-}$/CaMKIIδc$^{+/T}$ compared to CaMKIIδc$^{+/T}$ cardiomyocytes. Data were analyzed by one-way ANOVA with the post hoc two-stage step-up method of Benjamini, Krieger, and Yekutieli. **g** Original confocal line scans images of CaMKIIδc$^{+/T}$ and SCN10A$^{-/-}$/CaMKIIδc$^{+/T}$ cardiomyocytes showing diastolic Ca$^{2+}$ waves. **h** Percentage of cells exhibiting waves was significantly less in SCN10A$^{-/-}$/CaMKIIδc$^{+/T}$ ($n = 74$ cells/7 mice) compared to CaMKIIδc$^{+/T}$ ($n = 104$ cells/9 mice). Data were analyzed by Chi-square test, two-tailed analysis. **i** Significantly decreased number of Ca$^{2+}$ waves per minute in SCN10A$^{-/-}$/CaMKIIδc$^{+/T}$ compared to CaMKIIδc$^{+/T}$. Data were analyzed by one-way ANOVA with post hoc Bonferroni's correction. Cells/mice studied, WT: $n = 48$ cells/5 mice, SCN10A$^{-/-}$: $n = 52$ cells/5 mice, CaMKIIδc$^{+/T}$: $n = 104$ cells/9 mice; SCN10A$^{-/-}$/CaMKIIδc$^{+/T}$: $n = 74$ cells/7 mice. Data were presented as mean values ± SEM.

knockout of Na$_V$1.8 resulted in a ~45% decrease in I$_{NaL}$ in SCN10A$^{-/-}$/CaMKIIδc$^{+/T}$ mice compared with CaMKIIδc$^{+/T}$. Of note, at basal/unstimulated conditions, I$_{NaL}$ did not differ between WT and SCN10A$^{-/-}$.

To evaluate whether chronic ablation of Na$_V$1.8 in CaMKIIδc$^{+/T}$ mice may reduce proarrhythmic cellular activity, we performed electrophysiological measurements (Fig. 5c–f). Patch-clamp experiments revealed that CaMKIIδc$^{+/T}$ cardiomyocytes exhibited approximately fivefold more delayed afterdepolarizations/min (DADs) compared to WT and SCN10A$^{-/-}$. *Scn10a* knockout reduced the fraction of CaMKIIδc-induced DADs/min by ~60% (Fig. 5c, d). Comparable observations were made regarding the occurrence of early afterdepolarizations (EADs, Fig. 5e, f). While WT and SCN10A$^{-/-}$ cardiomyocytes developed almost no EADs, 13.6 ± 3.2 EADs/min were recorded in CaMKIIδc$^{+/T}$. Na$_V$1.8 knockout caused an 80% reduction of EADs/min in these cells. A detailed description of the action potential characteristics of these measurements is provided in the Supplementary information (Supplementary Table 1).

Furthermore, to evaluate whether inhibition of Na$_V$1.8 may decrease the number of cardiomyocytes exhibiting Ca$^{2+}$-derived proarrhythmic events, we quantified the occurrence of Ca$^{2+}$ waves in mouse ventricular cardiomyocytes. The fraction of cardiomyocytes developing Ca$^{2+}$ waves was significantly less in SCN10A$^{-/-}$/CaMKIIδc$^{+/T}$ versus CaMKIIδc$^{+/T}$ (14.8 vs 27.8%, Fig. 5g, h). However, some cardiomyocytes showed more than one Ca$^{2+}$ wave. Therefore, we calculated the occurring events per minute. In CaMKIIδc$^{+/T}$ the frequency of Ca$^{2+}$ waves was ~2.5-fold higher compared to WT and SCN10A$^{-/-}$ and was reduced by ~55% in cardiomyocytes from SCN10A$^{-/-}$/CaMKIIδc$^{+/T}$ compared to CaMKIIδc$^{+/T}$ (Fig. 5i). Of note, Ca$^{2+}$ transient amplitude and Ca$^{2+}$ extrusion from cytosol was unchanged between SCN10A$^{-/-}$/CaMKIIδc$^{+/T}$ and CaMKIIδc$^{+/T}$ (Supplementary information, Supplementary Fig. 9). In summary, we describe a cellular rescue of the proarrhythmic CaMKIIδc$^{+/T}$ phenotype due to genetic ablation of Na$_V$1.8, which is associated with improved animal survival.

### *Scn10a* knockout reduces ventricular arrhythmias in CaMKIIδc$^{+/T}$ mice.

To further investigate whether the proarrhythmic potential of *Scn10a* knockout on the cellular level is translatable into the in vivo situation, we implanted telemetric monitors into 8 week old SCN10A$^{-/-}$/CaMKIIδc$^{+/T}$ and CaMKIIδc$^{+/T}$ mice. After 10 days of recovery from surgery, telemetric measurements were performed twice a week for 24 h over a period of 2 weeks. Baseline ECG characteristics are presented in the Supplementary information (Supplementary Table 2). There was no relevant difference in overall physical animal activity between the groups (Fig. 6b). Blinded analysis revealed that SCN10A$^{-/-}$/CaMKIIδc$^{+/T}$ showed a strong trend towards a reduction of premature ventricular contractions (PVC) by ~93% compared to CaMKIIδc$^{+/T}$ (Fig. 6a, c, p = 0.08). Most importantly, the incidence of ventricular tachycardia (VT) was significantly lower (91%) in SCN10A$^{-/-}$/CaMKIIδc$^{+/T}$ (Fig. 6a, d). Therefore, the observed improved survival of SCN10A$^{-/-}$/CaMKIIδc$^{+/T}$-animals is associated with reduced ventricular arrhythmias.

### Discussion

In our present study, we demonstrate that CaMKIIδc interacts with the neuronal sodium channel Na$_V$1.8 in human ventricular cardiomyocytes. Using different approaches in human and mouse cardiomyocytes, we demonstrated the relevance of Na$_V$1.8 for I$_{NaL}$ generation in HF and that an enhanced CaMKIIδc, indeed, regulates this Na$_V$1.8-driven I$_{NaL}$. Isolated cardiomyocytes from SCN10A$^{-/-}$/CaMKIIδc$^{+/T}$ compared to CaMKIIδc$^{+/T}$ mice

exhibit reduced cellular arrhythmic events. While there was no change with respect to HF progression, i.e., similar left ventricular ejection fraction and chamber diameters, we found reduced ventricular arrhythmias and an improved animal survival of SCN10A$^{-/-}$/CaMKIIδc$^{+/T}$ animals. Thus, we identified a modifiable proarrhythmic CaMKIIδc downstream target in the failing heart.

We recently found that Na$_V$1.8 is upregulated and thereby contributes to I$_{NaL}$ under conditions of HF and cardiac hypertrophy where CaMKIIδc activity is known to be enhanced[31,32]. Therefore, the aim of the current study was to investigate a potential crosstalk between Na$_V$1.8 and CaMKIIδc and its consequences on I$_{NaL}$ generation and cellular arrhythmogenety. Accordingly, we revealed an interaction of Na$_V$1.8 and CaMKIIδc in human ventricular myocardium of both non-failing and HF samples. CaMKIIδc is known to also interact with Na$_V$1.5 channels at the intercalated disc where Na$_V$1.5 and CaMKIIδc are part of a macro complex with Ankyrin-G and ßIV-spectrin[36]. The interaction with CaMKIIδc in this complex influences Na$_V$1.5 channel function in several cardiac disease states such as HF[14,21]. As Na$_V$1.8 was previously shown to interact with Ankyrin-G in neurons[37] in a similar fashion as it is known for Na$_V$1.5 in cardiomyocytes, similar mechanisms of Na$_V$1.8 interaction with CaMKIIδc like known for Na$_V$1.5 are conceivable. Several Serin residues as well as CaMKII-binding consensus motifs were found to be conserved between Na$_V$1.5 and Na$_V$1.8 (for details see Supplementary Fig. 10).

In a variety of cardiac pathologies, enhanced CaMKIIδc activity and expression is a key contributor to maladaptive electrical remodeling and thereby promotes arrhythmias[9,38–40]. CaMKIIδc can influence I$_{NaL}$ magnitude by phosphorylating Na$_V$, which has been exclusively investigated for cardiac Na$_V$1.5 before, whereas a possible CaMKIIδc-dependent regulation of Na$_V$1.8 has not been investigated yet[14,21,22]. In our study, we investigated the contribution of Na$_V$1.8 to I$_{NaL}$ in an HF model that was exclusively induced by chronic CaMKIIδc overexpression. As previously shown, I$_{NaL}$ was augmented in cardiomyocytes from CaMKIIδc$^{+/T}$ compared to WT[21,40], while these effects were ameliorated by the application of specific Na$_V$1.8 blockers. Therefore, at least a relevant part of CaMKIIδc-induced I$_{NaL}$ appears to be driven by Na$_V$1.8. Additional support for this conclusion comes from our I$_{NaL}$ measurements from human failing cardiomyocytes where CaMKIIδc activity and I$_{NaL}$ are both known to be increased in parallel[2,13,16,21,41]. Inhibition of either CaMKIIδc or Na$_V$1.8 reduced I$_{NaL}$ as separately demonstrated before[21,31]. However, simultaneous inhibition of Na$_V$1.8 and CaMKIIδc had no additive effect compared to CaMKIIδc inhibition alone. Therefore, it can be assumed that the majority of Na$_V$1.8-driven I$_{NaL}$ was already abolished by CaMKIIδc inhibition and hence, seems to be CaMKIIδc-dependent. In addition, *Scn10a* knockout in CaMKIIδc$^{+/T}$ mice resulted in a reduction in I$_{NaL}$ comparable to Na$_V$1.8 inhibition upon specific blockers.

In previous studies, the functional relevance of Na$_V$1.8 expression in cardiomyocytes was questioned as an application of specific blockers had no effects on peak I$_{Na}$ and I$_{NaL}$ in healthy and unstimulated cardiomyocytes[42,43]. These data are not in conflict with our findings, as we also did not observe differences in I$_{NaL}$ magnitude in cardiomyocytes from healthy mice, while clear effects were evident under conditions of enhanced I$_{NaL}$ either by chronic CaMKIIδc overexpression or isoproterenol treatment of iPSC-cardiomyocytes. Therefore, the interaction of Na$_V$1.8 with enhanced CaMKIIδc activity might be necessary to generate meaningful effects on cardiomyocyte electrophysiology while Na$_V$1.8 appears to play a negligible role in healthy cardiomyocytes. This establishes Na$_V$1.8 to be a disease-specific target.

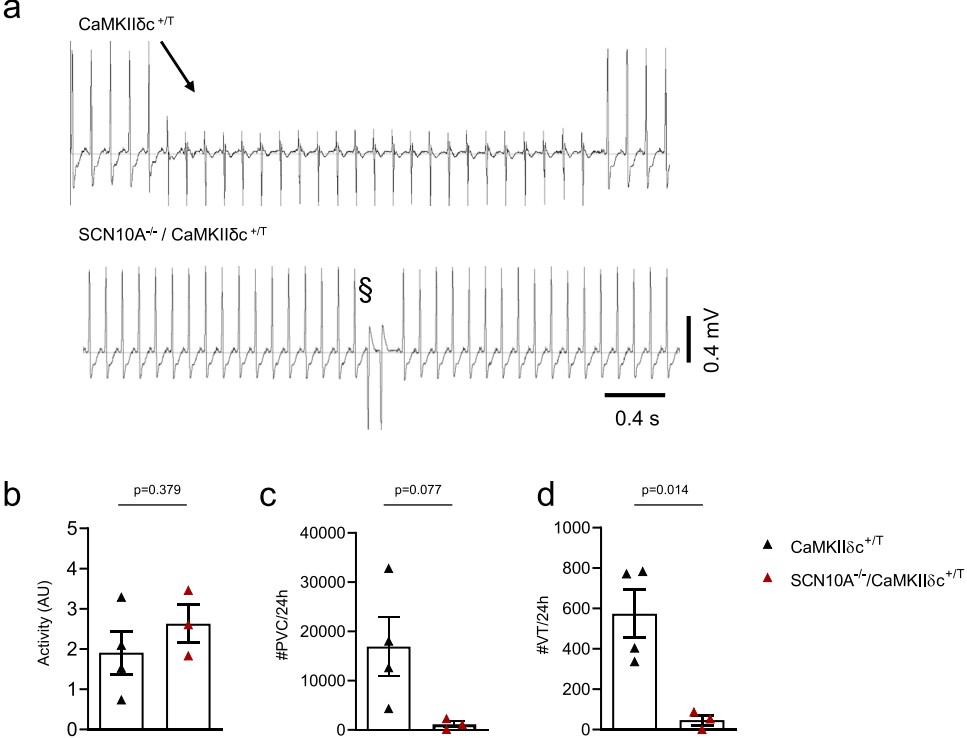

**Fig. 6 SCN10A$^{-/-}$/CaMKIIδc$^{+/T}$ exhibit less in vivo arrhythmias compared to CaMKIIδc$^{+/T}$ mice. a** Original ECG traces from telemetry recordings of 10-week-old CaMKIIδc$^{+/T}$ and SCN10A$^{-/-}$/CaMKIIδc$^{+/T}$ mice showing ventricular arrhythmias. **b** Unchanged activity levels in SCN10A$^{-/-}$/CaMKIIδc$^{+/T}$ compared to CaMKIIδc$^{+/T}$ (CaMKIIδc$^{+/T}$ (four mice); SCN10A$^{-/-}$/CaMKIIδc$^{+/T}$ (three mice), four individual recordings each. Data were presented as mean values ± SEM, analyzed by unpaired two-tailed Student's *t*-test. **c** Mean values of premature ventricular contractions (PVCs) in SCN10A$^{-/-}$/CaMKIIδc$^{+/T}$ ($p = 0.08$, Unpaired two-tailed Student's *t*-test), CaMKIIδc$^{+/T}$ (four mice); SCN10A$^{-/-}$/CaMKIIδc$^{+/T}$ (three mice), four individual recordings each. Data were presented as mean values ± SEM. **d** Reduction of ventricular tachycardia (VT) incidence in SCN10A$^{-/-}$/CaMKIIδc$^{+/T}$ ($p < 0.05$, Unpaired two-tailed Student's *t*-test), CaMKIIδc$^{+/T}$ (four mice); SCN10A$^{-/-}$/CaMKIIδc$^{+/T}$ (three mice), four individual recordings each. Data were presented as mean values ± SEM.

An augmentation of $I_{NaL}$ was demonstrated to cause a Na$^+$-dependent Ca$^{2+}$ overload and spontaneous SR-Ca$^{2+}$ release providing a substrate for cellular proarrhythmia[8,15,30,44]. We previously reported that inhibition of $I_{NaL}$ by specifically targeting Na$_V$1.8 is potent enough to reduce NCX reverse mode and thereby diastolic SR-Ca$^{2+}$ leak[11,31,32]. In the present work, we inhibited either CaMKIIδc or/and Na$_V$1.8 in failing humans and CaMKIIδc$^{+/T}$ mouse ventricular cardiomyocytes and correspondingly observed a decrease in $I_{NaL}$ and diastolic SR-Ca$^{2+}$ leak. A significant reduction of diastolic SR-Ca$^{2+}$ release events was already prominent after Na$_V$1.8 inhibition alone in failing human cardiomyocytes which was almost comparable to the effect caused by CaMKIIδc inhibition. Moreover, we did not observe a further reduction in SR-Ca$^{2+}$ leak when AIP was co-administrated with the Na$_V$1.8 blocker suggesting that effects of Nav1.8 inhibition on SR-Ca$^{2+}$ leak act via indirect inhibition of CaMKIIδc as it was previously shown for $I_{NaL}$ inhibition with TTX or Ranolazine[8,15]. The present findings, therefore confirm that Na$_V$1.8 inhibition abrogates the vicious proarrhythmic cycle between enhanced $I_{NaL}$ and CaMKIIδc.

Increased expression and enhanced activity of CaMKIIδc is not only known to increase proarrhythmic triggers in HF but is also associated with a strong HF phenotype and increased mortality in mice[9,13,16,45]. Our current findings illustrate that Na$_V$1.8 plays a relevant role for arrhythmogenesis under conditions of enhanced CaMKIIδc activity. For deeper mechanistic analysis, we crossbred our CaMKIIδc$^{+/T}$ mice with *Scn10a* knockout mice. In fact, we observed an improved survival in these SCN10A$^{-/-}$/CaMKIIδc$^{+/T}$ mice but still with an unchanged typical phenotype of dilated

cardiomyopathy in these blinded investigations. There may be three potential explanations for the improved survival, which were either reduced lethal arrhythmias, pump failure, or a combination of both. As we could demonstrate that induction and progression of HF is not influenced by the additional *Scn10a* knockout we propose a reduction of augmented $I_{NaL}$ with subsequent lower proarrhythmic activity to constitute the underlying mechanism of this improved survival. The link between enhanced $I_{NaL}$ and increased arrhythmia risk is rather complex. On the one hand, enhanced $I_{NaL}$ prolongs APD and may therefore give rise to the formation of EADs. On the other hand, enhanced $I_{NaL}$ causes Na$^+$ overload of the cardiomyocyte subsequently leading to Ca$^{2+}$ overload by activating NCX reverse mode[6,8,11,15]. This may trigger the occurrence of diastolic SR-Ca$^{2+}$ release and DADs due to enhanced CaMKIIδc-dependent RyR2 phosphorylation[8,10,15]. In our experiments, a reduction of $I_{NaL}$ in SCN10A$^{-/-}$/CaMKIIδc$^{+/T}$ compared to CaMKIIδc$^{+/T}$ cardiomyocytes was observed. This was clearly associated with a reduction in the occurrence of EADs, DADs and diastolic SR-Ca$^{2+}$-release events, thereby illustrating that the CaMKIIδc-induced proarrhythmic phenotype can be ameliorated on the cellular level by Na$_V$1.8 knockout.

In a recent study by our groups, we could demonstrate that selective inhibition of diastolic SR-Ca$^{2+}$ leak by the compound Rycal S36 improves survival in an HF mouse model, where $I_{NaL}$ and CaMKIIδc activity were also described to be enhanced[10,46]. As in our study, improved survival was caused by a significant reduction of malignant arrhythmias, while the severity of HF was unchanged. Likewise, we observed less ventricular arrhythmias in SCN10A$^{-/-}$/CaMKIIδc$^{+/T}$ mice in vivo. It is well known that

individuals with structural heart disease are at increased risk for sustained ventricular tachycardia and ventricular fibrillation[47]. Moreover, sustained ventricular tachycardia may degenerate to ventricular fibrillation[47]. In addition, clinical trials have shown an association of PVCs and ventricular tachycardia with adverse outcomes in patients with dilated cardiomyopathy[47–49]. The $Na_V1.8$ knockout significantly reversed a relevant part of the arrhythmogenic CaMKIIδc transgenic substrate in vitro and in vivo which is known to be associated with sudden cardiac death. However, due to the sporadic nature of sudden cardiac death, we were not able to correlate mortality with ventricular arrhythmias in these CaMKIIδc transgenic mice as this is technically and ethically not feasible. We are also not aware of any other experimental trial that investigates such a high number of HF mice with implanted telemetric recorders during death struggle. Finally, a chicken or the egg question would remain, as detected rhythm disorders cannot be differentiated to be either caused primary by arrhythmia or secondary due to hypoxia.

Persistent CaMKIIδc hyperactivity in hearts of HF patients has several detrimental effects such as influencing myocardial remodeling and promoting arrhythmias[33,39]. Despite the fact that several CaMKIIδc inhibitors have been demonstrated to effectively reduce arrhythmogenesis in HF in vitro[46,50], CaMKIIδc inhibition to manage cardiac arrhythmias has not been established in patients so far. This can be explained by the vast signaling network of this enzyme, which is required to keep a balance between pivotal off-target effects and the therapeutic benefits. Any other alternative downstream or upstream target of the CaMKIIδc pathway, which is involved in promoting arrhythmias may be a more specific therapeutic option. As it has consistently been shown that activation of CaMKIIδc is involved in enhancement of $I_{NaL}$ and vice versa[10,11,21,34,40] our proposed crosstalk between CaMKIIδc and $Na_V1.8$ causing enhanced $I_{NaL}$ in the presence of hyperactive CaMKIIδc may be a novel strategy to prevent detrimental CaMKIIδc-induced effects on cellular electrophysiology. Most importantly, we and others have shown that selective $Na_V1.8$ inhibition does not influence either peak $Na^+$ current or $Na^+$ current kinetics and thereby cardiac conduction as another critical regulator of proarrhythmia[11,29,31,42,51].

In summary, the results of our current study demonstrate that increased CaMKIIδc activity in HF contributes to enhanced $I_{NaL}$ via interaction with $Na_V1.8$. This augmented $I_{NaL}$ generated by $Na_V1.8$ detrimentally influences cellular electrophysiology by increasing RyR2-leakiness and can give rise to cellular proarrhythmic events in HF cardiomyocytes. Importantly, the results of the present study demonstrate that pharmacological inhibition and genetic ablation of $Na_V1.8$ can specifically reverse these detrimental proarrhythmic effects in the human HF heart and in our SCN10A$^{-/-}$/CaMKIIδc$^{+/T}$ mouse model. This is associated with improved animal survival. Therefore, targeting $Na_V1.8$ as a specific substrate of increased CaMKIIδc activity may constitute a promising antiarrhythmic approach in HF which merits further translational investigation.

## Methods

**Human tissue samples**. The study conforms to the declaration of Helsinki and was approved by the local ethics committee. All participants were informed about the study prior to inclusion. All patients signed informed consent. We obtained left ventricular tissues from explanted hearts of patients with end-stage HF (NYHA HF classification IV) who were undergoing heart transplantation. After explantation, the whole heart or myocardial tissues were immediately placed in a container having precooled cardioplegic solution containing (in mmol/L): NaCl 110, KCl 16, MgCl$_2$ 16, NaHCO$_3$ 16, CaCl$_2$ 1.2, and glucose 11. Myocardial samples for Western blot and co-immunoprecipitation were immediately frozen in liquid nitrogen and stored at −80 °C. The heart tissue for cell isolation was stored in cooled cardioprotective solution containing (in mmol/L): 156 Na$^+$, 3.6 K$^+$, 135 Cl$^-$, 25 HCO$_3^-$, 0.6 Mg$^{2+}$, 1.3 H$_2$PO$_4^-$, 0.6 SO$_4^{2-}$, 2.5 Ca$^{2+}$, 11.2 glucose, and 10 2,3-butane-dionmonoxime (BDM) aerated with 95% O$_2$ and 5% CO$_2$. We used healthy

myocardium from healthy donor hearts that could not be transplanted for technical reasons as controls co-immunoprecipitation. Patient characteristics are presented in the Supplementary information (Supplementary Table 3). The study was approved by the local ethics committee of the University Medicine of Goettingen.

**CaMKIIδc transgenic mice**. CaMKIIδc transgenic (CaMKIIδc$^{+/T}$) mice were generated using an α-MHC promoter[13]. We used 12- to 14-week-old mice for electrophysiology experiments. The animal investigations conform to the "Guide for the Care and Use of Laboratory Animals" published by the US National Institutes of Health (8th edition, revised 2011) and the guidelines from Directive 2010/63/EU of the European Parliament on the protection of animals used for scientific purposes. The mouse study was approved by the local ethics committee of the University Medicine of Goettingen and the public authority on animal welfare.

**Generation of SCN10A$^{-/-}$/CaMKIIδc$^{+/T}$ mice**. To generate the SCN10A$^{-/-}$/CaMKIIδc$^{+/T}$ mouse model we crossbred CaMKIIδc$^{+/T}$ mice from a Black-Swiss strain with $Na_V1.8$ knockout mice (SCN10A$^{-/-}$) from a C57BL/6 strain. Eight weeks old male CaMKIIδc$^{+/T}$ mice were mated with 8 weeks old female SCN10A$^{-/-}$ mice. This breeding resulted in a genotype carrying a WT or transgenic (TG) allele of the CaMKIIδc gene and a heterozygote knockout of *Scn10a*. From this first generation, 8 weeks old male mice carrying a TG CaMKIIδc allele (CaMKIIδc$^{+/T}$) were mated with 8 weeks old female mice carrying homozygous WT CaMKIIδc alleles and heterozygous for *Scn10a* gene. This breeding produced 25% mice with the desired genotype having heterozygous CaMKIIδc gene (CaMKIIδc$^{+/T}$) and homozygous *Scn10a* knockout (SCN10A$^{-/-}$/CaMKIIδc$^{+/T}$). Mice carrying a heterozygote *Scn10a* gene were excluded from survival analysis and experimental studies. A detailed scheme of the crossbreeding is displayed in the Supplementary information (Supplementary Fig. 11). The breeding rooms were maintained at 20–22 °C with 50–60% humidity. Mice were housed in a room with a 12-h light/dark cycle with ad libitum access to water and food.

Survival of CaMKIIδc$^{+/T}$ mice was documented in a database of the local facility of animal experiments. A survival curve was prepared from this database and a comparison of survival was made between the groups CaMKIIδc$^{+/T}$ and SCN10A$^{-/-}$/CaMKIIδc$^{+/T}$. Animals used for experiments were excluded from survival analysis. Before scarifying animals, body weight was recorded while after explantation heart weight and tibia length were measured to tabulate animal size. Some of the animals showed signs of heart failure with fluid retention while some had signs of terminal illness therefore, heart weight to tibia length ratio was analyzed instead of heart weight to body weight ratio. The mouse study was approved by the local ethics committee of the University Medicine of Goettingen and the public authority on animal welfare.

**Generation of homozygous knockout iPSCs using CRISPR/Cas9 and directed differentiation into iPSC-cardiomyocytes**. All procedures were performed according to the Declaration of Helsinki and were approved by the local ethics committee. Informed consent was signed by all tissue donors. Gene editing was conducted using two CRISPR guide RNAs (gRNA1: GTGACTCCGGAG-TAAAGCGACGG and gRNA2: ACGGAAGTTGTTAGTTTCGAGG, designed with IDTdna.com design tool) targeting *SCN10A* exon1. About $2 \times 10^6$ wild-type iPSCs were electroporated with 2.5 μl (100 μM) of each gRNA, 5 μl tracrRNA (100 μM), 2 μl Cas9 protein (10 ng/l, IDT), and 1 μl electroporation enhancer (100 μM, IDT) using the Amaxa Nucleofection II Device (Lonza, program B-016) and the corresponding Human Stem Cell Nucleofection Kit (Amaxa, VPH-5022). Electroporated iPS cells were expanded and analyzed by Sanger sequencing (Microsynth). After additional singularization two identical homozygous *SCN10A* knockout clones (K62.1 and K62.4) were chosen for pluripotency analysis and cardiac differentiation. Two $Na_V1.8$ knockout iPSC-lines (K62.1 and K62.4) and the corresponding isogenic control iPSC- line[52] were cultured feeder-free and adherent by cultivation on Geltrex-coated cell culture dishes in the presence of chemically defined medium E8 (Life Technologies).

Directed cardiac differentiation of iPSCs was done by manipulation of the Wnt signaling pathway[52]. Briefly, iPSCs were cultured in E8 medium as a monolayer on Geltrex-coated 12-well plates to a confluence of 85–95%. Wnt signaling was initiated by a medium change to RPMI1640 GlutaMAX (Thermo Fisher Scientific) supplemented with human recombinant albumin, L-ascorbic acid 2-phosphate including the GSK3β inhibitor CHIR99021 (4 μmol/L, Millipore). After 48 h, cells were treated with fresh medium supplemented with the inhibitor of Wnt production 2 (IWP2) (5 μmol/L, Millipore) for 48 h. On day 6, the medium was changed to RPMI1640 GlutaMAX with 1x B27 with insulin (Thermo Fisher Scientific) with a medium change every 2–3 days. Cardiomyocytes were purified by using a metabolic selection for 2–4 days with 4 mmol/L lactate as carbon source after 15–20 days of differentiation[53]. Cells were cultured for 60 days and passaged onto glass-bottom FluoroDishes (WPI, 30 K/dish) by trypsinizing for 3 min at 37 °C. Cells settled for 7 days prior to further measurements with medium change every 2 days. IPSC-cardiomyocytes were analyzed 8–10 weeks after initiation of differentiation except when mentioned otherwise. Following differentiation, purity of iPSC-CM was determined by flow analysis (>90% cardiac TNT+) and qPCR of ventricular cardiomyocytes marker MLC2v. Four to five differentiation

experiments into ventricular iPSC-CMs of two $Na_V1.8$ knockout lines and the corresponding healthy isogenic control line were used.

## Cardiomyocyte isolation

*Human.* Human myocardium was rinsed, cut into small pieces, and incubated at 37 °C in a spinner flask filled with Joklik-MEM solution (JMEM; AppliChem, Darmstadt, Germany) containing 1.0 mg/mL collagenase (type 1, 185 U/mg; Worthington Biochem, New Jersey, USA) and trypsin (2.5 g/L, Life Technologies, Carlsbad, California, USA). After 45 min of digestion, the supernatant was discarded, and the tissue was incubated with fresh JMEM solution containing only collagenase. The solution was incubated for 10 to 20 min until cardiomyocytes were disaggregated using a Pasteur pipette. The supernatant containing disaggregated cells was removed and centrifuged (700 r.p.m., 5 min). Fresh JMEM with collagenase was added to the remaining tissue. This procedure was repeated 4 to 7 times. After every step, the centrifuged cells were resuspended in JMEM medium containing (in mmol/L): 10 taurine, 70 glutamic acid, 25 KCl, 10 $KH_2PO_4$, 22 dextrose, 0.5 EGTA, and 10% bovine calf serum (pH 7.4 adjusted with KOH at room temperature). Only cell solutions containing elongated, non-granulated cardiomyocytes with cross-striations were selected for experiments, plated on laminin-coated recording chambers, and allowed to settle for 30 min.

*Mouse.* Explanted mouse hearts from male and female wild-type (WT), SCN10A$^{−/−}$, CaMKIIδc$^{+/T}$, and SCN10A$^{−/−}$/CaMKIIδc$^{+/T}$ mice were retrogradely perfused with a nominally $Ca^{2+}$-free Tyrode's solution containing (in mmol/L): 113 NaCl, 4.7 KCl, 0.6 $KH_2PO_4$, 0.6 $Na_2HPO_4 \times 2H_2O$, 1.2 $MgSO_4 \times 7H_2O$, 12 $NaHCO_3$, 10 $KHCO_3$, 10 HEPES, 30 taurine, 10 BDM, 5.5 glucose, and 0.032 phenol red (pH 7.4, with NaOH at 37 °C) using a Langendorff perfusion apparatus. Then, 0.05 mg/mL liberase TM (Roche Diagnostics, Mannheim, Germany), 0.014% trypsin, and 0.1 mmol/L $CaCl_2$ were added to the perfusion solution. Once the heart became flaccid, ventricular tissue was removed, cut into small pieces, and dispersed. $Ca^{2+}$ reintroductions were performed carefully by increasing $Ca^{2+}$ stepwise from 0.1 to 0.4 mmol/L for the patch-clamp and to 1.6 mmol/L for $Ca^{2+}$ spark experiments. Cells were plated on laminin-coated chambers, allowed to settle for 15 min, and then used for measurements.

**Co-immunoprecipitation.** Tissue homogenates from human ventricular myocardium were suspended in lysis buffer containing (1% CHAPS in RIPA buffer containing (in mmol/L): 50 Tris-HCl, 120 NaCl, 200 NaF, 1 $Na_3VO_4$, 1 DTT, (pH 7.4), and complete protease and phosphatase inhibitor cocktails (Roche Diagnostics). Protein concentration was determined by a BCA assay. CaMKIIδc was immunoprecipitated with rabbit polyclonal anti-CaMKIIδ antibody (3 μg/500 μg of protein; preincubation at 4 °C overnight, a gift from Prof. Donald M. Bers, Department of Pharmacology, University of California Davis, USA) and protein G Plus Agarose (prewashed and blocked with 5% BSA overnight at 4 °C; Pierce, #28851) for 2 h and 30 min at 4 °C. As a control, rabbit anti-control IgG (3 μg/ 500 μg protein Santa Cruz, sc-2027) was used and a negative control without antibody. The pellets were collected and washed with the RIPA buffer without CHAPS 3 times. The immunoprecipitated proteins were eluted in 2x Laemmli sample buffer containing β-mercaptoethanol (10 min at 70 °C). Supernatants were subjected to Western blotting to detect $Na_V1.8$ (mouse monoclonal, LSBio, LS-C109037), CaMKIIδ (rabbit polyclonal, Thermo Scientific, PA5-22168), and GAPDH (mouse monoclonal, Biotrend, BTMC-A437-9).

**Immunofluorescence staining of human isolated cardiomyocytes.** Isolated human ventricular cardiomyocytes were allowed to attach the poly-L-lysine coated glass chambers for 30 to 45 min at room temperature. Cardiomyocytes were fixed in ice-cold 99.2% ethanol for 20 min at −20 °C. After fixation, cardiomyocytes were washed with phosphate buffer saline (1X PBS) and subsequently blocked and permeabilized with 0.5% Triton X-100 and 5% BSA (bovine serum albumin) in PBS at room temperature.

After blocking, cells were washed with PBS (3x 5 min) and incubated with a primary antibody: mouse monoclonal anti-$Na_V1.8$ (1:50, LSBio, LS-C109037). The antibody was diluted in antibody diluent (Dako) containing 0.5% Triton X-100 and incubated overnight at 4 °C. On the next day, cells were washed with PBS (3x 10 min). After washing, cells were incubated for 1 h at room temperature in darkness with the fluorescent-conjugated secondary antibody: Alexa Fluor-488 goat anti-mouse (1:200, Life Technologies, A-11029). In the next step, cells were again incubated overnight at 4 °C with a primary antibody: rabbit polyclonal anti-CaMKIIδ (1:100, gift from D. M. Bers, Department of Pharmacology, University of California Davis, USA). After washing with PBS (3x 5 min) cells were incubated for 1 h at room temperature in darkness with a fluorescent-conjugated secondary antibody: Alexa Fluor-555 goat anti-rabbit (1:200, Life Technologies, A-21424). After washing, cells were covered with Vectashield HardSet Mounting Medium (Vector Laboratories) and viewed using an LSM 5 Pascal confocal microscope (Zeiss) with a 40x oil immersion objective.

**Immunofluorescence of iPSCs and iPSC-cardiomyocytes.** $Na_V1.8$ KO-iPSCs and iPSC-CMs were fixed with 4% Histofix solution (Sigma) for 20 min at room temperature and subsequently blocked in 1% BSA overnight at 4 °C. The primary antibodies mouse anti-SSEA4 (1:200, Abcam), mouse anti-α-actinin (1:750, Sigma),

rabbit anti-Titin-M8/M9 (1:750, MyoMedix) were incubated overnight at 4 °C. The fluorescently labeled secondary antibodies were added for 1 h at 37 °C (AF488 donkey anti-mouse IgG (1:1000, Invitrogen); AF555 donkey anti-rabbit IgG (1:750, Invitrogen). Images were acquired with the Axiovert 200 fluorescence microscope (Zeiss) and the Axiovision software (Rel 4.8).

## Patch-clamp experiments

*$I_{NaL}$ measurements.* To measure $I_{NaL}$ in human and mouse ventricular cardiomyocytes, a ruptured whole-cell patch-clamp was performed at room temperature. The resistance of the pipette was between 2 and 3 mega-Ohm (MΩ) when filled with pipette solution containing (in mmol/L): 95 CsCl, 40 Cs-glutamate, 10 NaCl, 0.92 $MgCl_2$, 5 Mg-ATP, 0.3 Li-GTP, 5 HEPES, 0.03 niflumic acid (to block $Ca^{2+}$-activated chloride current), 0.02 nifedipine (to block $Ca^{2+}$ current), 0.004 strophanthidin (to block Na$^+$/K$^+$ ATPase current) 1 EGTA, and 0.36 $CaCl_2$ (free $[Ca^{2+}]_i$,100 nmol/L) (pH 7.2 with CsOH at room temperature). Cardiomyocytes were maintained in the bath solution containing (in mmol/L): 135 NaCl, 5 tetramethylammonium chloride, 4 CsCl, 2 $MgCl_2$, 10 glucose, 10 HEPES (pH 7.4 with CsOH at room temperature). To minimize contaminating $Ca^{2+}$ currents during $I_{NaL}$ measurements, $Ca^{2+}$ was omitted from the bath solution. $I_{NaL}$ was measured only in those cardiomyocytes where a seal of more than 1 giga-Ohm was achieved and the access resistance remained <7 MΩ. When whole-cell patch configuration was achieved, cardiomyocytes were given a period of 3 min to be stabilized before conducting measurements. Then cardiomyocytes were held at −120 mV before depolarization to −35 mV for a duration of 1000 ms with 10 pulses and a basic cycle length of 2 s. $I_{NaL}$ was measured as integral current amplitude between 100 and 500 ms and was normalized to the membrane capacitance. Mouse cardiomyocytes were incubated with either A-803467 (30 nmol/L, Sigma) or PF-01247324 (1 μmol/L, Sigma) for 15 min before starting $I_{NaL}$ measurements. Human cardiomyocytes were incubated with autocamptide-2-related inhibitor peptide (AIP, 1 μmol/L), PF-01247324 (1 μmol/L, Sigma), or PF-01247324+AIP for 15 min. For control groups, cardiomyocytes were incubated for 15 min in a normal bath solution to equilibrate the measurements.

*Current–voltage relationship of $I_{NaL}$.* Ruptured whole-cell patch-clamp experiments were applied to establish the current–voltage relationship of $I_{NaL}$. The pipette resistance ranged between 1.5 and 3.5 MΩ when filled with the pipette solution containing (in mmol/L): 95 CsCl, 40 Cs-glutamate, 10 NaCl, 0.92 $MgCl_2$, 5 Mg-ATP, 0.3 Li-GTP, 5 HEPES, 0.03 niflumic acid (to block $Ca^{2+}$-activated chloride current), 0.02 nifedipine (to block $Ca^{2+}$ current), 0.004 strophanthidin (to block Na$^+$/K$^+$ ATPase current), 1 EGTA, and 0.36 $CaCl_2$ (free $[Ca^{2+}]_i$,100 nmol/L) (pH 7.2 with CsOH at room temperature). The bath solution contained (in mmol/L): 135 NaCl, 5 tetramethylammonium chloride, 4 CsCl, 2 $MgCl_2$, 10 glucose, 10 HEPES (pH 7.4 with CsOH at room temperature). In some experiments, cardiomyocytes were incubated with PF-01247324 (1 μmol/L, Sigma) for 15 min before initiation of $I_{NaL}$ measurements. After patch rupture cardiomyocytes were allowed to stabilize for at least 3 min. Currents were elicited using a voltage step protocol with 10 mV increments ranging from −120 to +30 mV (step duration: 1 s, holding potential: −120 mV). $I_{NaL}$ was determined as the mean current density between 180 to 190 ms of each depolarization step. PF-sensitive $I_{NaL}$ was calculated as the difference between the mean $I_{NaL}$ densities of the vehicle and PF-treated cardiomyocytes at each membrane potential.

*Peak Na$^+$ current measurements.* Ruptured-patch whole-cell voltage-clamp was used to measure $I_{Na}$ in mouse cardiomyocytes with microelectrodes (2–3 MΩ, room temperature). Pipettes were filled with (in mmol/L): 80 CsCl, 40 Cs-glutamate, 5 NaCl, 0.92 $MgCl_2$, 5 Mg-ATP, 0.3 Li-GTP, 5 HEPES, 0.03 niflumic acid (to block $Ca^{2+}$-activated chloride current), 0.02 nifedipine (to block $Ca^{2+}$ current), 0.004 strophanthidin (to block Na$^+$/K$^+$ ATPase current), 1 EGTA, and 0.36 $CaCl_2$ (free $[Ca^{2+}]_i$,100 nmol/L) (pH 7.2 with CsOH at room temperature). Cardiomyocytes were maintained in the bath solution containing (in mmol/L): 5 NaCl, 135 tetramethylammonium chloride, 4 CsCl, 2 $MgCl_2$, 10 glucose, 10 HEPES, 0.4 $CaCl_2$ (pH 7.4 with KOH at room temperature). $I_{Na}$ was measured only in those cardiomyocytes where a seal of more than 1 Giga-Ohm was achieved and the access resistance remained <7 MΩ. Mouse cardiomyocytes were incubated with PF-01247324 (1 μmol/L, Sigma) for 15 min before starting $I_{Na}$ measurements. In all experiments, myocytes were mounted on the stage of a microscope (Nikon Eclipse TE2000-U). Liquid junction potentials were corrected with the pipette in the bath. Membrane capacitance and series resistance were compensated after patch rupture. Data were collected using Patchmaster 2.0 (HEKA Elektronik).

*Membrane potential recordings.* The whole-cell patch-clamp technique in current-clamp configuration was used to measure membrane potential in single isolated cardiomyocytes. Microelectrodes (3–5 MΩ, room temperature) were filled with (in mmol/L): 92 K-Aspartate, 48 KCl, 1 Mg-ATP, 10 HEPES, 0.02 EGTA, 0.1 GTP-Tris, and 4 Na2-ATP (pH 7.2, KOH). The bath solution contained (in mmol/L): 140 NaCl, 4 KCl, 1 $MgCl_2$, 2 $CaCl_2$, 10 glucose, and 10 HEPES (pH 7.4, NaOH). Action potentials were continuously elicited by square current pulses of 1–2 nA amplitude and 1–5 ms duration at increasing stimulation frequency (0.5–3 Hz). Access resistance was typically ~5–15 MΩ after patch rupture. Fast capacitance was compensated for in a cell-attached configuration. Recordings were commenced

after cell stabilization, which was ~5 min after rupture. Data were collected using Patchmaster 2.0 (HEKA Elektronik) and was analyzed using LabChart 7.37 (ADInstruments). Afterdepolarizations such as EADs and DADs were counted under continuous stimulation with 0.5 Hz. An EAD was counted as an EAD when a renewed depolarization of at least 1 mV occurred before complete membrane repolarization. Criteria for counting DADs were similar, but after achieving complete membrane repolarization.

**$Ca^{2+}$ spark and $Ca^{2+}$ wave measurement.** Human and mouse ventricular cardiomyocytes were loaded with a Fluo-4AM (10 μmol/L for 15 min, Molecular Probes) at room temperature. For some experiments, either A-803467 (30 nmol/L for 15 min, Sigma) or PF-01247324 (1 μmol/L for 15 min, Sigma) was added to the loading buffer to inhibit $Na_V1.8$. Cells were perfused with Tyrode's solution containing (in mmol/L): 136 NaCl, 4 KCl, 0.33 $NaH_2PO_4$, 4 $NaHCO_3$, 2 $CaCl_2$, 1.6 $MgCl_2$, 10 HEPES, 10 glucose (pH 7.4 adjusted with NaOH at room temperature) and the respective active agent. The cardiomyocytes were washed with Tyrode's solution for 15 min for de-esterification of the $Ca^{2+}$ dye. $Ca^{2+}$ spark measurements were performed with a laser scanning confocal microscope (LSM 5 Pascal, Zeiss) using a 40× oil immersion objective. Fluo-4AM was excited by an argon ion laser (488 nm), and emitted fluorescence was collected through a 505-nm long-pass emission filter. Fluorescence images were recorded in the line scan mode with 512 pixels per line (width of each scanline: 38.4 μm) and a pixel time of 0.64 μs. One image consists of 10,000 unidirectional line scans, equating to a measurement period of 7.68 s. Experiments were conducted under resting conditions after loading the sarcoplasmic reticulum with $Ca^{2+}$ by field stimulation at 1 Hz and 20 V. $Ca^{2+}$ sparks were analyzed with the program SparkMaster for ImageJ. The mean spark frequency of the respective cell (CaSpF) resulted from the number of sparks normalized to cell width and scan rate (100 μm/s). Cardiomyocytes exhibiting major arrhythmic events ($Ca^{2+}$ waves, spontaneous $Ca^{2+}$ transients, and spark clouds) were excluded from the quantification of the $Ca^{2+}$ spark. Cardiomyocytes showing $Ca^{2+}$ waves were analyzed separately. The percentage of cardiomyocytes exhibiting $Ca^{2+}$ waves was calculated and compared between the groups. Some cardiomyocytes showed more than one event therefore we calculated the events per time and represented this ratio as waves per minute. The time to the first arrhythmic event was measured in seconds.

**$Ca^{2+}$ transient measurements.** Cardiomyocytes were isolated from CaMKIIδc$^{+/T}$ mouse ventricles, plated as described above, and incubated with a Fura-2AM loading buffer (10 μmol/L, Molecular Probes, Eugene, Oregon, USA) for 20 min. After staining, the cardiomyocytes were washed with experimental solution (as described above in the $Ca^{2+}$ sparks measurement section, at room temperature) for 15 min before measurements were started to enable complete de-esterification of intracellular Fura-2AM and to allow cellular rebalance of $Ca^{2+}$ cycling properties. During measurements, cardiomyocytes were continuously superfused with an experimental solution. Measurements were performed with a Motic AE31 microscope provided with a fluorescence detection system (ION OPTIX Corp.) at room temperature. Cells were excited at 340 and 380 nm and the emitted fluorescence was collected at 510 nm. The intracellular $Ca^{2+}$ level was measured as the ratio of fluorescence at 340 and 380 nm (F340/F380 nm in ratio units, r.u.). $Ca^{2+}$ transients were recorded at steady-state conditions under constant field stimulation (1.0, 2.0, and 4.0 Hz, 20 V). The recorded $Ca^{2+}$ transients were analyzed with the software IONWizard$^R$ (ION OPTIX Corp.).

**Analysis of $Ca^{2+}$-transients from confocal line scans.** From confocal line scans that were used for quantification of diastolic $Ca^{2+}$ waves in cardiomyocytes from CaMKIIδc$^{+/T}$ and SCN10A$^{-/-}$/CaMKIIδc$^{+/T}$ mice $Ca^{2+}$-kinetics were analyzed from the $Ca^{2+}$-transients at the beginning of each line scan before 3 Hz steady-state stimulation was paused. $Ca^{2+}$-transient amplitude (F, normalized to diastole, F0) and time to half-maximal decay (RT50), and time to 90% of maximal decay were calculated from confocal line scan images using ImageJ software.

**Quantitative real-time PCR (RT-qPCR).** Human or mouse cardiac tissues, isolated human cardiomyocytes or iPSC-cardiomyocytes were snap-frozen in liquid nitrogen and stored at −80 °C. RNA was isolated by use of the ReliaPrep$^{TM}$ RNA Tissue Miniprep System (Promega). About 200 ng RNA was reverse transcribed into cDNA using standard protocols (QuantiNova Reverse Transcription Kit (QIAGEN) for mouse tissue and iScript cDNA Synthesis Kit (Bio-Rad) for human samples). For qPCR, 10 μL SYBR Green PCR Master Mix (Bio-Rad), 7 μL nuclease-free water, 1 μL forward and 1 μl reverse Primer, and 1 μL of cDNA were mixed. Q-PCR was carried out using the CFX Connect$^{TM}$ Real-Time System (Bio-Rad). Forty cycles of 15 s at 95 °C followed by 1 min of 60 °C were used and fluorescence was measured after each cycle. After 40 cycles melt curve analysis was performed to ensure the specificity of the products. Thresholds cycles were evaluated and normalized to housekeeping genes and controls. A list of all primers used is presented in Supplementary Table 4.

**Western blots.** Ventricular tissue from WT, SCN10A$^{-/-}$, CaMKIIδc$^{+/T}$, and SCN10A$^{-/-}$/CaMKIIδc$^{+/T}$ mice or pellets of iPSC-cardiomyocytes from WT and homozygous $Na_V1.8$ knockout (KO) lines were homogenized in Tris buffer

containing (in mmol/L): 20 Tris-HCl, 200 NaCl, 20 NaF, 1 $Na_3VO_4$, 1 DTT, 1% Triton X-100 (pH 7.4), and complete protease and phosphatase inhibitor cocktails (Roche Diagnostics). Protein concentration was determined by BCA assay (Pierce Biotechnology). Denatured tissue homogenates (10 min, 70 °C in 2% beta-mercaptoethanol) were separated on 8% SDS-polyacrylamide gels, then transferred to a nitrocellulose membrane, and incubated with the following primary antibodies: rabbit polyclonal anti-CaMKIIδ (1:5000, Thermo Scientific, PA5-22168), rabbit polyclonal anti-$Na_V1.5$ (1:2000, Alomone labs, ASC-005), and mouse monoclonal anti-GAPDH (1:20000, BIOTREND, BTMC-A473-9) at 4 °C overnight. Secondary antibodies included HRP-conjugated goat anti-rabbit and goat anti-mouse (1:20000, Jackson Immunoresearch, 111-035-144 and 115-035-062, respectively). The membrane was incubated with secondary antibodies for 1 h at RT. ImmobilonTM Western Chemiluminescent HRP Substrate (Millipore) was used for chemiluminescent detection. Analysis was performed using Image Studio Lite Ver 5.2. The full scan blots for $Na_V1.5$ and CaMKIIδ are shown in the Data Availability file.

**Histology.** Freshly explanted hearts were fixed in 4% buffered formaldehyde at 4 °C for 24 h. LV tissue was harvested, fixed in 4% buffered formaldehyde overnight, paraffin-embedded, sectioned (5 μm), and stained with fluorescein-conjugated wheat germ agglutinin (WGA-Alexa Fluor 594; Invitrogen, USA) for cross-sectional area (CSA) assessment. At least 300 randomly selected cardiomyocytes per animal from different sections and different regions (basal, mid-ventricular, apical) of each heart were measured using the ImageJ software (Bethesda, USA).

**Mouse echocardiography.** For echocardiography, mice were anesthetized using 1.5% isoflurane, and echocardiography was performed using a VS-VEVO 660/230 (VisualSonics, Canada). During this procedure, the core temperature was maintained at 37 °C, and heart rates were kept consistent between experimental groups at 400–500 b.p.m. Electrocardiogram monitoring was obtained using hind limb electrodes. LV geometry and systolic function were assessed by using standard 2D parasternal long and short-axis views. The examiner was blinded towards group assignment.

**Telemetric ECG recordings in mice.** Mice were implanted with an intraperitoneal telemetric ECG transmitter (TA11ETA-F10, Data Sciences International) with its Cable in lead II configuration. After a postoperative recovery period of 10 days, we recorded ECGs during periods of normal activity (24 h continuous recording, twice a week for 2 weeks). ECG parameters were analyzed, and the QT interval was corrected (QTc) using Bazett's formula, QTc = QT/(RR/100)1/2 established for mice with QT and RR expressed in ms. Premature ventricular contractions and ventricular tachycardia (>3 beats) were counted per 24 h and a medium value was per animal was calculated from all measurements. Further, physical activity was recorded by the telemetric monitors.

**Statistics.** All data were expressed as the mean ± standard error of the mean (SEM). Where appropriate, one-way ANOVA with multiple comparison tests (post hoc Bonferroni's and two-stage step-up methods of Benjamini, Krieger, and Yekutieli correction) was used. Mixed-effects analysis with Holm–Sidak's post hoc test was used to analyze I-V curves. Otherwise, Student's unpaired $t$-tests were used. The Chi-Square test was used to compare the occurrence of diastolic $Ca^{2+}$ waves. Log-rank (Mantel–Cox) test was used to compare survival between CaMKIIδc$^{+/T}$ and SCN10A$^{-/-}$/CaMKIIδc$^{+/T}$ mice. Two-sided $p ≤ 0.05$ was considered significant.

**Reporting Summary.** Further information on research design is available in the Nature Research Reporting Summary linked to this article.

## Data availability

All data generated or analyzed during this study are available within the Article and its Supplementary Information. All raw data supporting the findings from this study are available from the corresponding author upon reasonable request. Source data are provided with this paper. Any remaining raw data will be available from the corresponding author upon reasonable request.

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

## Acknowledgements

We thank Prof. John Wood (University College London) for providing us a pair of SCN10A$^{+/-}$ mice. We thank Timo Schulte, Yvonne Metz, Johanna Heine, Sabrina Koszewa, Manar El Kenani, and Sarah Zafar for technical assistance. This work was supported by the German Heart Foundation/German Foundation of Heart Research (to P.B., P.T., and S.S.); by the University Hospital Regensburg (ReForM C program) (to L.S.M. and S.S.); by the Marga und Walter Boll-Stiftung (to S.A. and S.S.); the College of Translational Medicine by the Ministry of Culture and Science, State of Lower Saxony (P.B.), the German Center for Cardiovascular Research (DZHK; to K.S.-B.), the Else Kröner-Fresenius Stiftung (N.H. and M.T.) and the Deutsche Forschungsgemeinschaft

(DFG) through the International Research Training Group Award (IRTG) 1816 (to K.S.-B.; W.M. is a fellow under IRTG 1816) and SFB 1002 (to K.T., N.D., and G.H.).

## Author contributions

P.B., S.A., and S.S. conceived the study. K.S.-B., P.B., and S.S. designed the experiments and wrote the manuscript. P.B., N.D., P.T., S.A., B.M., N.H., M.C.K., W.M., S.P., and M.T. carried out experimental work and analyzed the data. K.S.-B. designed the CRISPR experiments. J.G., H.M., and S.L.-H. acquired and provided human tissue. K.T. applied for the authorization to carry out animal experiments and helped to analyze in vivo data. J.M., S.W., L.S.M., and G.H. provided expertise and feedback. All authors discussed the results and had the opportunity to comment on the manuscript.

## Funding

## Competing interests

The authors declare no competing interests.
