## [Peer Review File · Nature Communications]

Reviewers' comments:

Reviewer #1 (Remarks to the Author):

This paper by Bengel et al. is a follow-up of a previous paper from the group (Dybkova et al, Cardiovascular Research 2018), where they described increased expression of the sodium channel Nav1.8 (encoded by the SCN10A gene) and decreased Nav1.5 (SCN5A) expression in human heart failure (HF). In that paper they also showed that pre-incubation of human HF cardiomyocytes with Nav1.8 blockers decreased late sodium current (late INa) and reduced action potential duration (APD), and prevented ISO-induced increase in late INa and APD prolongation in WT but not Nav1.8 KO mice. From that paper, the authors concluded that enhancement of Nav1.8-based late INa plays an important role in repolarization alterations and pro-arrhythmia in the setting of HF.

In the current manuscript, the authors aim to assess the localization of Nav1.8 in cardiomyocytes and its potential interaction with CamKII δ , and to explore the contribution of Nav1.8 to the detrimental consequences of enhanced CamKII δ activity; the latter is a well-known feature occurring in HF and is known to contribute to arrhythmias. The authors argue that inhibition of CamKII δ as a therapeutic strategy is challenging, and hence they propose that blocking downstream effectors of enhanced CamKII δ activity (for instance, Nav1.8) may be more feasible.

To study this hypothesis, the authors performed immunohistochemistry and co-IP in human ventricular tissue and observe co-localization and interaction between Nav1.8 and CamKII δ .

They then show that pharmacological Nav1.8 inhibition decreases late INa and calcium sparks in mice overexpressing CamKII δ . Moreover, the CamKII δ inhibitor AIP reduced late INa and calcium sparks in human failing cardiomyocytes, while combining AIP and Nav1.8 blockade gave similar reduction in late INa and sparks. Surprisingly, the observed effects on calcium sparks were not accompanied by changes in calcium homeostasis or SR content.

From these observations, in particular the fact that inhibition of both CamKII δ and Nav1.8 together resulted in similar reduction in late INa and sparks as compared to Nav1.8 inhibition alone, the authors deduce that Nav1.8 activation is CamKII δ -dependent.

They finally investigated if genetic Nav1.8 inhibition may prevent cardiac hypertrophy and improve survival in CamKII δ overexpressing mice by crossbreeding the latter with Scn10a knockout mice. Double mutant mice (i.e. with increased CamKII δ activity in the absence of Scn10a) showed similar development of hypertrophy and similar decrease in ejection fraction compared to CamKII δ TG mice with normal Scn10a expression, yet showed some improvement in survival. In these double mutant mice, the authors furthermore observed reduced late INa, as well as decreased afterdepolarizations and calcium waves. From these findings, the authors conclude that the reduced pro-arrhythmia may underlie the observed improvement in survival.

Comments on the conclusions and overall message of the paper:

Overall, the results are not very surprising and do not provide much novel mechanistic insight:

- The conclusion that inhibition of Nav1.8-based late INa is beneficial is not novel, and has previously been shown by the authors and others.

- The finding that increased CamKII δ activity in HF contributes to enhanced late sodium current via interaction with Nav1.8 is in a way expected given the known effects of CamKII δ on sodium channels. Also, HF is associated with general cellular dysregulation including alterations in calcium homeostasis which in turn impacts on late INa (as well as being caused by it). So, any condition associated with hypertrophy or HF will likely result in increased Nav1.8, which may be modulated by additional processes other than CamKII δ . No data is provided on late INa in control mice, hence it is unclear if late INa in CamKII δ TG mice is in fact increased (I assume so), and whether inhibition of Nav1.8 restores late INa values to WT levels. Conversely, what happens if HF is induced in Scn10a knockout mice? Are they protected from e.g. hypertrophy, calcium sparks, arrhythmias?

- The observations on the distribution of Nav1.8 and CamKII δ in cardiomyocytes are not very relevant for the paper since no functional studies are associated with it, and the relevance of this distribution is not addressed.

- The authors conclude that Nav1.8-induced late INa is CamKII δ -dependent. However, this conclusion is not supported by the data. CamKII δ overexpressing (TG) mice have an overt HF phenotype which is sure to modulate many different pathways and processes, including abnormal calcium homeostasis etc. The fact that blocking Nav1.8 reduces late INa and afterdepolarizations in these mice is not surprising since it is known that these channels are upregulated during HF (which occurs in CamKII δ TG mice). Deletion of Scn10a in CamKII δ TG mice showed some improvement in survival, but was not sufficient to prevent HF in mice with increased CamKII δ activity, so clearly the detrimental effects of CamKII δ act also via pathways other than Nav1.8. If the authors want to make the point that Nav1.8-induced late INa is CamKII δ dependent, they should take the opposite approach and show that in mice lacking CamKII δ (or with its activity blocked), induction of HF through e.g. TAC does not increase Nav1.8.

- Inhibition of Nav1.8 in CamKII δ TG mice was not sufficient to prevent hypertrophy and HF. Although some improvements were observed in survival, it is unclear whether this was due to decreased arrhythmia incidence or not. Hence, the proposition that Nav1.8 inhibition would be a good substitute for inhibition of the detrimental effects of CamKII δ activation, is not valid. Likely, inhibition of Nav1.8 is not sufficient due to the fact that other pathological processes are still occurring.

Comments on the methodology and data interpretation:

In addition to the general comments listed above, there are a number of concerns regarding methodology and interpretation of the data:

- Late sodium current measurements: these are performed at room temperature, which seriously limits the potential relevance of the data since it has been shown that this enhanced late current may only be present at room temperature but not at 37 degrees. Late current should be measured as TTX-sensitive current because otherwise it is impossible to say if it is truly a sodium current. Also, the integral measurements are potentially problematic. In the examples shown, the current between 100-500 ms after the peak is not stable but continues to decrease which indicates a potential drift/leak. A true late current caused by channels that do not inactivate properly typically remains constant during this time frame. Hence, these measurements should be repeated at 37 degrees, measuring late I_{Na} as TTX-sensitive current measured at a single time point (i.e. at 500 ms), not as an integral since the latter may overestimate the late current. Ideally, these should be performed at various voltage steps to validate that it is a sodium current. Also, how do the authors know that they are not looking at a window current, since they only look at one voltage step? Peak current measurements with kinetics should be performed, at least in the mouse experiments. This is also essential since it is important to know if the observed alterations in late I_{Na} are not simply due to an overall reduction in peak sodium current.

- The fact that cells were pre-incubated with the drugs (and not through acute wash-in) is also problematic since it precludes measuring the effect of blockers in a pair-wise fashion. This way, it is impossible to say whether cells actually had a (large) late I_{Na} to begin with or not. It is essential to show that in those cells that have a large late I_{Na}, the blockers actually reduce late I_{Na} in that cell. This is particularly relevant since it appears that not all human hearts were studied in each treatment group. Were cells from the same heart investigated in each treatment group, and was this evenly distributed among groups? How can the authors be sure that all human hearts and cells had similar late sodium current magnitude to begin with? Were all heart failure patients clinically affected in a similar fashion? Please include detailed information on clinical status, age, gender, medication, etc. The current methodology is very risky and carries the inherent risk of selection bias.

The authors should show that acute wash-in of the drugs (and not pre-incubation) gives similar results (ideally, washout should also be attempted). This is the only way to unequivocally show that the inhibitors really block late INa, and to what extent. This methodological concern also applies to the calcium and AP measurements.

- For all mouse data, results for wild type littermates should also be shown. This is essential to assess i) the impact of the CamKII δ overactivity, and ii) whether the interventions actually restore the various parameters back to the WT situation.

- If no alterations are observed in calcium transients and SR content (Figure 5), then how do the authors explain the observed differences in calcium sparks (Figure 4)?

- CamKII δ TG mice are crossed with Scn10a KO mice. However, no information is provided on the genetic backgrounds (strains) of the 2 mouse lines. If the mice are actually of 2 separate strains, then the observed differences in survival may also be caused by this. What was the mode of death? Did the mice die of heart failure or sudden? Authors should make an effort to at least do ECG measurements and check for possible arrhythmias; ideally, with telemetry. The authors propose that decreased incidence of arrhythmias underlies the improved survival, but they do not provide any evidence for this. In vivo or ex vivo (arrhythmia inducibility in explanted hearts) evidence for this should be provided.

- Figure 7: Again, the methodology for late INa measurement is not appropriate. The typical example shows only a very small difference in late INa at the end of the 500 ms trace, and certainly not a doubling as indicated in the average values in panel B. These should be repeated as indicated above, and values for Scn10 KO and WT mice should also be provided, in addition to peak sodium current values. In panels C/D, the protocol for EAD/DAD measurements is suboptimal, since it is impossible to say whether the observed extra APs are spontaneous or triggered. Instead, a fast pacing protocol should be used to assess triggered activity, EADs and DADs. Additional hearts should be studied since n=3 is a very small number and it leaves open the possibility that the data is driven by for instance one suboptimal isolation leading to low quality cells which are depolarized and show increased spontaneous activity. Hence, all AP parameters should be provided, including resting membrane potential, amplitude and duration; depolarized cells should be excluded from the analysis.

- Details are missing from the methods: gender of the mice; temperature at which the AP measurements were performed, temperature at which the calcium transients and sparks were performed

Reviewer #2 (Remarks to the Author):

This is an interesting paper that presents new and potentially important evidence supporting a proarrhythmic interaction between CaMKII and Nav1.8. The strengths of the study are its novelty, clinical/translational importance, and the use of a Nav1.8 (SCN10A^{-/-}) knock out model. However, there are important weaknesses that should be addressed. These include the need to determine if the now well known interaction between CaMKII and Nav1.5 is conserved for CaMKII and Nav1.8, establish specificity of the Nav1.8 antagonist drugs, using the Nav1.8 knock out cardiomyocytes as controls, measuring in vivo arrhythmias, which are the presumed basis for partial longevity rescue in the CaMKII^d TG x SCN10A^{-/-} interbred mice, and increasing the number of unique human samples.

Specific comments:

Are the domains of Nav1.5 known to be important for CaMKII binding and enhanced late current conserved in Nav1.8? Does CaMKII bind Nav1.5 and Nav1.8 by similar sites/mechanisms?

The human studies shown in Fig 1, with a very small number of unique samples, need some form of quantification. Why only n=4? Even for hard to obtain human specimens, this number is low - too low to make the point that it is an important attribute of failing myocardium in patients.

Was the specificity of these antagonists proven in the Nav1.8 ko myocardial cells? Does it lack all activity against Nav1.5? Does Nav1.8 ko change expression of Nav1.5?

Does AIP have an amphipathic tag for membrane permeation or was it dialyzed in the pipette solution? If the former (for voltage clamp or Ca²⁺ spark studies), there should be a control peptide with a similar tag because membrane currents are typically affected by myristoylation, palmitoylation, TAT or other molecular adducts designed to enhance cell membrane permeation.

Death by arrhythmia is increased in CaMKII^d TG mice, so it is plausible that SCN10A^{-/-} interbreeding reduces mortality by decreasing arrhythmia. However, this should be tested directly in vivo.

Similarly, the Nav1.8 antagonists, after proof of specificity in SCN10A^{-/-} cardiomyocytes, should be used to protect against arrhythmias in vivo in CaMKII^d TG mice, at least in an acute study.

Reviewer #3 (Remarks to the Author):

The authors present data supporting an intriguing conclusion that a majority of I-NaL is due to Nav1.8 and its interaction with CaMKII. This finding has important mechanistic implications and as

well implications for development of future anti-arrhythmic therapy. Given the novelty and potential importance of the findings, shifting focus of cardiac I-NaL from Nav1.5 to Nav1.8, the rigor of the studies must be improved. Contribution of non-myocyte cardiac cells to tissue Nav1.8 needs to be ruled out. Additionally, specificity of the drug studies needs to be established as does efficiency and specificity of the knockout models to rule out secondary effects on Nav1.5. Most importantly, the authors need to clarify the full survival curve and arrhythmogenic phenotype of the animal models rather than draw conclusions from an apparently abbreviated survival curve and subsequent cellular studies. Specific requests are provided below.

1. Please quantify amount of Nav1.8 in non-failing and failing human ventricular muscle, relative to Nav1.5. Provide quantification in both intact ventricle and, to consider origin from non-muscle cardiac cells, quantify channel content in human or mouse isolated ventricular cardiomyocytes as well.

2. It is not clear that the pharmaceutical agents A-806734 and PF-01247324 do not block Nav1.5 as well, and thus the effects of blockade of Nav1.5 are attributed in the manuscript to blockade of Nav1.8. Please repeat the experiments Figure 3 and 4 in the setting of genetic or pharmacologic block of Nav1.5.

3. Please provide RNA and western blot evidence of cardiomyocyte knockout of Nav1.8 and CamKII in each mouse model used. Please also confirm that Nav1.5 is not affected in adult mice of each mouse model.

4. The survival advantage of knocking out Nav1.8 which starts at 60 days appears to dissipate at 100 days (Figure 6A). The low ejection fraction of the mutant mice suggests all the mice will die at a relatively young age, affecting the conclusions regarding survival. Please complete the survival curve until either all animals die or the curves plateau to more accurately portray steady state survival.

5. The low ejection fraction of the mutant mice suggests the mice die from progressive heart failure rather than ventricular arrhythmogenesis. Please provide telemetry evidence of in vivo arrhythmogenesis and sudden cardiac death of the mice studied. Given the concern of progressive heart failure, please also provide body weight and activity levels versus age of the mice.

6. For ratiometric measurement of diastolic calcium, please provide the methodological detail regarding obtaining minimal and maximal fluorescent ratios and convert all measured calcium concentration ratios to units of nM.

7. Please provide clinical details of the Stage IV explanted human hearts- at a minimum age and sex of patients, cause of cardiomyopathy, ejection fraction, and LV dimensions. Also, please provide cross-clamp time of the control human hearts and as well as age, sex, and cause of death.

Reviewer #1 (Remarks to the Author):

This paper by Bengel et al. is a follow-up of a previous paper from the group (Dybkova et al, Cardiovascular Research 2018), where they described increased expression of the sodium channel Nav1.8 (encoded by the SCN10A gene) and decreased Nav1.5 (SCN5A) expression in human heart failure (HF). In that paper they also showed that pre-incubation of human HF cardiomyocytes with Nav1.8 blockers decreased late sodium current (late INa) and reduced action potential duration (APD), and prevented ISO-induced increase in late INa and APD prolongation in WT but not Nav1.8 KO mice. From that paper, the authors concluded that enhancement of Nav1.8-based late INa plays an important role in repolarization alterations and pro-arrhythmia in the setting of HF.

In the current manuscript, the authors aim to assess the localization of Nav1.8 in cardiomyocytes and its potential interaction with CamKII δ , and to explore the contribution of Nav1.8 to the detrimental consequences of enhanced CamKII δ activity; the latter is a well-known feature occurring in HF and is known to contribute to arrhythmias. The authors argue that inhibition of CamKII δ as a therapeutic strategy is challenging, and hence they propose that blocking downstream effectors of enhanced CamKII δ activity (for instance, Nav1.8) may be more feasible.

To study this hypothesis, the authors performed immunohistochemistry and co-IP in human ventricular tissue and observe co-localization and interaction between Nav1.8 and CamKII δ . They then show that pharmacological Nav1.8 inhibition decreases late INa and calcium sparks in mice overexpressing CamKII δ . Moreover, the CamKII δ inhibitor AIP reduced late INa and calcium sparks in human failing cardiomyocytes, while combining AIP and Nav1.8 blockade gave similar reduction in late INa and sparks. Surprisingly, the observed effects on calcium sparks were not accompanied by changes in calcium homeostasis or SR content. From these observations, in particular the fact that inhibition of both CamKII δ and Nav1.8 together resulted in similar reduction in late INa and sparks as compared to Nav1.8 inhibition alone, the authors deduce that Nav1.8 activation is CamKII δ -dependent. They finally investigated if genetic Nav1.8 inhibition may prevent cardiac hypertrophy and improve survival in CamKII δ overexpressing mice by crossbreeding the latter with Scn10a knockout mice. Double mutant mice (i.e. with increased CamKII δ activity in the absence of Scn10a) showed similar development of hypertrophy and similar decrease in ejection fraction compared to CamKII δ TG mice with normal Scn10a expression, yet showed some improvement in survival. In these double mutant mice, the authors furthermore observed reduced late INa, as well as decreased afterdepolarizations and calcium waves. From these findings, the authors conclude that the reduced pro-arrhythmia may underlie the observed improvement in survival.

Comments on the conclusions and overall message of the paper:

Overall, the results are not very surprising and do not provide much novel mechanistic insight:

- The conclusion that inhibition of Nav1.8-based late INa is beneficial is not novel, and has previously been shown by the authors and others.

Answer: We thank you for the detailed analysis of our work. We took your concerns very serious and have tried to compensate for any limitation by performing a plethora of new in-vivo and in vitro-experiments in order to improve the novelty, mechanistic strength, but also translational significance.

We believe that our manuscript contains translational, novel, and mechanistic insights because of

1. First time of demonstration of the detrimental interaction of CaMKII and Nav1.8 (additional data)
2. Proving the existence and functional relevance of Nav1.8 in cardiomyocytes (still under debate) in a new CRISPR-Cas9-generated iPS-cardiomyocyte knock-out of Nav1.8 (new)
3. We show that knock-out of Nav1.8 in CaMKII transgenic mice with heart failure improves survival. This result is of strong translational significance (survival curve now improved and extended)
4. This can be attributed to improved cellular electrophysiology rather the progress of heart failure. In fact, these mice have reduced ventricular arrhythmias demonstrated on the cellular level (new data) but now also using implanted telemetric monitors in-vivo (new)

We hope to convince the reviewer that these findings are novel and of significant translational importance. In addition, for our knowledge of late Na^+ -current and SR Ca^{2+} -leak per se and their potential targeting we provide new insights on their role for mortality and arrhythmia formation in-vivo and in-vitro.

- The finding that increased CamKII δ activity in HF contributes to enhanced late sodium current via interaction with Nav1.8 is in a way expected given the known effects of CamKII δ on sodium channels. Also, HF is associated with general cellular dysregulation including alterations in calcium homeostasis which in turn impacts on late I_{Na} (as well as being caused by it). So, any condition associated with hypertrophy or HF will likely result in increased Nav1.8, which may be modulated by additional processes other than CamKII δ . No data is provided on late I_{Na} in control mice, hence it is unclear if late I_{Na} in CamKII δ TG mice is in fact increased (I assume so), and whether inhibition of Nav1.8 restores late I_{Na} values to WT levels. Conversely, what happens if HF is induced in Scn10a knockout mice? Are they protected from e.g. hypertrophy, calcium sparks, arrhythmias?

Answer: We thank the reviewer for her/his comment. Although the reviewer feels that some finding can be assumed, scientific efforts are needed to prove it. Most importantly, from a translational point of view targeting downstream targets of CaMKII is of particular interest as a potential antiarrhythmic strategy. Inhibition of Nav1.5 is problematic in structural heart disease and increases mortality (e.g. CAST-Trial). Therefore, targeting Nav1.8 is an attractive strategy to avoid many serious problems associated with Nav1.5 modulation. Thus, we believe that the “CaMKII/Nav1.5-story” cannot directly be compared to our findings from a technical-scientific but also translational and clinical point of view.

Accordingly, based on our hypothesis and your assumption that increased CaMKII expression and activity in HF may contribute to enhanced I_{NaL} via interaction with Nav1.8, we tried to clarify this. As suggested by you we measured I_{NaL} in Wild-Type littermates of our CaMKII transgenic mice and found an increase in I_{NaL} by +150% in cardiomyocytes from CaMKII transgenic mice compared to Wild-Type. Application of one of the Nav1.8 inhibitors (A-803467 or PF-01247324) showed a decrease in late I_{Na} by -40% compared to untreated cardiomyocytes from CaMKII transgenic mice. These new measurements are now presented in the manuscript in the figure 2A & 2B and show that inhibition of Nav1.8 leads to a significant decrease in I_{NaL} but does not restore I_{NaL} values to WT levels. This can be explained by the fact that CaMKII still phosphorylates other Nav channels like Nav1.5 leading to increased I_{NaL} .

Fig. 2: **A:** Original traces of I_{NaL} in WT and $CaMKII\delta c^{+/T}$ mouse ventricular cardiomyocytes elicited using the protocol shown in inset. **B:** Mean data \pm SEM along with individual values shown in the graph plotting WT ($n=10$ cells/5 mice) and $CaMKII\delta c^{+/T}$ (untreated: $n=15$ cells/7 mice; A-806467: $n=12$ cells/7 mice; PF-01247324: $n=12$ cells/7 mice). Probability vs. untreated (One-way ANOVA).

The reviewer asks what happens if heart failure is induced in SCN10A knock-out mice. In our study we used our transgenic mouse model with CaMKII overexpression to achieve this. These animals develop a progressing phenotype of severe heart failure. We therefore crossbred CaMKII overexpressing mice with SCN10A knock-out mice. As our results indicate, SCN10A knock-out had no effect on hypertrophy and heart failure progression in these mice. To further investigate the effect of SCN10A knock-out on structural remodeling we performed additional experiments which were added to the manuscript (Fig. 4). Besides the unchanged heart-weight to tibia length ratio and the findings of the echocardiography we found that, SCN10A knock-out had no significant effect on hypertrophy (measured as cross-sectional area of the cardiomyocyte) in CaMKII overexpressing mice. Further, expression of important Na^+ and Ca^{2+} regulating proteins like Nav1.5 and CaMKII itself was not changed between mice with or without Nav1.8 (Supplementary Fig. S6 & S7). In contrast, in the revised version of the manuscript we provide new *in vivo* and *in vitro* data demonstrating a potential protection of SCN10A knock-out mice from arrhythmias (Fig. 5 & 6). Most importantly, we implanted telemetric monitors and could show a potent reduction of ventricular arrhythmias *in vivo* as investigated in a blinded manner (Fig. 6).

Fig. 4: **A:** Survival curve of $CaMKII\delta^{c+/T}$ and $SCN10A^{-/-}/CaMKII\delta^{c+/T}$ (43 vs 63 animals). Log-rank (Mantel-cox test and Gehan-Breslow-Wilcoxon test) were performed to calculate the survival percentage of mice. Probability vs $CaMKII\delta^{c+/T}$. **B:** Original hearts from WT, $SCN10A^{-/-}$, $CaMKII\delta^{c+/T}$ and $SCN10A^{-/-}/CaMKII\delta^{c+/T}$ mice. **C:** Ratio of heart weight to tibia length as a parameter of cardiac hypertrophy. $CaMKII\delta^{c+/T}$ and $SCN10A^{-/-}/CaMKII\delta^{c+/T}$ showed a significant increase in this ratio compared to WT and $SCN10A^{-/-}$ mice. Data was analysed by One-way ANOVA with post-hoc Bonferroni's correction. (N=hearts studied, WT=10, $SCN10A^{-/-}$ =15, $CaMKII\delta^{c+/T}$ =13 and $SCN10A^{-/-}/CaMKII\delta^{c+/T}$ =23). **D:** Original histological WGA staining from WT, $SCN10A^{-/-}$, $CaMKII\delta^{c+/T}$ and $SCN10A^{-/-}/CaMKII\delta^{c+/T}$ mice. Scale bars=75 μ m **E:** Cardiomyocyte cross-sectional-area (CSA) as parameter for cellular hypertrophy. $CaMKII\delta^{c+/T}$ and $SCN10A^{-/-}/CaMKII\delta^{c+/T}$ showed a significant increase in CSA compared to WT and $SCN10A^{-/-}$ mice. CSA in $CaMKII\delta^{c+/T}$ and $SCN10A^{-/-}/CaMKII\delta^{c+/T}$ mice did not significantly differ. Data was analysed by One-way ANOVA with post-hoc Newman-Keuls's correction. N=hearts studied (>300 cardiomyocytes were studied per heart), WT=5 hearts, $SCN10A^{-/-}$ =5 hearts, $CaMKII\delta^{c+/T}$ =4 hearts, $SCN10A^{-/-}/CaMKII\delta^{c+/T}$ =5 hearts **F:** Original echo recordings from WT, $SCN10A^{-/-}$, $CaMKII\delta^{c+/T}$ and $SCN10A^{-/-}/CaMKII\delta^{c+/T}$ at M-mode in 12-13-week-old mice **G:** Echo recordings revealed a decrease in left ventricular ejection fraction (EF%) in $CaMKII\delta^{c+/T}$ (6 mice) and $SCN10A^{-/-}/CaMKII\delta^{c+/T}$ (6 mice) compared to WT (7 mice) or $SCN10A^{-/-}$ (8 mice) ($p<0.0001$). EF was not significantly different in WT vs $SCN10A^{-/-}$ or $CaMKII\delta^{c+/T}$ vs $SCN10A^{-/-}/CaMKII\delta^{c+/T}$. **H:** Echo recordings revealed a significant increase in left ventricular end diastolic diameter (LVEDD) in $CaMKII\delta^{c+/T}$ (6 mice) and $SCN10A^{-/-}/CaMKII\delta^{c+/T}$ (6 mice) compared to WT (7 mice) or $SCN10A^{-/-}$ (8 mice) ($p<0.0001$). LVEDD was not significantly different in WT vs $SCN10A^{-/-}$ or $CaMKII\delta^{c+/T}$ vs $SCN10A^{-/-}/CaMKII\delta^{c+/T}$.

Without effects on structural remodeling, $SCN10A$ knock-out improved survival in $CaMKII$ overexpressing mice, as well as the occurrence of cellular proarrhythmic triggers like Ca^{2+} -waves and EADs as well as DADs were reduced. Therefore, we conclude, that the survival benefit in mice lacking Nav1.8 is likely due to beneficial effects on arrhythmias rather than the progression of the disease itself.

We also try to stick to the principles of the "3Rs" of animal experiments and limit in vivo investigations to absolutely necessary experiments (e.g. one heart failure model). As we seek to investigate the interaction with $CaMKII$, the best model was the $CaMKII$ overexpressing mice accordingly, regarding additional in-vivo investigations we focused on implantation of telemetries as suggested by the reviewer because this is indeed essential for the most important mechanistic point of view. The new telemetry data show a reduction of in vivo arrhythmias by $SCN10A$ knock-out in $CaMKII$ transgenic mice (see below).

- The observations on the distribution of Nav1.8 and CamKII δ in cardiomyocytes are not very relevant for the paper since no functional studies are associated with it, and the relevance of this distribution is not addressed.

Answer:

We agree with the reviewer, and therefore decided to exclude Figure 2 from the original manuscript. Our observations on Nav1.8 distribution within the cardiomyocyte is now limited to the one staining in Figure 1 revealing the co-localization of Nav1.8 and CaMKII in human failing cardiomyocytes. These stainings support the interaction studies between Nav1.8 and CaMKII shown in Figure 1A, since they are localized at the same microdomains in cardiomyocytes.

- The authors conclude that Nav1.8-induced late I_{Na} is CamKII δ -dependent. However, this conclusion is not supported by the data. CamKII δ overexpressing (TG) mice have an overt HF phenotype which is sure to modulate many different pathways and processes, including abnormal calcium homeostasis etc. The fact that blocking Nav1.8 reduces late I_{Na} and afterdepolarizations in these mice is not surprising since it is known that these channels are upregulated during HF (which occurs in CamKII δ TG mice). Deletion of Scn10a in CamKII δ TG mice showed some improvement in survival but was not sufficient to prevent HF in mice with increased CamKII δ activity, so clearly the detrimental effects of CamKII δ act also via pathways other than Nav1.8. If the authors want to make the point that Nav1.8-induced late I_{Na} is CamKII δ dependent, they should take the opposite approach and show that in mice lacking CamKII δ (or with its activity blocked), induction of HF through e.g. TAC does not increase Nav1.8.

Answer: We fully agree to the reviewer that CaMKII has detrimental effects on progression of heart failure and cardiomyocyte function by several different pathways and have to apologize if this was not intensively pointed out. The reviewer suggests proving CaMKII-dependent induction of Nav1.8-dependent I_{NaL} by showing this in a model where CaMKII is knocked down or its activity is blocked. In fact, we aimed for this approach in our experiments in human failing cardiomyocytes, where we inhibited CaMKII with AIP and Nav1.8 with PF-01247324 at the same time. We found a profound reduction in I_{NaL} while blocking Nav1.8 alone, as well as after inhibition of CaMKII. Therefore, we can conclude that a relevant part of I_{NaL} is Nav1.8 dependent. The fact, that CaMKII-dependent phosphorylation of sodium channels increases I_{NaL} was previously demonstrated by many groups, including ours. Accordingly, a CaMKII-dependent phosphorylation of the predominating cardiac sodium channel isoform Nav1.5 was shown. Therefore, inhibition of CaMKII results in a reduction of I_{NaL} in remodeled cardiomyocytes. In our experiments we did not observe an additional reduction in I_{NaL} when inhibiting CaMKII and Nav1.8 together, concluding that Nav1.8-dependent I_{NaL} is already abolished after CaMKII inhibition. We tried to carry out this now more carefully in our revised manuscript:

"In our study we investigated the contribution of Nav1.8 to I_{NaL} in a HF model that was exclusively induced by chronic CaMKII δ c overexpression. As previously shown, I_{NaL} was augmented in cardiomyocytes from CaMKII δ c^{+T} compared to WT^{21,40}, while these effects were ameliorated by application of specific Nav1.8 blockers. Therefore, at least a relevant part of CaMKII δ c-induced I_{NaL} appears to be driven by Nav1.8. Additional support for this conclusion comes from our I_{NaL} measurements from human failing cardiomyocytes where CaMKII δ c activity and I_{NaL} are both known to be increased in parallel^{2,13,16,21,41}. Inhibition of either CaMKII δ c or Nav1.8 reduced I_{NaL} as separately demonstrated before^{21,31}. However, simultaneous inhibition of Nav1.8 and CaMKII δ c had no additive effect compared to CaMKII δ c inhibition alone. Therefore, it can be assumed that the majority of Nav1.8-driven I_{NaL} was already abolished by CaMKII δ c inhibition and hence, seems to be CaMKII δ c-dependent. In addition, SCN10A knock-out in CaMKII δ c^{+T} mice resulted in a reduction in I_{NaL} comparable to Nav1.8 inhibition upon specific blockers. In previous studies the functional relevance of Nav1.8 expression in the cardiomyocyte was questioned as application of specific blockers had no effects on peak I_{Na} as well as I_{NaL} in healthy and unstimulated cardiomyocytes^{42,43}. These data are not in conflict with our findings, as we also did not observe differences in I_{NaL} magnitude in cardiomyocytes from healthy mice, while clear effects were evident under conditions of enhanced I_{NaL} either by chronic CaMKII δ c overexpression or isoproterenol treatment of iPS-cardiomyocytes. Therefore, the interaction of Nav1.8 with enhanced CaMKII δ c activity might be necessary to generate meaningful effects on cardiomyocyte electrophysiology while Nav1.8 appears to

play a negligible role in healthy cardiomyocytes. This establishes Nav1.8 to be a disease-specific target.” (page 8, line 207-225).

The reviewer proposes also to induce heart failure in another model lacking CaMKII to show that Nav1.8 expression is increased by CaMKII-dependent mechanisms.

However, as pointed out above in detail we believe that we demonstrate on the cellular level, that Nav1.8-dependent I_{NaL} is CaMKII-dependent and not its overexpression in the failing heart. We hope that the reviewer agrees that these cellular investigations are appropriate, and another heart failure mouse model is not justified with respect to the “3Rs” of animal experiments and our novel in vivo data (telemetry). Moreover, as also critically mentioned by the reviewer knock-out as well as overexpression of CaMKII causes a plethora of regulative cellular changes including potential allosteric alterations at the RyR2. As a consequence, potential derived data from such a mouse model may also be interpreted with caution. Finally, we limit our conclusion to a more precise formulation such as Nav1.8 as a mediator of **arrhythmias** may constitute a downstream target of CaMKII.

The reviewer claimed, that detrimental effects of CaMKII act also via other pathways than Nav1.8 in heart failure. We fully agree to this statement. As we show in our experiments the deletion of SCN10A does not prevent CaMKII overexpressing mice from heart failure development as CaMKII is involved in several signaling pathways that are involved in hypertrophy and heart failure. Therefore, improved survival in our CaMKII overexpressing mice lacking SCN10A can, as far as we know, only be explained by a significant reduction of arrhythmias. Our results are therefore comparable with a study from our groups (Mohammed et al. Sci Transl Med 2018). The authors pharmacologically inhibited SR- Ca^{2+} -leak in mice with heart failure induced by TAC or myocardial infarction. Inhibition of SR- Ca^{2+} -leak resulted in reduced arrhythmias on the cellular level, as well as in vivo. However, the progression of heart failure in these animals remained unaffected by this intervention. Apparently, the reduction of arrhythmias alone was sufficient enough to improve survival of animals treated with a selective inhibitor of SR- Ca^{2+} -leak. In our study SR- Ca^{2+} -leak is reduced by an indirect approach. Likewise, we find similar observations. Taken together our data suggests a role of CaMKII in regulation of Nav1.8-dependent I_{NaL} in heart failure which was shown to contribute to cellular proarrhythmia. We fully agree, Nav1.8 seems to be not involved in significant structural CaMKII-dependent remodeling processes in heart failure.

- Inhibition of Nav1.8 in CamKIId TG mice was not sufficient to prevent hypertrophy and HF. Although some improvements were observed in survival, it is unclear whether this was due to decreased arrhythmia incidence or not. Hence, the proposition that Nav1.8 inhibition would be a good substitute for inhibition of the detrimental effects of CamKIId activation, is not valid. Likely, inhibition of Nav1.8 is not sufficient due to the fact that other pathological processes are still occurring.

Answer: We agree with reviewer that in CaMKII overexpressing mice lacking SCN10A transcriptional and structural changes still occur despite the genetic ablation of Nav1.8 and have extended the data on this.

The assumption that the improved survival (please also note new included mice in the survival curve) is due to improved cellular electrophysiology and arrhythmias is now supported by our new telemetry data that show a reduction of in vivo arrhythmias by SCN10A knock-out in CaMKII transgenic mice (Fig. 6).

Fig. 6: **A:** Original ECG-traces of $CaMKII\delta c^{+/T}$ and $SCN10A^{-/-}/CaMKII\delta c^{+/T}$ showing ventricular arrhythmias. **B:** Unchanged activity levels in $SCN10A^{-/-}/CaMKII\delta c^{+/T}$ compared to $CaMKII\delta c^{+/T}$ ($CaMKII\delta c^{+/T}$ (4 mice); $SCN10A^{-/-}/CaMKII\delta c^{+/T}$ (3 mice)). **C:** Reduction of premature ventricular contractions in $SCN10A^{-/-}/CaMKII\delta c^{+/T}$ (p=0.08, student's t-test). **D:** Reduction of ventricular tachycardia in $SCN10A^{-/-}/CaMKII\delta c^{+/T}$ (p<0.05, student's t-test).

Moreover, to satisfy the reviewer we downplayed our conclusion that Nav1.8 inhibition would be a good substitute for CaMKII inhibition in our manuscript. Of course, Nav1.8 inhibition is not sufficient to reverse all detrimental effects of CaMKII overexpression and overactivity in heart failure. Nevertheless, unfavorable actions of increased CaMKII activity on calcium handling and proarrhythmogenic triggers could be significantly ameliorated by genetic depletion of Nav1.8, as shown in our cellular experiments. These observations can be attributed by an indirect inhibition of the CaMKII-dependent activation of I_{NaL} . In our revised manuscript we now changed the statement to:

"As it has consistently been shown that activation of $CaMKII\delta c$ is involved in enhancement of I_{NaL} and vice versa^{11,12,22,44,57} our proposed crosstalk between $CaMKII\delta c$ and $Nav1.8$ causing enhanced I_{NaL} in the presence of hyperactive $CaMKII\delta c$ may be a novel strategy to prevent detrimental $CaMKII\delta c$ -induced effects on cellular electrophysiology." (page 11 line 275-278).

We hope the reviewer agrees with this more precise expression and our novel in vivo experiments. Of note, the reviewer judges the effect size of our mouse survival some times to be modest. It cannot be expected that a modification of one of thousands targets involved in arrhythmia generation leads to effects of e.g. 50% or more. We performed a very careful and most importantly blinded investigation with a potent number of animals leading to a statistical significant result. Importantly, in clinical trials the magnitude of the observed survival difference in our experiments would be remarkable in a collective with severe heart failure (in particular due to rhythm modulation).

Comments on the methodology and data interpretation:

In addition to the general comments listed above, there are a number of concerns regarding methodology and interpretation of the data:

- Late sodium current measurements: these are performed at room temperature, which seriously limits the potential relevance of the data since it has been shown that this enhanced late current may only be present at room temperature but not at 37 degrees. Late current should be measured as TTX-sensitive current because otherwise it is impossible to say if it is truly a sodium current. Also, the integral measurements are potentially problematic. In the examples shown, the current between 100-500 ms after the peak is not stable but continues to decrease which indicates a potential drift/leak. A true late current caused by channels that do not inactivate properly typically remains constant during this time frame. Hence, these measurements should be repeated at 37 degrees, measuring late I_{NaL} as TTX-sensitive current measured at a single time point (i.e. at 500 ms), not as an integral since the latter may overestimate the late current. Ideally, these should be performed at various voltage steps to validate that it is a sodium current. Also, how do the authors know that they are not looking at a window current, since they only look at one voltage step? Peak current measurements with kinetics should be performed, at least in the mouse experiments. This is also essential since it is important to know if the observed alterations in late I_{NaL} are not simply due to an overall reduction in peak sodium current.

Answer: We thank you for your critical analysis of our measurements and your experience. Our lab has decades of experience in measuring late Na current with many publications. Although we believe that some controversies between our opinions will remain, we tried our best to discuss this issue and have specifically performed many new experiments to improve the quality of our work, guided by your valued suggestions.

In our previous publications we performed I_{NaL} measurements at room temperature in mice and in human cardiomyocytes (e.g. Fischer et al Cardiovasc Res 2015, Dybkova et al Cardiovasc. Res 2018, Toischer et al J Mol Cell Cardiol 2013, Sossalla et al., JACC 2010). In the current study we also measured cardiomyocytes under the same condition to be able to compare them with our previous data and others (including the “specific Nav1.8-publication” by Yang et al Circ. Res 2012). To be honest, some groups measure at room temperature, some at body temperature with advantages and disadvantages. We could cite a plethora of references here for the validity of both approaches but try to avoid such kind of discussion and seek to answer more constructively.

Accordingly, to take also this concern serious we evaluated the influence of temperature on I_{NaL} in our present study and performed new key measurements in CaMKII TG mice. We here provide a data set of patch clamp measurements, where I_{NaL} was not changed at 37°C compared to room temperature. The results are demonstrated below only for your perusal. As we did not find significant differences in these data, we believe that the conditions for I_{NaL} -measurements in our study are appropriate.

Fig. A1: Mean data \pm SEM of I_{NaL} measured at room and body temperature (n=8 cells/4 mice) and at 37°C (n=6 cells/4 mice) in cardiomyocytes from CaMKII δ c^{+T} mice (p=XX, tests). I_{NaL} elicited at -35 mV for 1000 ms.

The reviewer is absolutely right regarding her/his statement on contamination of the measurements by other ion-currents: we cannot completely rule out that contaminating currents do not influence the magnitude of the measured low amplitude I_{NaL} . However, we tried (as always, see references above) to minimize these contaminating currents not with TTX but by 1) adding the Ca^{2+} -channel blocker nifedipine, 2) using niflumic acid to block Ca^{2+} -activated chloride currents, 3) adding strophantidine to block Na^+/K^+ -ATPase currents, and 4) we have omitted Ca^{2+} from the bath solution and added a sufficient amount of Mg^{2+} (Method section, page 14-15, lines 382-399).

We are very certain that measurement of I_{NaL} integral is an appropriate method to investigate and quantify late Na^+ current. This holds also true for the estimation of the amount of Na^+ which enters the cell as an important determinant of cellular Ca^{2+} handling. In particular, we trust that the integral is a more sensitive and reliable parameter to the overall changes in inactivation, since the change in the whole current is summed up, in our case on the interval of 100-500 ms after depolarization. We believe that measurement at one time point is not the best appropriate approach to represent the full I_{NaL} and might be misleading due to noise and changing kinetics. It is in sharp contrast to measurements at a defined time point which may miss other kinetics. Our method is a well-established approach that has been used in numerous published studies from our laboratory in the last ten years (e.g. Wagner et al J Clin Investig 2006, Sossalla et al J Mol Cell Cardiol 2008, Sossalla et al., J Am Coll Cardiol 2010, Toischer et al., J Mol Cell Cardiol 2013, Fischer et al., Cardiovasc Res 2015) as well as by others.

We have also tried to measure I_{NaL} at multiple potentials before, however the amplitude/values for other potentials were sometimes too low for proper analysis. Moreover, to be honest we were not that interested in current kinetics, rather in the integral which contributes to increased Ca^{2+} entry via NCX-reverse mode in the perspective of the translational background of the study. Therefore, we decided to measure I_{NaL} at a potential at which the maximum value was observed.

The reviewer also requested measurements of the peak I_{Na} . Accordingly, we performed new experiments on this. We measured I_{Na} properties in Wild-Type, Wild-Type with PF-01247324 and SCN10A knock-out cardiomyocytes. The new results are now present in the online supplement Fig. S1 and indeed show that observed alterations in I_{NaL} are not due to an overall reduction in peak sodium current. Our and previously published data (Payne et al., 2015; Yang et al. Circ Res, 2012, Casini et al. Cardiovasc Drugs Ther. 2020) confirmed that selective inhibition of Nav1.8 does not influence either the amplitude of peak I_{Na} nor its kinetics.

Fig. S3: A: Original traces and average current-voltage (I - V) relationships in mouse cardiomyocytes from wild type (WT) with or without PF-012473224 and SCN10A^{-/-} mice (WT: $n=11$ cells/5 mice; PF-01247324: $n=11$ cells/4 mice; SCN10A^{-/-}: $n=10$ cells/2 mice). Pharmacological inhibition or genetic knock-out of Nav1.8 did not affect Na^+ peak. One-way ANOVA.

B: Normalized original traces and mean data \pm SEM for steady-state inactivation (measured with the protocol in inset). The mean data for half-inactivation ($V_{1/2}$) are shown in the right panel. $V_{1/2}$ is derived from the curve by fitting it to standard Boltzmann equation: $h_{\infty} = 1 / \{1 + \exp[(V_{1/2} - V) / K_{\infty}]\}$ (WT: $n=9$ cells/5 mice; PF-01247324: $n=6$ cells/3 mice; SCN10A^{-/-}: $n=8$ cells/2 mice). Pharmacological inhibition or genetic knock-out of Nav1.8 did not affect steady-state inactivation. One-way ANOVA. **C:** Mean data \pm SEM for I_{Na} intermediate inactivation. Increasing conditioning pulse duration (P_1) reduced I_{Na} amplitude assessed with a second pulse (P_2) consistent with entry of Na^+ channels into intermediate inactivation (protocol in inset). The mean data for plateau

y_0 are shown in right panel (WT: $n=12$ cells/5 mice; PF-01247324: $n=8$ cells/5 mice; SCN10A^{-/-}: $n=11$ cells/3 mice). Pharmacological inhibition or genetic knock-out of Nav1.8 did not affect intermediate inactivation. One-way ANOVA **D**: Mean data \pm SEM for recovery from inactivation. Increasing duration of the recovery interval between conditioning pulse (P1, causing I_{Na} inactivation) and test pulse (P2) results in an exponential increase in the amplitude of I_{Na} upon P2 consistent with I_{Na} recovery (protocol in inset). The mean data for the rate constant of recovery k_{rec} are shown in right panel. Pharmacological inhibition or genetic knock-out of Nav1.8 did not affect recovery from inactivation. One-way ANOVA

- The fact that cells were pre-incubated with the drugs (and not through acute wash-in) is also problematic since it precludes measuring the effect of blockers in a pair-wise fashion. This way, it is impossible to say whether cells actually had a (large) late I_{Na} to begin with or not. It is essential to show that in those cells that have a large late I_{Na} , the blockers actually reduce late I_{Na} in that cell. This is particularly relevant since it appears that not all human hearts were studied in each treatment group. Were cells from the same heart investigated in each treatment group, and was this evenly distributed among groups? How can the authors be sure that all human hearts and cells had similar late sodium current magnitude to begin with? Were all heart failure patients clinically affected in a similar fashion? Please include detailed information on clinical status, age, gender, medication, etc. The current methodology is very risky and carries the inherent risk of selection bias. The authors should show that acute wash-in of the drugs (and not pre-incubation) gives similar results (ideally, washout should also be attempted). This is the only way to unequivocally show that the inhibitors really block late I_{Na} , and to what extent. This methodological concern also applies to the calcium and AP measurements.

Answer: The reviewer discusses different general scientific approaches. Design of experiments and clinical trials can be paired-wise and non-paired. However, paired wise approaches lack adequate controls that compensate for e.g. changes over time with respect to the mentioned experiment here. In our opinion, both approaches have their justification with different advantages and disadvantages (e.g. statistical power). We agree with the reviewer to the point, that a paired-wise measurement with a drug wash-in would be a nice approach in case of a theoretically always stable cell. However, alterations may occur with time and this may affect the measured result. In severely diseased human failing myocytes a wash-in of a drug over a time period of about 15 minutes can be problematic. With more time after rupture the severely diseased cardiomyocyte may get sensitive to changes like movements within the bath-solution due to superfusion with the drug. This may sometimes lead to increased leak currents as well as damage of the cell which consecutively falsifies current measurements. Therefore, wash-in and wash out of a drug is barely feasible and, in our experience, not the best experimental design in these diseased cardiomyocytes.

Here, we measured cells under the same condition (namely we always started to record I_{NaL} three minutes after rupturing, immediately before measurement we detected membrane capacitance, the recording lasted only some seconds). Moreover, we did not have problems with drift in the baseline which usually occurs in paired measurements due to superfusion and time effects. Therefore we believe that our approach is appropriate to investigate a drug effect on I_{NaL} by whole-cell patch clamp and this technique has been used by many other investigators and in many of our previous publication (Wagner et al., JCI 2006, Sossalla et al., J Mol Cell Card, Sossalla et al., Basic Res Cardiol 2010, Wagner et al., Circ Res 2011, Toischer et al., J Mol Cell Card 2013, Fischer et al., Cardiovasc Res 2015, Flenner et al. Cardiovasc Res 2016).

Importantly, the pre-incubation time of the drugs was always similar in all the groups. Further, in every heart all different treatment groups were measured.

Anyway, we took your expertise serious and performed novel paired-wise I_{NaL} key measurements in cardiomyocytes of CaMKII overexpressing mice with increased I_{NaL} . The specific Nav1.8-blocker PF-01247324 were washed-in as suggested. 15 minutes after wash-in of PF-01247324 I_{NaL} was significantly decreased in these cardiomyocytes in a paired-wise approach.

We hope that the reviewer appreciates our I_{NaL} -measurements in a paired-wise approach with wash-in of PF-01247324 below.

Fig. A2: paired-wise measurements of I_{NaL} in cardiomyocytes from $CaMKII\delta c^{T/T}$ mice, I_{NaL} was significantly decreased after wash-in of PF-01247324 ($p < 0.05$, paired t-test).

In addition, as requested we now provide full information about patient characteristics, clinical status and medication. From each patient cardiomyocytes were used for control measurements and treatment groups, so that inherent selection bias should not occur. The table with all patient characteristics was now added to the online supplement as supplementary figure 1.

Table S1. Overview of age, ejection fraction and medical therapy of human failing hearts used for experiments (14 patients): ICM: ischemic cardiomyopathy; EF: ejection fraction; LVEDD: Left ventricular end-diastolic diameter; ACE: ACE-inhibitor; β -B: β -blocker; DIU: diuretic; AMIO: Amiodarone (anti-arrhythmic); AT1: AT-1 receptor antagonists; MRA: mineralocorticoid receptor antagonist; CAT: catecholamines

Patient data	mean \pm SEM
Male sex (%)	85.7
age (years)	48 \pm 5
ICM (%)	50.0
Diabetes (%)	21.4
EF (%)	23 \pm 1
LVEDD (mm)	61.7 \pm 3.2
ACE (%)	57.1
β -B (%)	85.7
DIU (%)	78.5
AMIO (%)	57.1
AT1 (%)	21.4
MRA (%)	50.0
CAT (%)	21.4

- For all mouse data, results for wild type littermates should also be shown. This is essential to assess i) the impact of the CamKII δ overactivity, and ii) whether the interventions actually restore the various parameters back to the WT situation.

Answer: We thank the reviewer for this helpful suggestion. As proposed, we performed additional experiments in Wild-Type littermates of CaMKII transgenic mice for all requested data sets to assess whether Nav1.8 inhibition restores the detrimental effects of CaMKII overexpression to a Wild-Type level. The new data was added to the revised figures 2-4 and figure 5&6 and for the reviewer's perusal below. Importantly, although Nav1.8-inhibition as well as SCN10A knock-out resulted in a significant reduction in I_{NaL} and Ca²⁺-Spark-frequency, the initial level of WT littermates could not be restored. This is explainable by the fact that only Nav1.8 is blocked in our approach. Other results would be surprising in our view. I_{NaL} driven by other sodium channel isoforms like Nav1.5 and other neuronal isoforms is still present and modulated by increased CaMKII activity. With regard to the increased Ca²⁺-Spark-frequency it has to be stated that increased CaMKII-activity alone in CaMKII overexpressing mice constitutes a major driver of SR-Ca²⁺-leak.

Fig. 2: **A:** Original traces of I_{NaL} in WT and CaMKII $\delta c^{+/T}$ mouse ventricular cardiomyocytes elicited using the protocol shown in inset. **B:** Mean data \pm SEM along with individual values shown in the graph plotting WT ($n=11$ cells/5 mice) and CaMKII $\delta c^{+/T}$ (untreated: $n=17$ cells/7 mice; A-806467: $n=12$ cells/7 mice; PF-01247324: $n=12$ cells/7 mice). Probability vs. untreated (One-way ANOVA).

Fig. 3: **A:** Representative line scan images of CaMKII $\delta c^{+/T}$ ventricular cardiomyocytes. **B:** CaSpF data shown as mean \pm SEM for Wild-Type (untreated: $n=58$ cells/4 mice; PF-01247324: $n=41$ cells/4 mice; A-806467: $n=41$ cells/4 mice) and CaMKII $\delta c^{+/T}$ (untreated: $n=122$ cells/8 mice; PF-01247324: $n=105$ cells/7 mice; A-806467: $n=101$ cells/8 mice). Data was analysed by One-way ANOVA with post-hoc Bonferroni's correction.

Fig. 5: **A:** Original traces of I_{NaL} in WT, $SCN10A^{-/-}$, $CaMKII\delta^{c+/T}$ and $SCN10A^{-/-}/CaMKII\delta^{c+/T}$ mouse ventricular cardiomyocytes elicited using the protocol shown in inset. **B:** Mean data \pm SEM along with individual values shown in the graph plotting (WT: $n=7$ cells/4 mice, $SCN10A^{-/-}$: $n=10$ cells/5 mice, $CaMKII\delta^{c+/T}$: $n=9$ cells/5 mice; $SCN10A^{-/-}/CaMKII\delta^{c+/T}$: $n=4$ cells/4 mice), there was a significantly less I_{NaL} in $SCN10A^{-/-}/CaMKII\delta^{c+/T}$ cardiomyocytes compared to $CaMKII\delta^{c+/T}$. Data was analysed by One-way ANOVA with post-hoc Bonferroni's correction. **C:** Original traces of action potentials showing triggered action potentials originating from DADs in $CaMKII\delta^{c+/T}$ and $SCN10A^{-/-}/CaMKII\delta^{c+/T}$ cardiomyocytes. **D:** Graph of Mean data \pm SEM along with individual values showing DADs per minute in WT ($n=10$ cells/5 mice), $SCN10A^{-/-}$ ($n=12$ cells/5 mice), $CaMKII\delta^{c+/T}$ ($n=22$ cells/5 mice) and $SCN10A^{-/-}/CaMKII\delta^{c+/T}$ ($n=16$ cells/5 mice) cardiomyocytes. There were significantly less events of afterdepolarizations in $SCN10A^{-/-}/CaMKII\delta^{c+/T}$ compared to $CaMKII\delta^{c+/T}$ cardiomyocytes. Data was analyzed by One-way ANOVA with post-hoc Bonferroni's correction. **E:** Original traces of action potential showing EADs in $CaMKII\delta^{c+/T}$ and $SCN10A^{-/-}/CaMKII\delta^{c+/T}$ cardiomyocytes. **F:** Graph of Mean data \pm SEM along with individual values showing EADs per minute in WT ($n=10$ cells/5 mice), $SCN10A^{-/-}$ ($n=12$ cells/5 mice), $CaMKII\delta^{c+/T}$ (22 cells/5 mice) and $SCN10A^{-/-}/CaMKII\delta^{c+/T}$ (16 cells/5 mice) cardiomyocytes. There were significantly less events of afterdepolarizations in $SCN10A^{-/-}/CaMKII\delta^{c+/T}$ compared to $CaMKII\delta^{c+/T}$ cardiomyocytes. Data was analyzed by One-way ANOVA with post-hoc Bonferroni's correction. **G:** Original confocal line scans images of $CaMKII\delta^{c+/T}$ and $SCN10A^{-/-}/CaMKII\delta^{c+/T}$ cardiomyocytes showing diastolic Ca^{2+} waves **H:** Percentage of cells exhibiting waves was significantly less in $SCN10A^{-/-}/CaMKII\delta^{c+/T}$ (74 cells, 7 mice) compared to $CaMKII\delta^{c+/T}$ (104 cells, 9 mice). Data was analyzed by Chi-square test. **I:** Significantly decreased number of Ca^{2+} -waves per minute in $SCN10A^{-/-}/CaMKII\delta^{c+/T}$ compared to $CaMKII\delta^{c+/T}$. Data was analyzed by Student's t-test. Cells/mice studied, WT: $n=48$ cells/5 mice, $SCN10A^{-/-}$: $n=52$ cells/5 mice, $CaMKII\delta^{c+/T}$: $n=104$ cells/9 mice; $SCN10A^{-/-}/CaMKII\delta^{c+/T}$: $n=74$ cells/7 mice

- If no alterations are observed in calcium transients and SR content (Figure 5), then how do the authors explain the observed differences in calcium sparks (Figure 4)?

Answer: We thank the reviewer for this smart comment. Previous data of our group showed, that an increase of Ca^{2+} -transient amplitude and SR- Ca^{2+} -content is not mandatory to induce diastolic SR- Ca^{2+} -leak. In mice, Ca^{2+} -transient amplitude and SR- Ca^{2+} -content are mainly regulated by SERCA and L-Type- Ca^{2+} -current which constitute predominantly PKA-regulated mechanisms.

SR- Ca^{2+} -leak is not only dependent on SR- Ca^{2+} -content, but also on CaMKII-dependent phosphorylation of RyR2. In a very recent study from our group we found a significant induction of diastolic SR- Ca^{2+} -leak in cardiomyocytes with pharmacologically enhanced I_{NaL} that could be reversed by either inhibition of CaMKII or PKA (Eiringhaus et al, Basic Res Cardiol. 2019). Interestingly, enhanced I_{NaL} did not significantly increase Ca^{2+} -transient amplitude or SR- Ca^{2+} -content in this and some other studies. Further, CaMKII inhibition had no significant effect on Ca^{2+} -transient amplitude as well as SR- Ca^{2+} -content, whereas PKA-inhibition reduced Ca^{2+} -transient amplitude and SR- Ca^{2+} -content. This has been also shown before by our group in atrial cardiomyocytes (Fischer et. al, Cardiovasc Res. 2015). Therefore, the PKA-dependent reduction in SR- Ca^{2+} -leak can be explained by a reduction of SR- Ca^{2+} -load, whereas CaMKII-dependent SR- Ca^{2+} -leak is independent of SR- Ca^{2+} -load. We recently confirmed this hypothesis even in relation to Nav1.8-dependent I_{NaL} -inhibition. In ATX-II treated cardiomyocytes SR- Ca^{2+} -leak was increased like it was shown before, whereas selective inhibition of Nav1.8 significantly reduced SR- Ca^{2+} -leak (Bengel et al. J Mol Cell Cardiol, 2020). Interestingly, Ca^{2+} -transient amplitude and SR- Ca^{2+} -content were not significantly altered in these experiments so that mainly CaMKII-dependent mechanisms may be the major driver of SR- Ca^{2+} -leak. In line with this assumption, we found increased CaMKII-dependent phosphorylation of RyR2 at its CaMKII dependent site Ser2814 after ATX-II-treatment which was reversed after additional inhibition of Nav1.8.

These findings are explainable by the fact that increased I_{NaL} induces the reverse mode of NCX which increases the amount of Ca^{2+} within the dyadic cleft. Here, Ca^{2+} not only activates CaMKII by autophosphorylation, but also directly induces Ca^{2+} -leak from the SR. A significant reduction of reverse mode NCX by Nav1.8 inhibition could also be demonstrated in our previous publication (Bengel et al. J Mol Cell Cardiol. 2020).

- CamKII α TG mice are crossed with Scn10a KO mice. However, no information is provided on the genetic backgrounds (strains) of the 2 mouse lines. If the mice are actually of 2 separate strains, then the observed differences in survival may also be caused by this. What was the mode of death? Did the mice die of heart failure or sudden? Authors should make an effort to at least do ECG measurements and check for possible arrhythmias; ideally, with telemetry. The authors propose that decreased incidence of arrhythmias underlies the improved survival, but they do not provide any evidence for this. In vivo or ex vivo (arrhythmia inducibility in explanted hearts) evidence for this should be provided.

Answer: We are thankful to the reviewer pointing for these valid suggestions. For our project CaMKII transgenic mice from a Black-Swiss strain were crossbred with SCN10A knock-out mice from a C57BL/6 strain. The first generation of litters contained animals with heterozygote SCN10A alleles and either wild-type or transgenic CaMKII alleles. Animals from this first generation were then crossbred to achieve mice with homozygote SCN10A knock-out and transgenic CaMKII gene. For the experiments and the survival curve only animals from third generation were used. Therefore, all animals used for comparison between CaMKII transgenic with or without SCN10A knock-out (Figures 4-6) were from the same genetic background. For better illustration of the genetic background a pedigree of our breeding is shown

below. To also improve transparency on this in the manuscript we added this scheme to the online supplement.

Scheme of crossbreeding of $SCN10A^{-/-}$ and $CaMKII\delta c^{+T}$ mice: Female $SCN10A^{-/-}$ were bred with male $CaMKII\delta c^{+T}$ mice, the F1 generation revealed heterozygote $SCN10A$ knock-out and mice with transgenic or WT $CaMKII\delta c$ gene. The first genotype specification refers to the $SCN10A$ gene, the second to $CaMKII\delta c$. At the first position, $-/-$ indicates homozygote knock-out of $SCN10A$, while $-/+$ only heterozygote and $+/+$ wild-type $SCN10A$. At the second position, $+/+$ indicates an unmodified $CaMKII\delta c$ gene, while $+/T$ indicates heterozygote overexpression of $CaMKII\delta c$.

In addition, we took the reviewer's suggestion serious to further investigate ECG-parameters and telemetries in our mice. For ECG-parameters in our particular mouse model we can also refer to our recent publication (Pabel et al. Basic Res Card 2020) where we did not find any significant differences in baseline ECG-parameters between wild-type and $SCN10A$ knock-out mice. These findings go along with data from Stroud et al. (J Am heart Assoc. 2016) with similar results in ECG-parameters in these mice.

In addition, after getting acceptance for these novel animal experiments, we indeed implanted telemetries in $CaMKII$ transgenic mice with or without $SCN10A$ knock-out. As described above (see also the figure and original) we found a reduction of ventricular tachycardia in $CaMKII$ transgenic mice with $Nav1.8$ knock-out (blinded investigations). Although, it is not possible to directly observe sudden cardiac death from a technical (telemetries for the full survival curves would be needed) and ethical point of view (too many animals needed) we believe that the significant and obvious reduction of arrhythmias (in vivo and in vitro) is the major driver of increased survival in our study.

In addition, we added the ECG parameters from our telemetry experiments to the online supplement as suggested by you (Table S3).

Table S3: Baseline ECG-parameters of $CaMKII\delta c^{+T}$ and $SCN10A^{-/-} / CaMKII\delta c^{+T}$ mice: Mean data \pm SEM of HR=heart rate, QTcB (QT corrected by Bazett's formula), * $p < 0.05$ by students t-test, $n=4$ $CaMKII\delta c^{+T}$ and $n=3$ $SCN10A^{-/-} / CaMKII\delta c^{+T}$ mice

	CaMKII δ c ^{+T} (n=4)	SCN10A ^{-/-} / CaMKII δ c ^{+T} (n=3)	p-value
HR (bpm)	500.0 \pm 30.9	523.8 \pm 13.8	0.56
RR interval (ms)	125.2 \pm 9.1	117.6 \pm 3.7	0.53
ST interval (ms)	37.0 \pm 1.7	36.3 \pm 1.4	0.77
QRS complex (ms)	21.0 \pm 1.7	16.5 \pm 2.0	0.08
PR interval (ms)	35.8 \pm 3.0	34.3 \pm 0.4	0.68
QT interval (ms)	51.3 \pm 1.0	47.6 \pm 1.2	0.07
QTcB interval (ms)	146.9 \pm 1.8	139.8 \pm 1.9	0.04 (*)

- Figure 7: Again, the methodology for late I_{Na} measurement is not appropriate. The typical example shows only a very small difference in late I_{Na} at the end of the 500 ms trace, and certainly not a doubling as indicated in the average values in panel B. These should be repeated as indicated above, and values for Scn10 KO and WT mice should also be provided, in addition to peak sodium current values. In panels C/D, the protocol for EAD/DAD measurements is suboptimal, since it is impossible to say whether the observed extra APs are spontaneous or triggered. Instead, a fast pacing protocol should be used to assess triggered activity, EADs and DADs. Additional hearts should be studied since n=3 is a very small number and it leaves open the possibility that the data is driven by for instance one suboptimal isolation leading to low quality cells which are depolarized and show increased spontaneous activity. Hence, all AP parameters should be provided, including resting membrane potential, amplitude and duration; depolarized cells should be excluded from the analysis.

Answer: We apologize for not presenting a perfect representative original registration in Fig.5 A and replaced this with one which fits better to the mean values. In addition, we performed new I_{NaL} measurements in wild-type and SCN10A knock-out cardiomyocytes as requested. As expected, there were no significant differences in I_{NaL} between Wild-Type and SCN10A knock-out cardiomyocytes at baseline. SCN10A knock-out in CaMKII transgenic mice led to a significant reduction in I_{NaL} compared to CaMKII transgenic alone. However, the level of I_{NaL} in WT or SCN10A knock-out cardiomyocytes could not be restored. This is explainable by the fact that other sodium channel isoforms than Nav1.8 are still expressed and their function is still modulated by increased CaMKII activity. These new results are now presented in the manuscript in the revised figure 6B.

Fig. 5: **A:** Original traces of I_{NaL} in WT, $SCN10A^{-/-}$, $CaMKII\delta c^{+/T}$ and $SCN10A^{-/-}/CaMKII\delta c^{+/T}$ mouse ventricular cardiomyocytes elicited using the protocol shown in inset. **B:** Mean data \pm SEM along with individual values shown in the graph plotting (WT: $n=7$ cells/4 mice, $SCN10A^{-/-}$: $n=10$ cells/5 mice, $CaMKII\delta c^{+/T}$: $n=9$ cells/5 mice; $SCN10A^{-/-}/CaMKII\delta c^{+/T}$: $n=4$ cells/4 mice), there was a significantly less I_{NaL} in $SCN10A^{-/-}/CaMKII\delta c^{+/T}$ cardiomyocytes compared to $CaMKII\delta c^{+/T}$. Data was analysed by One-way ANOVA with post-hoc Bonferroni's correction. **C:** Original traces of action potentials showing triggered action potentials originating from DADs in $CaMKII\delta c^{+/T}$ and $SCN10A^{-/-}/CaMKII\delta c^{+/T}$ cardiomyocytes. **D:** Graph of Mean data \pm SEM along with individual values showing DADs per minute in WT ($n=10$ cells/5 mice), $SCN10A^{-/-}$ ($n=12$ cells/5 mice), $CaMKII\delta c^{+/T}$ ($n=22$ cells/5 mice) and $SCN10A^{-/-}/CaMKII\delta c^{+/T}$ ($n=16$ cells/ 5 mice) cardiomyocytes. There were significantly less events of afterdepolarizations in $SCN10A^{-/-}/CaMKII\delta c^{+/T}$ compared to $CaMKII\delta c^{+/T}$ cardiomyocytes. Data was analyzed by One-way ANOVA with post-hoc Bonferroni's correction. **E:** Original traces of action potential showing EADs in $CaMKII\delta c^{+/T}$ and $SCN10A^{-/-}/CaMKII\delta c^{+/T}$ cardiomyocytes. **F:** Graph of Mean data \pm SEM along with individual values showing EADs per minute in WT ($n=10$ cells/5 mice), $SCN10A^{-/-}$ ($n=12$ cells/5 mice), $CaMKII\delta c^{+/T}$ (22 cells/5 mice) and $SCN10A^{-/-}/CaMKII\delta c^{+/T}$ (16 cells/ 5 mice) cardiomyocytes. There were significantly less events of afterdepolarizations in $SCN10A^{-/-}/CaMKII\delta c^{+/T}$ compared to $CaMKII\delta c^{+/T}$ cardiomyocytes. Data was analyzed by One-way ANOVA with post-hoc Bonferroni's correction. **G:** Original confocal line scans images of $CaMKII\delta c^{+/T}$ and $SCN10A^{-/-}/CaMKII\delta c^{+/T}$ cardiomyocytes showing diastolic Ca^{2+} waves **H:** Percentage of cells exhibiting waves was significantly less in $SCN10A^{-/-}/CaMKII\delta c^{+/T}$ (74 cells, 7 mice) compared to $CaMKII\delta c^{+/T}$ (104 cells, 9 mice). Data was analyzed by Chi-square test. **I:** Significantly decreased number of Ca^{2+} -waves per minute in $SCN10A^{-/-}/CaMKII\delta c^{+/T}$ compared to $CaMKII\delta c^{+/T}$. Data was analyzed by Student's *t*-test. Cells/mice studied, WT: $n=48$ cells/5 mice, $SCN10A^{-/-}$: $n=52$ cells/5 mice, $CaMKII\delta c^{+/T}$: $n=104$ cells/9 mice; $SCN10A^{-/-}/CaMKII\delta c^{+/T}$: $n=74$ cells/7 mice

We further studied extra hearts for EAD and DAD measurements. Also missing wild-type and SCN10A knock-out controls were added as requested. The new results are presented in Figure 7A, D, E, F and I. As already indicated before, measurements of cardiomyocytes isolated from SCN10A knock-out in CaMKII transgenic mice exhibit significantly reduced incidence of both EADs and DADs compared to CaMKII TG alone. Here also, a complete restoration of the wild-type and SCN10A knock-out level could

not be achieved because of the reasons elucidated above. Related to the incidence of DADs it further has to be stated, that SR-Ca²⁺-leak caused by CaMKII overexpression alone might be one of the drivers of DAD generation.

In addition, we now provide the requested table with all AP parameters of our experiments (AP-duration, amplitude, resting membrane potential and upstroke velocity). This table was added to the online supplement. Whereas AP-duration was significantly abbreviated in cardiomyocytes from CaMKII transgenic mice with SCN10A knock-out compared to CaMKII TG alone, other AP-parameters were not affected. SCN10A knock-out cardiomyocytes without CaMKII overexpression did not differ from WT cardiomyocytes in all AP-parameters.

Table S2: Action potential characteristics of ventricular cardiomyocytes from Wild-Type, SCN10A^{-/-}, CaMKIIδc^{+T} and SCN10A^{-/-} / CaMKIIδc^{+T} mice (Supplement to Fig. 5): Mean data±SEM of dv/dtmax, action-potential amplitude, resting membrane potential and action potential duration at 0.5, 1 and 2 Hz. Significance is indicated as *p<0.05 vs. WT

	Wild-Type	SCN10A ^{-/-}	CaMKIIδc ^{+T}	SCN10A ^{-/-} / CaMKIIδc ^{+T}
dv/dtmax (mV/ms)	80.7 ± 2.8	74.5 ± 11.3	73.7 ± 9.7	86.4 ± 12.8
AP amplitude (mV)	97.0 ± 4.2	89.9 ± 4.2	95.7 ± 3.7	99.36 ± 4.9
RM Potential (mV)	-68.3 ± 1.3	-68.4 ± 1.1	-68.1 ± 1.2	-70.8 ± 1.7
APD 90 0.5 Hz (ms)	43.6 ± 5.9	49.0 ± 8.9	148.8 ± 34.6 *	129.7 ± 18.9*
APD 90 1 Hz (ms)	42.9 ± 5.8	49.6 ± 9.1	147.8 ± 39.0 *	127.1 ± 18.2*
APD 90 2 Hz (ms)	48.4 ± 8.0	58.2 ± 11.0	156.8 ± 38.8 *	144.1 ± 20.2*

- Details are missing from the methods: gender of the mice; temperature at which the AP measurements were performed, temperature at which the calcium transients and sparks were performed

Answer: We thank the reviewer for these suggestions regarding our method section. In the revised manuscript we now provide the requested information for every method used in our experiments.

In summary, we thank you very much for the intensive and critical review of our manuscript. We have tried everything possible with new experiments (*in vitro* and *in vivo*), comparisons, technical demonstrations, and endeavored discussions to meet the reviews and think that your efforts have improved our work significantly.

Reviewer #2 (Remarks to the Author):

This is an interesting paper that presents new and potentially important evidence supporting a proarrhythmic interaction between CaMKII and Nav1.8. The strengths of the study are its novelty, clinical/translational importance, and the use of a Nav1.8 (SCN10A^{-/-}) knock out model. However, there are important weaknesses that should be addressed. These include the need to determine if the now well known interaction between CaMKII and Nav1.5 is conserved for CaMKII and Nav1.8, establish specificity of the Nav1.8 antagonist drugs, using the Nav1.8 knock out cardiomyocytes as controls, measuring in vivo arrhythmias, which are the presumed basis for partial longevity rescue in the CaMKII^d TG x SCN10A^{-/-} interbred mice, and increasing the number of unique human samples.

Answer: We thank the reviewer for appreciating our work and the precise help in order to improve the manuscript.

Specific comments:

Are the domains of Nav1.5 known to be important for CaMKII binding and enhanced late current conserved in Nav1.8? Does CaMKII bind Nav1.5 and Nav1.8 by similar sites/mechanisms?

Answer: We thank the reviewer for this question. However, the answer to this question is rather complex and hence, we tried to approach this topic extensively in a theoretic manner. The interplay of Nav1.5 and CaMKII was investigated by several groups including ours before. Of note, despite a lot of research has been done some issues still remain under debate. The interaction of Nav1.5 and CaMKII was first described by Wagner et al (J Clin Invest, 2006). They found an interaction between Nav1.5 and CaMKII (co-immunoprecipitation) and a co-localization between CaMKII and Nav1.5 (immunostainings), and most importantly, a CaMKII-dependent phosphorylation of Nav1.5. Further, CaMKII overexpression was shown to enhance I_{NaL} , which was reversible on CaMKII inhibition with KN93. The known effects of CaMKII interaction on Nav1.5 channel are a slowed fast inactivation, a hyperpolarizing shift in steady-state inactivation as well as an increased late I_{Na} , and a slowed recovery from inactivation.

Structure of Nav1.5 and Nav1.8 - mechanism of CaMKII binding

Both Nav1.5 and Nav1.8 are voltage gated sodium channels with similar structure and transmembrane organization. Both channels form 4 transmembrane domains and 3 intracellular loops that may become target for posttranslational modification. For both channels the first intracellular loop is known to be involved in posttranslational modulation of channel function e.g. by phosphorylation. In order to further investigate possible similarities between Nav1.5. and Nav1.8, we performed a NCBI-Protein-Blast for the two proteins and found that the amino acid sequence of both channels matches in 62.3%.

Research from the Mohler group revealed, that CaMKII is associated to Nav1.5 at the intercalated disc within a large macromolecular complex. For binding of Nav1.5 to the cell surface, binding of Nav1.5 with Ankyrin-G at a specific binding motif (amino acids 1047-1055: **VPIAVAESD**) within the DII-III linker is required (Mohler et al. PNAS 2004). Similar observations were made for Nav1.8, where a binding of Nav1.8 to Ankyrin-G was reported to bind at the Nav1.8 amino acids 996-1029 (**VPIAEGESD**LDDLEDDGGEDAQSFQQEVIPKGQ) containing a nearly identical sequence (marked in yellow) in loop 2 within the DII-III linker (Montersino et al. J. Neurochem. 2014). A single aa mutation in human Nav1.5 at position E1053 was shown to block ankyrin-G binding, also disrupts Nav1.5 surface expression in cardiomyocytes (Mohler et al., 2014). Within cardiomyocytes, Ankyrin-G is further necessary for binding of β 4-spectrin to the intercalated disc and to Nav1.5 (Makara et al. Circ Res 2014). This interaction with β 4-spectrin is important as β 4-spectrin binds CaMKII and therefore targets CaMKII to the intercalated disc and to Nav1.5 (Hund et al JCI 2010) (Figure A3).

Fig. A3: Model of CaMKII targeted to Nav1.5 in a macrodomain with Ankyrin-G and β 4-spectrin (Figure from Koval et al Circulation 2012)

CaMKII-dependent phosphorylation sites of Nav1.5 regulating the I_{Na} :

As the first, the Mohler group identified Ser571 within the DI-DII linker as a potential target of CaMKII as they found a decreased phosphorylation of Ser571 after disruption of the β IV-spectrin dependent interaction between CaMKII and Nav1.5 (Hund et al JCI 2010). Later, Ashpole et al identified Ser516 and Thr594 by P32 incorporation with *in vitro* kinase assays (Ashpole et al J Biol Chem. 2012). However, no evidence for phosphorylation at the Ser571 site was identified in this study. In HEK293 cells expressing the Nav1.5 alpha subunit, CaMKII dependent effects on I_{Na} could be observed. Inhibition of CaMKII with AIP or mutation of any of the three phosphorylation sites Ser516, Ser571 or T594 to non-phosphorylatable alanine reversed these effects.. An interesting phospho-proteomic study from Marionneau et al (J Proteome Res 2012) found Ser571 to be phosphorylated already at baseline in unstimulated mouse Nav1.5, whereas phosphorylation of Ser516 and Thr594 was not seen under these conditions. In contrast to the studies of the Bers group, Glynn et al showed that phosphorylation at Ser571 does not regulate channel properties that were linked to CaMKII dependent phosphorylation before, but exclusively regulates I_{NaL} (Glynn et al, Circulation 2015). This was further linked to action potential prolongation and increased arrhythmia *in-vivo*. Interestingly, the Ashpole study in HEK293 cells did not detect any CaMKII induced I_{NaL} in their setting indicating that phosphorylation at the alpha subunit alone seems not to be sufficient to generate I_{NaL} . Therefore, other proteins like the β -subunits of Nav1.5 might be necessary which were present in the cardiomyocyte studies from Glynn et al.

Using mass spectrometry Herren et al. confirmed CaMKII-dependent phosphorylation at Ser516 and Ser571 in human Nav1.5 from HEK293 cells, without confirming phosphorylation at Thr594 (J Proteome Res 2015). In further studies surprisingly, CaMKII dependent phosphorylation at Ser516 was decreased in human heart failure despite confirmed CaMKII overactivity demonstrated by increased pCaMKII (Thr287) in the tissue. In contrast, Koval et al (Circulation 2012) showed that Ser571 appears to be hyperphosphorylated in human failing hearts.

In summary, although CaMKII dependent regulation of Nav1.5 was approved in several studies, the exact mechanisms of channel regulation and I_{NaL} formation, as well as the role of the different phosphorylation sites of Nav1.5 still remain a matter of debate.

In addition to the three described phosphorylation sites that are more or less established (Ser516, Ser571 or T594), the study from Herren et al (J Proteome Res 2015) found in total 23 different possibly CaMKII dependent phosphorylation sites in Nav1.5 using label free mass-spectrometry. 18 of these sites are located in the DI-II linker. Especially for Thr455/Ser460 and Ser510/Ser516 it was shown that CaMKII increases the phosphorylation at these residues. Therefore, CaMKII might also regulate Nav1.5 function by other Serin or Threonine residues than Ser516, Ser571 and Thr594 that were not specifically examined by now.

Conservation of CaMKII dependent Serin residues and binding motifs in Nav1.8:

Compared to this broad knowledge about CaMKII-dependent phosphorylation of Nav1.5, there is only little evidence about modification of Nav1.8 function by phosphorylation. By now, phosphorylation of Nav1.8 within the first intracellular loop was reported for PKA and MAPK-P38 at multiple different Serin and Threonine residues (Fitzgerald et al. J. physiol. 1999, Hudmon et al. J. Neurosci 2008). However, an interaction of Nav1.8 with CaMKII was not described before. Therefore, despite several homologous amino acid residues between Nav1.5 and Nav1.8, statements on binding sites/mechanisms and relevance for I_{NaL} augmentation of CaMKII and Nav1.8 remain speculative. However, there are certain

similarities between Nav1.5 and Nav1.8 supporting a phosphorylation and regulation of Nav1.8 by CaMKII:

A closer look to the DI-II loop revealed that several Serin and Threonine residues are conserved between Nav1.5 and Nav1.8. Nav1.5 Serin residues 457, 460, 464, 483, 499 and 528 that were described to be phosphorylated by CaMKII in the Herren et al study are conserved in Nav1.8 as Ser441, 444, 448, 467, 479 and 502. Additionally, Nav1.5 Ser664 and Ser667 are conserved as Ser612 and Ser615 in Nav1.8. The blast of the amino-acid sequences of the DI-II linker of Nav1.5 and Nav1.8 are shown below. The CaMKII-dependent phosphorylation sites in Nav1.5 from Herren et al are marked in yellow. Conserved residues between Nav1.5 and Nav1.8 are marked in green.

I-II Loop

```

Nav1.5 HUMAN 416 YEEQNQATIAETEEKEKRFQEAEMMLKKEHEALTI-----RGVDT-----VSRSSLEMSPL
Nav1.8 HUMAN 400 YEEQNQATTDEIEAKEKKFQEALEMLRKEQEVLAALGIDT-----TGLSHNGSPL

Nav1.5 HUMAN 467 APVNSHERRSKRRK-----SSGTEECGEDRLPKSDESDGPRAMN-----HLSL--
Nav1.8 HUMAN 451 TSKNASERRHRKPRV-----EGSTE-D-NKSPRDOPYN-Q-----

Nav1.5 HUMAN 512 T-----SRTSMKPRSSRGSIFTFR-R-R--DLGSEADFADDENSTAGESESHHTSLLVWPWL--TSA---
Nav1.8 HUMAN 489 FLGLASG--K-----HG--VFHFRSPG-----LPEGVTDDGVFPGDHESHRSLLLGGGAGQQGP-----

Nav1.5 HUMAN 573 --QGQPS-PGTSAPGHALHGKKNSTVDCNGVVSLLGAGDPE-ATSPGSHLLRPVMLEHPP-D-TTTPSEE
Nav1.8 HUMAN 549 -----LPRSPLPQPSNPDS-R-HG-----EDEHQPPPTSELAP-G-----A-VDV

Nav1.5 HUMAN 637 PGGPQMLTSQAPCVDGFE---EPGARQ-----SAVSVLTSA-LEELEESRHKCPPCWNRLAQRyliWECCPL
Nav1.8 HUMAN 585 SAFDAGQKKTFLSAEYLD---EPFRAQ-----VSVSIITSV-LEELEESEQKCPCLTSLSQKYLWIDCCPM

```

Fig. S9: Amino acid sequence comparison of the I-II intracellular loop of Nav1.5 and Nav1.8, amino acids sequences as published by Herren et al. *J Proteome Re.* 2015 (online supplemental Fig.S9 in our work), CaMKII dependent phosphorylation sites from Herren et al are marked in yellow. Conserved Serin residues between Nav1.5 and Nav1.8 are marked in green. Serin residues with the CaMKII consensus motif are marked in red (serin/threonin residues of the consensus motif are marked in yellow).

Hund et al proposed a binding of CaMKII to a specific motif in Nav1.5 Arg-X-X-Ser to be necessary for CaMKII binding at Ser571. Moreover, this consensus motif Arg-X-X-Ser is also present at the phosphorylation site Ser516 and S664 of Nav1.5.

Interestingly, Nav1.8 contains the same binding consensus motif at positions Ser 488, Ser500, Ser515 and Ser612 (homologous to Nav1.5 Ser 664).

The sequence and the Serin residues that show the Arg-X-X-Ser binding motif are marked in red in the figure above which was added to the online supplement as supplementary Fig. S9. We included the description in the online supplement and the discussion (page 8, line 196-203).

To further investigate possible phosphorylation at these Serin residues we performed a phosphorylation prediction. A score of 1.0 would mean a high probability of phosphorylation by a kinase at the specific site. Interestingly, the prediction score for CaMKII at the known phosphorylation sites Ser516 and Ser571 of Nav1.5 is a 0.467 and 0.457. Interestingly, for Nav1.8 several Serin residues could be identified with a similar phosphorylation prediction by CaMKII. The results of the phosphorylation prediction are reported below for the sites with the consensus motif Ser488 (0.497), Ser500 (0.464), Ser 515 (0.424), and Ser612 (0.477).

Nav1.5

Sequence	#	x	Context	Score	Kinase	Answer
Sequence	516	S	TRGLSRTSM	0.467	CaM-II	.
Sequence	516	S	TRGLSRTSM	0.339	p38MAPK	.
Sequence	516	S	TRGLSRTSM	0.246	PKA	.
Sequence	571	S	LRRPSTQGG	0.996	unsp	YES
Sequence	571	S	LRRPSTQGG	0.849	PKA	YES
Sequence	571	S	LRRPSTQGG	0.457	CaM-II	.

# Sequence	571	S	LRRPSTQGQ	0.288	p38MAPK	.
Nav1.8						
Sequence	#	x	Context	Score	Kinase	Answer
# Sequence	488	S	QRRMSFLGL	0.985	unsp	YES
# Sequence	488	S	QRRMSFLGL	0.773	PKA	YES
# Sequence	488	S	QRRMSFLGL	0.497	CaM-II	.
# Sequence	488	S	QRRMSFLGL	0.284	p38MAPK	.
# Sequence	500	S	KRRASHGSV	0.997	unsp	YES
# Sequence	500	S	KRRASHGSV	0.851	PKA	YES
# Sequence	500	S	KRRASHGSV	0.464	CaM-II	.
# Sequence	500	S	KRRASHGSV	0.329	p38MAPK	.
# Sequence	515	S	GRDISLPEG	0.984	unsp	YES
# Sequence	515	S	GRDISLPEG	0.785	PKA	YES
# Sequence	515	S	GRDISLPEG	0.424	CaM-II	.
# Sequence	515	S	GRDISLPEG	0.237	p38MAPK	.
# Sequence	612	S	QRAMSVVSI	0.987	unsp	YES
# Sequence	612	S	QRAMSVVSI	0.516	PKA	YES
# Sequence	612	S	QRAMSVVSI	0.477	CaM-II	.
# Sequence	612	S	QRAMSVVSI	0.332	p38MAPK	.

Fig. A4: Phosphorylation prediction of CaMKII for different phosphorylation sites at Nav1.5 and Nav1.8

However, despite we performed this theoretical work the mechanisms of CaMKII binding to Nav1.8 and its phosphorylation-dependent regulation remain highly speculative (as seen for Nav1.5).

In summary, CaMKII dependent phosphorylation of Nav1.5 is a well-known regulatory mechanism of Nav1.5 channel function. Nevertheless, the exact mechanisms how the interaction of CaMKII and Nav1.5 leads to an increased I_{NaL} are not completely elucidated by now. Within the first intracellular loop of Nav1.8 several Serin and Threonine residues were predicted to be targeted by CaMKII. Among these, four residues exist that show the consensus binding motif which was reported for CaMKII binding at Nav1.5. Phosphorylation prediction scores for these sites are comparable to the scores that were found at the established CaMKII dependent sites of Nav1.5. Furthermore, we demonstrated nearly identical amino acid motifs in Nav1.5 and Nav1.8 binding Ankyrin-G for cell surface localization.

We hope the reviewer is satisfied by this theoretical work. An experimental analysis of binding sites and CaMKII dependent phosphorylation of Nav1.8 including amino acid substitutions would fall beyond the scope of this manuscript and should be part of a separate follow-up study.

The human studies shown in Fig 1, with a very small number of unique samples, need some form of quantification. Why only n=4? Even for hard to obtain human specimens, this number is low – too low to make the point that it is an important attribute of failing myocardium in patients.

Answer: We thank the reviewer for these helpful comments. We took this criticism seriously and performed additional experiments using further human failing and non-failing samples. These new results are now included in the manuscript to increase the number of experiments. Now the total number of patients is 7 each. In the revised manuscript we replaced figure 1A with hopefully better new originals.

Fig. 1: A Co-immunoprecipitation of CaMKII δ and Nav1.8 from left ventricular homogenates of human non-failing and failing hearts. These experiments were performed in 7 independent NF and 7 independent HF myocardium samples.

Was the specificity of these antagonists proven in the Nav1.8 ko myocardial cells? Does it lack all activity against Nav1.5?

Answer: We thank the Reviewer for her/his comment. We allow ourselves to give a brief overview of the evidence on this topic and then present the newly carried out experiments on this particular question. We and others could show before, that both blockers A-803467 and PF-01247324 are Nav1.8-specific at the used concentrations (30 nml/L for A-803467 and 1 μ mol/L for PF-01247324). Yang et al (Circ Res 2012) performed experiments in ND7/23 cells showing a dose dependent selectivity for A-803467 on Nav1.8 over Nav1.5. While 30 nmol/L of A-803467 abolished all Nav1.8 dependent I_{NaL} -current, 1000 nmol/L did not show any relevant effects on Nav.1.5-dependent I_{NaL} . After I_{NaL} -augmentation with ATX-II an IC50 of 10 nmol/L for Nav1.8 dependent I_{NaL} was calculated, while IC50 of A-803467 for Nav1.5 was 5000 nmol/L.

PF-01247324 was tested for its Nav1.8-selectivity by Payne et al (Payne et al. Br J Pharmacol 2015). Here, a 50-fold Nav1.8-selectivity over Nav1.5 of the compound could be observed. At a concentration of 1 μ mol/L a complete block of Nav1.8-channels occurred while Nav1.5-dependent current was unaffected at this dose.

In line with these reports, we showed in our previous work that both blockers mediated significant effects on I_{NaL} in wild-type murine cardiomyocytes treated with either isoprenaline or ATX-II (Dybкова et al. Cardiovasc Res 2018, Bengel et al, J Mol Cell Cardiol 2020). However, in these studies in SCN10A knock-out cardiomyocytes there were no significant effects of both blockers on I_{NaL} , AP-duration or SR-Ca $^{2+}$ -leak, while profound effects in cardiomyocytes from wild-type could be detected. These results support the hypothesis that the compounds A-803467 and PF-01247324 are selective for Nav1.8 in the used concentrations.

However, to provide also sufficient new experiments on your question, we established a CRISPR-Cas9 homologous knock-out of *SCN10A* in iPSCs and differentiated them into ventricular iPS cardiomyocytes. Here, we also investigated the drug PF-01247324. After enhancement of I_{NaL} with isoproterenol, we found profound effects of the specific blocker PF-01247324 in control iPS cardiomyocytes. In iPS-cardiomyocytes with *SCN10A* knock-out we did not observe any effect of the blocker on I_{NaL} again, confirming the specificity of the drug. The results are presented below and were included in the manuscript as Figure 2E & 2F.

Fig. 2: **E:** Original traces of I_{NaL} from human ventricular SCN10A knock-out iPSC-cardiomyocytes elicited using the protocol shown in inset. **F:** Mean data \pm SEM along with individual values shown in the graph plotting (control + Iso: $n=19$ cells/x cardiac differentiations; control + Iso+ PF: $n=4$ cells/x differentiations; KO + Iso: $n=11$ cells/x differentiations; KO+Iso+PF-01247324=12 cells/x differentiations). Probability vs. control+ Iso (One-way ANOVA).

Moreover, to ultimately respond to your question we performed peak I_{Na} measurements in wild-type mouse cardiomyocytes from our utilized mouse model in the presence of the blocker PF-01247324. We found no relevant effects of PF-01247324 compared to untreated controls promoting our hypothesis that Nav1.5 is not affected by the used concentration of PF-01247324. The results of these experiments are presented below and will be provided as a new Fig. S3 in the online supplement and described in the results (page 5, lines 97-98).

Fig. S3: **A:** Original traces and average current-voltage ($I-V$) relationships in mouse cardiomyocytes from wild type (WT) with or without PF-01247324 and SCN10A^{-/-} mice (WT: $n=11$ cells/5 mice; PF-01247324: $n=11$ cells/4 mice; SCN10A^{-/-}: $n=10$ cells/2 mice). Pharmacological inhibition or genetic knock-out of Nav1.8 did not affect Na⁺ peak. One-way ANOVA.

B: Normalized original traces and mean data \pm SEM for steady-state inactivation (measured with the protocol in inset). The mean data for half-inactivation ($V_{1/2}$) are shown in the right panel. $V_{1/2}$ is derived from the curve by fitting it to standard Boltzmann equation: $h_{\infty} = 1 / \{1 + \exp[(V_{1/2} - V) / K_{1/2}]\}$ (WT: $n=9$ cells/5 mice; PF-01247324: $n=6$ cells/3 mice; SCN10A^{-/-}: $n=8$ cells/2 mice). Pharmacological inhibition or genetic knock-out of Nav1.8 did not affect steady-state inactivation. One-way ANOVA. **C:** Mean data \pm SEM for I_{Na} intermediate inactivation. Increasing conditioning pulse duration (P_1) reduced I_{Na} amplitude assessed with a second pulse (P_2) consistent with entry of Na⁺ channels into intermediate inactivation (protocol in inset). The mean data for plateau y_0 are shown in right panel (WT: $n=12$ cells/5 mice; PF-01247324: $n=8$ cells/5 mice; SCN10A^{-/-}: $n=11$ cells/3 mice).

Pharmacological inhibition or genetic knock-out of $Na_v1.8$ did not affect intermediate inactivation. One-way ANOVA **D**: Mean data \pm SEM for recovery from inactivation. Increasing duration of the recovery interval between conditioning pulse (P1, causing I_{Na} inactivation) and test pulse (P2) results in an exponential increase in the amplitude of I_{Na} upon P2 consistent with I_{Na} recovery (protocol in inset). The mean data for the rate constant of recovery k_{rec} are shown in right panel. Pharmacological inhibition or genetic knock-out of $Na_v1.8$ did not affect recovery from inactivation. One-way ANOVA

Does $Na_v1.8$ ko change expression of $Na_v1.5$?

Answer: We thank the Reviewer for her/his comment. The mouse model used in our study is generated by a disruption of the exons 4 and 5 of $SCN5A$. Strout et al used the same KO mice model (originally introduced by Akopian et al, Nat Neurosci. 1999) and demonstrated a similar expression of $Nav1.5$ in hearts from WT and $SCN10A$ knock-out mice on the mRNA level (Stroud et al. J Am Heart Assoc. 2016).

However, to more specifically address the reviewers question we performed Western-Blot experiments and qPCR analysis from tissue of Wild-Type and $SCN10A$ -knock-out heart with and without $CaMKII$ overexpression. As the novel figure S6 below depicts, there was no significant effect of $SCN10A$ knock-out on $Nav1.5$ protein and mRNA expression. We included the figure in the supplements as supplemental Fig. S6 and in the results/discussion (page 6, lines 141-43). In addition, we performed functional peak I_{Na} measurements comparing WT and $SCN10A$ -KO cardiomyocytes, which showed no significant differences in peak I_{Na} amplitude or channel kinetics. Therefore, it can be assumed that $Nav1.5$ expression and function is not affected by $SCN10A$ knock-out. We hope the reviewer accepts these experiments to resolve any doubts.

Fig. S6: **A:** Original western blots of $Nav1.5$ in myocardial tissue of WT, $SCN10A^{-/-}$, $CaMKII\delta c^{+/T}$ and $SCN10A^{-/-}/CaMKII\delta c^{+/T}$ mice. **B:** Western blot quantification of $Nav1.5$ normalized to global protein loading. $Nav1.8$ knock-out did not influence $Nav1.5$ expression neither in $SCN10A^{-/-}$ vs. WT nor in $SCN10A^{-/-}/CaMKII\delta c^{+/T}$ vs. $CaMKII\delta c^{+/T}$ in mouse myocardium tissues (WT: n=6, $SCN10A^{-/-}$: n=5, $CaMKII\delta c^{+/T}$: n=6; $SCN10A^{-/-}/CaMKII\delta c^{+/T}$ =7. One-way ANOVA **C:** mRNA expression of $Nav1.5$ in myocardial tissue and isolated murine cardiomyocytes. $Nav1.8$ knock-out did not influence $Nav1.5$ mRNA expression neither in mouse myocardial tissue (WT n=5, $SCN10A^{-/-}$ n=4, $CaMKII\delta c^{+/T}$ n=4; $SCN10A^{-/-}/CaMKII\delta c^{+/T}$ =5) nor in cardiomyocytes (WT n=5, $SCN10A^{-/-}$ n=5, $CaMKII\delta c^{+/T}$ =5; $SCN10A^{-/-}/CaMKII\delta c^{+/T}$ =5). One-way ANOVA

Does AIP have an amphipathic tag for membrane permeation or was it dialyzed in the pipette solution? If the former (for voltage clamp or Ca^{2+} spark studies), there should be a control peptide with a similar tag because membrane currents are typically affected by myristoylation, palmitoylation, TAT or other molecular adducts designed to enhance cell membrane permeation.

Answer: We thank the reviewer for this comment. In the present study we used myristoylated AIP and preincubated cardiomyocytes with the inhibitor for 15 min before measurements. We did not use a control peptide and are not aware of such a compound. However, inhibition of $CaMKII$ has been performed in our laboratory for the last fifteen years and the inhibitor has been used in numerous projects published with significant impact (e.g. Wagner et al., J Clin Investig. 2006, Sossalla et al Circ Res 2010, Wagner et al., Circ Res 2011, Toischer et al., J Mol Cell Cardiol 2013, Fischer et al., Eur J Heart Fail. 2014). We are aware of the need to use a control compound for the $CaMKII$ inhibitor KN93. AIP is widely used without a control compound and this is accepted to be state of the art. Unless we are not very much mistaken, there exists a control peptide for ACE3-I, but not for AIP.

Death by arrhythmia is increased in $CaMKII\delta$ TG mice, so it is plausible that $SCN10A^{-/-}$

interbreeding reduces mortality by decreasing arrhythmia. However, this should be tested directly **in vivo**.

Answer: We thank the reviewer for her/his comment.

However, to satisfy the reviewer we designed an amendment to our existing permission (which unfortunately took us many months) that allowed us to implant telemetry monitors in a limited number of CaMKII animals.

With the available equipment we tried to analyse the cause of death in a subset of mice (CaMKII TG vs. CaMKII transgenic + Nav1.8 knock-out) for 2 weeks at an age older than 8 weeks. As expected, a significant number of episodes of PVCs and VTs were detected in CaMKII transgenic mice that was significantly lower in CaMKII transgenic mice with additional Nav1.8 knock-out. Unfortunately, death occurred neither in CaMKII transgenic nor in CaMKII transgenic + Nav1.8 knock-out mice during this time period.

We therefore included the following sentence in the result section of the revised manuscript: "The Nav1.8 knock-out significantly reversed a relevant part of the arrhythmogenic CaMKII δ c transgenic substrate *in vitro* and *in vivo* which is known to be associated with sudden cardiac death. However, due to the sporadic nature of sudden cardiac death, we were not able to correlate mortality with ventricular arrhythmias in these CaMKII δ c transgenic mice as this is technically and ethically not feasible." (Page 10, Line 261-265).

The results of the recordings are presented below and in the revised manuscript as Fig.6.

Fig. 6: **A:** Original ECG-traces of CaMKII δ c^{+T} and SCN10A^{-/-}/CaMKII δ c^{+T} showing ventricular arrhythmias. **B:** Unchanged activity levels in SCN10A^{-/-}/CaMKII δ c^{+T} compared to CaMKII δ c^{+T} (CaMKII δ c^{+T} (4 mice); SCN10A^{-/-}/CaMKII δ c^{+T} (3 mice)). **C:** Reduction of premature ventricular contractions in SCN10A^{-/-}/CaMKII δ c^{+T} ($p=0.08$, student's t-test). **D:** Reduction of ventricular tachycardia in SCN10A^{-/-}/CaMKII δ c^{+T} ($p<0.05$, student's t-test).

Similarly, the NaV1.8 antagonists, after proof of specificity in SCN10A^{-/-} cardiomyocytes,

should be used to protect against arrhythmias in vivo in CaMKII δ TG mice, at least in an acute study.

Answer: We again thank the reviewer for her/his comment. Indeed, it would be very interesting to test a Nav1.8 antagonist in vivo in CaMKII transgenic mice. However, both implantation of telemetries and invasive EP-studies in animals require permission of the local ethics authority. As CaMKII transgenic mice suffer from heart failure it is was hard to get a permission to perform surgery or EP-studies in these mice. This kind of additional in vivo studies is nearly impossible to attain. We also try to stick to the principles of the 3Rs of animal experiments and limit in vivo investigations to absolutely necessary experiments (e.g. one heart failure model). We believe from a mechanistic point of view, a KO of Nav1.8 in CaMKII overexpressing mice, is the best and most specific approach to support our hypothesis. Therefore, regarding additional in-vivo investigations we focused on implantation of telemetries in these animals in the absence of any additional drugs as suggested by you above.

In addition, not much is known about pharmacokinetics of the compounds and the way of application. It would be hard to obtain all these basal experiments with measuring plasma concentrations, half-life, EC50 etc. before even starting the key experiments. In the end, nobody knows about potential toxicity to other organs, side effects and so on of the compounds. Thus we believe, the results may not be directly related to pure Nav1.8 inhibition at least *in vivo*. In addition, although we have proven the selectivity of the drugs before on the cellular level, this issue still remains controversial as suggested by the reviewer's questions. We believe that the best and most selective approach was to crossbreed the CaMKII transgenic with the Nav1.8 knock-out mice in order to now show improved survival which is associated with reduced arrhythmias in vivo. However, we fully agree with the reviewer that the next translational step would be to develop pharmacological application strategy to selectively target Nav1.8 in vivo.

Reviewer #3 (Remarks to the Author):

The authors present data supporting an intriguing conclusion that a majority of I-NaL is due to Nav1.8 and its interaction with CaMKII. This finding has important mechanistic implications and as well implications for development of future anti-arrhythmic therapy. Given the novelty and potential importance of the findings, shifting focus of cardiac I-NaL from Nav1.5 to Nav1.8, the rigor of the studies must be improved. **Contribution of non-myocyte cardiac cells to tissue Nav1.8 needs to be ruled out.** Additionally, **specificity of the drug studies** needs to be established as does efficiency and specificity of the knockout models to rule out secondary effects on Nav1.5. Most importantly, the authors need to **clarify the full survival** curve and arrhythmogenic phenotype of the animal models rather than draw conclusions from an apparently abbreviated survival curve and subsequent cellular studies. Specific requests are provided below.

Answer: We thank you for the appreciative assessment and the helpful scientific suggestions to improve the validity of the work. As suggested by you, we performed new work on specificity of the drugs, channel differentiation, in vivo arrhythmia quantification, and most importantly clarified the full survival curve.

1. Please quantify amount of Nav1.8 in non-failing and failing human ventricular muscle, relative to Nav1.5. Provide quantification in both intact ventricle and, to consider origin from non-muscle cardiac cells, quantify channel content in human or mouse isolated ventricular cardiomyocytes as well.

Answer: We thank the reviewer for this important comment. It would be of high interest to know how cell surface expression of Nav1.8 differs from expression of Nav1.8 in whole ventricular tissue to consider origin from non-muscle cardiac cells. To answer this question, we here provide *SCN10A* and *SCN5A* mRNA expression not only in human ventricular tissue (non-failing "NF" and heart failure "HF") but also in human ventricular cardiomyocytes (hCMs) isolated from failing hearts. As shown in the figure S1 below the Nav1.8 mRNA expression was significantly upregulated in myocardium from failing hearts compared to non-failing tissue. These results are in line with previous publications from our group (Dybkova et al., 2018). Of note, *SCN10A* mRNA expression is higher in isolated cardiomyocytes from failing hearts compared to ventricular failing myocardium.

Thus, it can be stated that a significant amount of *SCN10A* mRNA is originating from cardiomyocytes in the human heart. Although we can demonstrate the cardiomyocyte significance of Nav1.8 with this experiment, the presence of *SCN10A* in other cell types, in particular in cardiac neurons might also contribute at a certain level to whole Nav1.8 expression in the heart and to cardiac electrophysiology. Unfortunately, we are not able to provide data in cardiomyocytes from non-failing hearts, because we do not have access to this at our institution (only frozen samples).

We further compared the amount of *SCN10A* mRNA to *SCN5A* mRNA. As expected and shown before, we demonstrated that the expression of *SCN10A* in cardiac tissue as well as in isolated cardiomyocytes is lower compared to *SCN5A*. We found a 6.91-fold lower *SCN10A*-expression compared to *SCN5A* in HF myocardium, but only a 5.25-fold lower expression of *SCN10A* compared to *SCN5A* in isolated human cardiomyocytes (see. Figure S1).

We included the figure as supplemental Fig. S1 in the online supplement and in the results/discussion (page 4, lines 78-81).

Fig. S1: Comparison of SCN5A and SCN10A mRNA levels in tissue from non-failing and failing hearts, as well as isolates cardiomyocytes from failing hearts, RT-qPCR showing up-regulation of Nav1.8 mRNA in human HF ventricular myocardium and its presence as mRNA in isolated cardiomyocytes from human HF (NF: n=3, HF: n=3, CM n=3,).

We additionally investigated the levels of SCN10A mRNA in isolated mouse cardiomyocytes and whole mouse ventricular non-failing and failing tissue (WT and CaMKII δ c^{+T} mice). qPCR data of both tissue and cardiomyocytes from WT and CaMKII δ c^{+T} mice demonstrated that a significant amount of SCN10A mRNA in originates from cardiomyocytes.

Fig. S2: A significant amount of Nav_v1.8 mRNA in murine myocardium originates from cardiomyocytes, n=mice, tissue: WT = 5, CaMKII δ c^{+T}=5; cells: WT = 5, CaMKII δ c^{+T}=5.

In addition, we tried to improve the validity and significance of Nav1.8 in the single cardiomyocyte and hence to answer your criticisms we used an innovative approach and generated human ventricular cardiomyocytes derived from induced pluripotent stem cells (iPS-CM) from healthy individuals with Nav1.8 knock-out by CRISPR-CAS9 technology and measured I_{NaL} . Because of the very small amplitude of I_{NaL} under physiological conditions (different to heart failure), we increased I_{NaL} via β -adrenergic stimulation of iPS-CM using isoproterenol (50 μ mol/L). In our measurements, the application of isoproterenol to Nav1.8 knock-out iPS-CM resulted in significantly lower I_{NaL} compared to control iPS-CM (WT) stimulated with isoproterenol. These data clearly confirm that indeed Nav1.8 significantly contributes to I_{NaL} in cardiomyocytes. Furthermore, pre-incubation of WT iPS-CM with the specific Nav1.8 blocker PF-01247324 resulted in a significant reduction of I_{NaL} after isoproterenol stimulation. In sharp contrast, pre-incubation of knock-out iPS-CM with the Nav1.8 inhibitor did not affect the isoproterenol-induced enhancement of I_{NaL} , confirming the specificity of the drug. These new data now are included into the revised manuscript in Fig. 2E & 2F and in results (page 5, lines 99-107). Taken together our findings in iPS-cardiomyocytes underline that Nav1.8 plays a significant role for I_{NaL} -generation in the cardiomyocyte.

Fig. 2: **E:** Original traces of I_{NaL} from human ventricular SCN10A knock-out iPS-cardiomyocytes elicited using the protocol shown in inset. **F:** Mean data \pm SEM along with individual values shown in the graph plotting (control + Iso: $n=19$ cells/3 cardiac differentiations; control + Iso+ PF: $n=4$ cells/3 differentiations; KO + Iso: $n=11$ cells/3 differentiations; KO+Iso+PF-01247324= 12 cells/3 differentiations). Probability vs. control+ Iso (One-way ANOVA).

2. It is not clear that the pharmaceutical agents A-806734 and PF-01247324 do not block Nav1.5 as well, and thus the effects of blockade of Nav1.5 are attributed in the manuscript to blockade of Nav1.8. Please repeat the experiments Figure 3 and 4 in the setting of genetic or pharmacologic block of Nav1.5.

Answer: We thank the reviewer for this comment. We and others could show before, that both blockers A-803467 and PF-01247324 are Nav1.8-specific at the used concentrations (30 nmol/L for A-803467 and 1 μ mol/L for PF-01247324). Yang et al performed experiments ND7/23 cells showing a dose dependent selectivity for A-803467 on Nav1.8 over Nav1.5. While 30 nmol/L of A-803467 abolished all Nav1.8 dependent I_{NaL} -current, a dose of 1000 nmol/L did not show any relevant effects on Nav1.5 dependent I_{NaL} . After I_{NaL} augmentation with ATX-II an IC_{50} of 10nmol/L for Nav1.8 dependent I_{NaL} was calculated, while IC_{50} for Nav1.5 was 5000 nmol/L.

PF-01247324 was tested for its Nav1.8-selectivity by Payne et al (Payne et al, BJP, 2015). Here, a 50-fold Nav1.8-selectivity over Nav1.5 of the compound could be observed. At a concentration of 1 μ mol/L a complete block of Nav1.8-channels occurred while Nav1.5 dependent current was unaffected at this dose.

In line with these reports we could show in our previous work, that both blockers showed significant effects on I_{NaL} in Wild-Type murine cardiomyocytes after augmentation of I_{NaL} with either Isoproterenol or ATX-II (Dybkova et al. Cardiovasc Res, 2018; Bengel et al. J Mol Cell Cardiol, 2020). However, in SCN10A knock-out cardiomyocytes there were no relevant effects of both blockers on I_{NaL} . Experiments in SCN10A knock-out cardiomyocytes with 2 μ mol/L of TTX revealed a remaining I_{NaL} that has to be driven by other channel isoforms. These results support the thesis that the compounds A-803467 and PF-01247324 are selective for Nav1.8 in the used concentrations.

Although it would be an interesting approach to study the role of other sodium channels in the cardiomyocyte a complete genetic Knock-Out of Nav1.5 in a mouse model as suggested by the reviewer is unfortunately not possible as mice are not viable. Also, a complete pharmacological block of Nav1.5 is limited and we do expect a clear answer from such an experiment. TTX is known to inhibit Nav1.5 in high concentrations. However, also here a complete block of the channel will not occur. In addition to that using higher concentrations that would cause a complete block of Nav1.5 could also affect Nav1.8 channels.

Nevertheless, to ultimately correspond to your criticism we used the most appropriate approach and performed peak I_{Na} measurements in Wild-Type cardiomyocytes to test for potential effects of the blocker PF-01247324 on Nav1.5. We found no relevant effects of PF-01247324 in these cardiomyocytes compared to untreated controls promoting our hypothesis that Nav1.5 is not affected by the used concentration of PF-01247324. The results of these experiments are presented below and will be provided as Fig, S1 in our online supplement.

Fig. S3: A: Original traces and average current-voltage (*I-V*) relationships in mouse cardiomyocytes from wild type (WT) with or without PF-012473224 and SCN10A^{-/-} mice (WT: *n*=11 cells/5 mice; PF-01247324: *n*=11 cells/4 mice; SCN10A^{-/-}: *n*=10 cells/2 mice). Pharmacological inhibition or genetic knock-out of Nav1.8 did not affect Na⁺ peak. One-way ANOVA.

B: Normalized original traces and mean data±SEM for steady-state inactivation (measured with the protocol in inset). The mean data for half-inactivation (*V*_{1/2}) are shown in the right panel. *V*_{1/2} is derived from the curve by fitting it to standard Boltzmann equation: $h_{\infty} = 1 / \{1 + \exp[(V_{1/2} - V) / K_{\infty}]\}$ (WT: *n*=9 cells/5 mice; PF-01247324: *n*=6 cells/3 mice; SCN10A^{-/-}: *n*=8 cells/2 mice). Pharmacological inhibition or genetic knock-out of Nav1.8 did not affect steady-state inactivation. One-way ANOVA. **C:** Mean data±SEM for *I*_{Na} intermediate inactivation. Increasing conditioning pulse duration (*P*₁) reduced *I*_{Na} amplitude assessed with a second pulse (*P*₂) consistent with entry of Na⁺ channels into intermediate inactivation (protocol in inset). The mean data for plateau *y*₀ are shown in right panel (WT: *n*=12 cells/5 mice; PF-01247324: *n*=8 cells/5 mice; SCN10A^{-/-}: *n*=11 cells/3 mice). Pharmacological inhibition or genetic knock-out of Nav1.8 did not affect intermediate inactivation. One-way ANOVA. **D:** Mean data±SEM for recovery from inactivation. Increasing duration of the recovery interval between conditioning pulse (*P*₁, causing *I*_{Na} inactivation) and test pulse (*P*₂) results in an exponential increase in the amplitude of *I*_{Na} upon *P*₂ consistent with *I*_{Na} recovery (protocol in inset). The mean data for the rate constant of recovery *k*_{rec} are shown in right panel. Pharmacological inhibition or genetic knock-out of Nav1.8 did not affect recovery from inactivation. One-way ANOVA

In addition, pre-incubation of the above mentioned WT iPS-CM (Fig 2E & 2F) with the specific Nav1.8 blocker PF-01247324 resulted in a significant reduction of *I*_{NaL} after isoproterenol stimulation. In contrast, pre-incubation of knock-out iPS-CM with the Nav1.8 inhibitor did not affect the isoproterenol-induced enhancement of *I*_{NaL} pointing towards specificity of the drug with respect to Na channel isoforms.

In summary, we believe that we could demonstrate by discussing the evidence and performing new experiments that the compounds are specific for Nav1.8 compared to 1.5 and hope that the reviewer agrees.

3. Please provide RNA and western blot evidence of cardiomyocyte knockout of Nav1.8 and CamKII in each mouse model used. Please also confirm that Nav1.5 is not affected in adult mice of each mouse model.

Answer:

We thank the reviewer for this important question. To demonstrate that Nav1.5 is not affected in adult mice of each model, we performed Western-Blot experiments and qPCR analysis from tissue of Wild-Type and SCN10A-Knock-Out hearts with and without CaMKIIδc overexpression. As the new figure below indicates, there was no significant effect of SCN10A knock-out on Nav1.5 protein and mRNA expression (Fig. S6). We included this figure in the online supplement as Fig. S6. In addition, peak *I*_{Na} measurements comparing WT and SCN10-KO cardiomyocytes showed no significant differences in peak *I*_{Na} amplitude or channel kinetics so that it can be assumed that Nav1.5 expression and function

are not affected by SCN10A knock-out. We hope the reviewer accepts these experiments to resolve any doubts. A similar expression of Nav1.5 in hearts from WT and SCN10A knock-out mice was already shown by Stroud et al on the mRNA level (Stroud et al. J Am Heart Assoc. 2016) who used the same SCN10A knock-out model (originally introduced by Akopian et al, Nat Neurosci. 1999).

Fig. S6: **A:** Original western blots of Nav1.5 in myocardial tissue of WT, SCN10A^{-/-}, CaMKIIδc^{+T} and SCN10A^{-/-}/CaMKIIδc^{+T} mice. **B:** Western blot quantification of Nav1.5 normalized to global protein loading. Nav1.8 knock-out did not influence Nav1.5 expression neither in SCN10A^{-/-} vs. WT nor in SCN10A^{-/-}/CaMKIIδc^{+T} vs. CaMKIIδc^{+T} in mouse myocardium tissues (WT: n=6, SCN10A^{-/-}:n=5, CaMKIIδc^{+T}: n=6; SCN10A^{-/-}/CaMKIIδc^{+T} =7. One-way ANOVA **C:** mRNA expression of Nav1.5 in myocardial tissue and isolated murine cardiomyocytes. Nav1.8 knock-out did not influence Nav1.5 mRNA expression neither in mouse myocardial tissue (WT n=5, SCN10A^{-/-} n=4, CaMKIIδc^{+T}=4; SCN10A^{-/-}/CaMKIIδc^{+T} =5) nor in cardiomyocytes (WT n=5, SCN10A^{-/-} n=5, CaMKIIδc^{+T}=5; SCN10A^{-/-}/CaMKIIδc^{+T} =5). One-way ANOVA

We further performed experiments referring to your question regarding CaMKII.

CaMKII protein and mRNA levels are significantly increased in ventricular cardiomyocytes and tissue from CaMKII transgenic with and without Nav1.8 knock-out compared to those in Wild-Type. Most importantly, CaMKII levels did not significantly differ between CaMKII transgenic and CaMKII transgenic mice with Nav1.8 knock-out. Accordingly, we now provide Western Blot and mRNA evidence of the CaMKII overexpression in our model. The data is presented below and in the revised version in the online supplement Fig S7.

Fig. S7: **A:** Original western blots of CaMKIIδc in myocardial tissue of WT, SCN10A^{-/-}, CaMKIIδc^{+T} and SCN10A^{-/-}/CaMKIIδc^{+T} mice. **B:** Western blot quantification of CaMKIIδc normalized to global protein loading. Significant upregulation of CaMKIIδc could be confirmed in the myocardium from CaMKIIδc^{+T} and SCN10A^{-/-}/CaMKIIδc^{+T} mice (WT: n=6, SCN10A^{-/-}:n=6, CaMKIIδc^{+T}: n=6; SCN10A^{-/-}/CaMKIIδc^{+T}: n=6) **C:** mRNA expression of CaMKIIδc in myocardial tissue and isolated murine cardiomyocytes. Significant upregulation of CaMKIIδc could be confirmed in the myocardium (WT: n=5, SCN10A^{-/-}: n=4, CaMKIIδc^{+T}: n=5, SCN10A^{-/-}/CaMKIIδc^{+T}: n=5) and in the cardiomyocytes (WT n=5, SCN10A^{-/-} n=5, CaMKIIδc^{+T}: n=5, SCN10A^{-/-}/CaMKIIδc^{+T} =5) from CaMKIIδc^{+T} and SCN10A^{-/-}/CaMKIIδc^{+T}. One-way ANOVA

Unfortunately, the question regarding cardiomyocyte knock-out of Nav1.8 is not that easy to answer due to the Nav1.8 knock-out construct. We have invested a lot of time and are now reporting in detail on the process and our findings. To address the reviewers concern regarding the cardiomyocyte knock-out of Nav1.8 we performed qPCR in our mice and detected SCN10A mRNA in each genotype. Thus, our knock-out mice, which carries a deletion of SCN10A exons 4 and 5, still expresses mRNA containing exons downstream of exon 5. This is also in agreement with a previous study that showed the presence of long SCN10A mRNA transcripts in knock-out mice (Akopian et al, Nat neuroscience 1999).

As expected, also the western-blot data showed expression of Nav1.8 in each model. In fact, the Nav1.8 protein (UniProt ID SCN10A) consists of 1958 amino acids and in the knock-out it is slightly shortened to 1888 (the exon 4 and 5 encode 27 and 43 amino acids respectively. 27+43=70 amino acids are

absent, when the exon 3 and 6 are spliced together in knock-out mice, while exons 4 and 5 are skipped due to the presence of a Neomycin resistance cassette, which destroys the S4 voltage sensor of domain I). Therefore, in wild-type, Nav1.8 is functional while a non-functional channel is expressed in the knock-out model with nearly same amino acids length as already shown by the initial constructors of the mouse model (Akopian et al. Nat neuroscience 1999).

As we could not confirm SCN10A knock-out by RT-PCR or Western Blot approach we went for a semiquantitative mRNA transcriptional approach. Stroud et al. (J Am Hear Assoc) performed their transcriptional analysis with primers spanning exon 3 to 8 using the same mouse model in their paper. They used dorsal root ganglion tissue for their PCR getting a 450 base-pair fragment in SCN10A KO mice in contrast to 670 base-pairs in WT. We tried to go for a similar approach in ventricular tissue from WT and SCN10A knock-out mice to confirm cardiomyocyte knock-out of SCN10A. However, SCN10A mRNA expression in myocardial tissue appears to be too low for a detection by our semiquantitative PCR approach.

To confirm that the right mouse lines were used for the experiments, we included an original of our semiquantitative genomic PCR for genotyping which is performed from ear biopsies in our department.

Fig. A5: Original results from semi-quantitative genomic PCR in WT, SCN10^{+/-} and SCN10A^{-/-} mice that were used for genotyping.

Our anti-Nav1.8 western-blot exhibits bands in the samples from wild-type and Nav1.8 knock-out mice with or without CaMKII overexpression. This is because the antibody recognizes a peptide at the C-terminal part of Nav1.8, which is expressed in both wild-type and knock-out mice. The absence of 70 amino acids cannot be detected by size for such a big protein (a ca. 8 kDa difference for a protein migrating as 250 kDa is very difficult to see on the gels). Interestingly, our Western blot shows two signals, both correspond to large proteins (~250kDa, which roughly fit to the full-length protein, which must be 1958 amino-acids long). We speculate that the upper band is (hyper)phosphorylated Nav1.8, while the lower one constitutes the not phosphorylated protein. This fits nicely to the pattern of CaMKII: CaMKII transgenic overexpress CaMKII, and this results in increased amount of upper band of Nav1.8 (phospho).

Although neither RT-PCR nor Western-Blot experiment can confirm the knock-out of Nav1.8 due to the reasons listed above, our Nav1.8 knock-out model is well established and was published by several groups before.

Fig. A6: Original Western-Blots of Nav1.8 in tissue from WT, SCN10A^{-/-}, CaMKIIδc^{+T} and SCN10A^{-/-}/CaMKIIδc^{+T} hearts

4. The survival advantage of knocking out Nav1.8 which starts at 60 days appears to dissipate at 100 days (Figure 6A). The low ejection fraction of the mutant mice suggests all the mice will die at a relatively young age, affecting the conclusions regarding survival. Please complete the survival curve until either all animals die or the curves plateau to more accurately portray steady state survival.

Answer: We thank the reviewer for this helpful remark. For the revised manuscript we reanalyzed existing data on survival of the mice and now present survival data until day 200. We further observed additional mice (12 CaMKIIδc^{+T} and 13 SCN10A^{-/-}/CaMKIIδc^{+T}) and added their survival data to our analysis. In the new survival curve, it comes clear, that the survival benefit continues also after day 100 until nearly all animals died. Due to the added number of mice the survival benefit got even more significant. The reviewer is right, that mice die at a very young age. However, this is explainable by the fact, that an end-stage heart failure is present at 12 weeks of age due to CaMKII overexpression.

Fig. 4: A: Survival curve of CaMKIIδc^{+T} and SCN10A^{-/-}/CaMKIIδc^{+T} (43 vs 63 animals). Log-rank (Mantel-cox test and Gehan-Breslow-Wilcoxon test) were performed to calculate the survival percentage of mice. Probability vs CaMKIIδc^{+T}.

5. The low ejection fraction of the mutant mice suggests the mice die from progressive heart failure rather than ventricular arrhythmogenesis. Please provide telemetry evidence of in vivo arrhythmogenesis and sudden cardiac death of the mice studied. Given the concern of

progressive heart failure, please also provide body weight and activity levels versus age of the mice.

Answer: We thank the reviewer for her/his comment. Showing the reason of death (tachyarrhythmic vs. bradyarrhythmic vs. pump failure) is a difficult and nearly impossible task. To analyse this the telemetric device of the mice needs to be turned on and the mice house on the measurement plates all the time. Here a limited amount of available telemetric devices as well as a limitation of battery make such experiments not only expensive but also reduce the possibility of detection. Most importantly, it cannot be differentiated whether an arrhythmia is occurring primarily or whether it occurs secondary to pump failure or after PEA/other arrhythmia (pulseless electric activity) with primary hypoxia.

To correspond to the reviewer's suggestion, we designed an amendment to our existing permission that allowed us to implant the requested telemetry monitors in a limited number of CaMKII δ c transgenic animals.

We tried to analyse arrhythmias and potentially the cause of death in a subset of mice for 2-4 weeks at an age of 8 weeks. As expected, a significant amount of episodes of PVCs, non-sustained VTs and sustained VTs were detected in our CaMKII δ c transgenic mice. Death occurred neither in CaMKII TG nor in CaMKII δ c TG + Nav1.8 $^{-/-}$ mice during this time period. Most importantly, a relevant reduction in PVCs and in particular VT could be observed in double transgenic mice investigated in a blinded manner.

We therefore included the following sentence in the result section of the revised manuscript: "*The Nav1.8 knock-out was capable to significantly reverse a relevant part of the arrhythmogenic CaMKII δ c transgenic substrate which is associated with sudden cardiac death. This is strongly supported by a reduced incidence of ventricular arrhythmias in SCN10A $^{-/-}$ /CaMKII δ c $^{+/T}$ mice in vivo and the experimental findings that Nav1.8 inhibition and deletion reduce the occurrence of EADs, DADs, and diastolic SR-Ca $^{2+}$ leak as accepted arrhythmogenic substrates for malignant arrhythmias and sudden cardiac death. However, due to the sporadic nature of sudden cardiac death, we were not able to correlate mortality with ventricular arrhythmias in these CaMKII δ c transgenic mice as this is technically and ethically not feasible*" (Page 10, Line 261-265).

In addition to the arrhythmias the telemetry data provided us the requested insights into the activity of the mice. Here we found no significant differences in activity between CaMKII transgenic mice with or without additional Nav1.8 knock-out.

The results of the recordings are presented below and in the revised manuscript as Fig.6.

Of note body-weight did not significantly differ between WT, SCN10A $^{-/-}$, CaMKII δ c $^{+/T}$ and SCN10A $^{-/-}$ /CaMKII δ c $^{+/T}$. The fact, that mean body-weight in diseased animals was not increased is explainable by the fact, that some animals suffered from severe illness at the timepoint when body-weight was measured and hearts were explanted.

Fig. 6: **A:** Original ECG-traces of CaMKII δ c^{+T} and SCN10A^{-/-}/CaMKII δ c^{+T} showing ventricular arrhythmias. **B:** Unchanged activity levels in SCN10A^{-/-}/CaMKII δ c^{+T} compared to CaMKII δ c^{+T} (CaMKII δ c^{+T} (4 mice); SCN10A^{-/-}/CaMKII δ c^{+T} (3 mice)). **C:** Reduction of premature ventricular contractions in SCN10A^{-/-}/CaMKII δ c^{+T} ($p=0.08$, student's t -test). **D:** Reduction of ventricular tachycardia in SCN10A^{-/-}/CaMKII δ c^{+T} ($p<0.05$, student's t -test).

6. For ratiometric measurement of diastolic calcium, please provide the methodological detail regarding obtaining minimal and maximal fluorescent ratios and convert all measured calcium concentration **ratios to units of nM**.

Answer: We thank reviewer for her/his comment. Ca²⁺-transient measurements were performed using an epifluorescence microscope with an ION optix system. As Fura-2-AM is a ratiometric dye which requires illumination at two wavelengths (340 and 380nm) calibration into nM is not necessary. Kong and Lee showed that there is a linear correlation between the ratio of the wavelengths 340 and 380nm and the free intracellular calcium concentration. Therefore, the ratio of fluorescence obtained at these wavelengths provides an estimate of intracellular [Ca²⁺].

Intracellular calcium concentrations indicated in nM would indeed require calibration at the beginning of each experiment. As the exact concentration in nM does not provide indispensable additional information for our purpose, we gave up this approach.

7. Please provide **clinical details** of the Stage IV explanted human hearts- at a minimum age and sex of patients, cause of cardiomyopathy, ejection fraction, and LV dimensions. Also, please provide cross-clamp time of the control human hearts and as well as age, sex, and cause of death.

Answer: We thank the reviewer for pointing out this descriptive shortcoming. In the revised online supplement, we now provide a table with clinical details of the end-stage explanted human hearts. However, we cannot provide cause of death and cross clamp time of the control human hearts, as we have no access on this data from an ethical point of view.

Table S1. Overview of age, ejection fraction and medical therapy of human failing hearts used for experiments (14 patients): ICM: ischemic cardiomyopathy; EF: ejection fraction; LVEDD: Left ventricular end-diastolic diameter; ACE: ACE-inhibitor; β -B: β -blocker; DIU: diuretic; AMIO: Amiodarone (anti-arrhythmic); AT1: AT-1 receptor antagonists; MRA: mineralocorticoid receptor antagonist; CAT: catecholamines

Patient data	mean\pmSEM
Male sex (%)	85.7
age (years)	48 \pm 5
ICM (%)	50.0
Diabetes (%)	21.4
EF (%)	23 \pm 1
LVEDD (mm)	61.7 \pm 3.2
ACE (%)	57.1
β-B (%)	85.7
DIU (%)	78.5
AMIO (%)	57.1
AT1 (%)	21.4
MRA (%)	50.0
CAT (%)	21.4

Although we cannot offer to publish your paper in Nature Communications, the work may be appropriate for another journal in the Nature Research portfolio. If you wish to explore suitable journals and transfer your manuscript to a journal of your choice, please use our manuscript transfer portal. If you transfer to Nature-branded journals or to the Communications journals, you will not have to re-supply manuscript metadata and files. This link can only be used once and remains active until used.

All Nature Research journals are editorially independent, and the decision to consider your manuscript will be taken by their own editorial staff. For more information, please see our manuscript transfer FAQ page.

This email has been sent through the Springer Nature Tracking System NY-610A-NPG&MTS

Confidentiality Statement:

This e-mail is confidential and subject to copyright. Any unauthorised use or disclosure of its contents is prohibited. If you have received this email in error please notify our Manuscript Tracking System Helpdesk team at <http://platformsupport.nature.com>.

Details of the confidentiality and pre-publicity policy may be found here <http://www.nature.com/authors/policies/confidentiality.html>

Privacy Policy | Update Profile

REVIEWER COMMENTS

Reviewer #1 (Remarks to the Author):

In their revised manuscript, the authors have provided additional information and experimental data. However, a number of concerns and limitations remain.

The patch clamp experiments remain in my view suboptimal, in particular the fact that late sodium current was not measured at physiological temperature. Also, late sodium current was measured at a single potential. In their rebuttal, the authors state that they "are not interested in current kinetics" and that measured late sodium current at a potential at which the maximum values was observed. However, the key point here is that one should measure the current at different potentials for the exact reason to establish at which potential the current is maximal. The current may be maximal at different potentials in various groups, and comparing the current at various potentials is therefore essential.

I do appreciate the additional work and effort by the authors. In particular, the in vivo assessment of arrhythmias with telemetry is interesting, although I think that the example shown in Figure 6A is actually not a ventricular arrhythmia but rather a junctional tachycardia (or supraventricular, since P waves can still be seen?). However, overall the paper still provides limited novel mechanistic insight. The fact remains that the CamKII-TG mice are a model of heart failure, and it is expected that these have a pro-arrhythmic increased late sodium current driven by Nav1.8, since this has been previously demonstrated by the authors and others in other forms of heart failure. So in fact, all observations may simply be explained by the heart failure phenotype. The authors claim that their findings demonstrate evidence for a "CamKII-dependence of Nav1.8-driven late sodium current", but in fact this is only based on the observation in human HF cardiomyocytes that inhibition of both CamKII and Nav1.8 together resulted in a similar reduction in late sodium current and calcium sparks as compared to Nav1.8 inhibition alone. However, this is indirect evidence, and in my view the authors have not provided sufficient (additional) evidence for this conclusion. Co-immunoprecipitation may also reflect indirect interaction, not necessarily direct. Indeed, I would consider the findings in line with a functional "interaction" between CamKII and Nav1.8-based late sodium current, which is not novel.

As such, the insight remains incremental, and merely adds on to the previous work on this topic. The only concrete conclusion that can be drawn from the findings is that Nav1.8-based late sodium current is enhanced and pro-arrhythmic in CamKII-TG mice which is not surprising given their heart failure phenotype. While interesting and potentially relevant, the paper does not demonstrate a "new proarrhythmic CamKII downstream target".

Reviewer #2 (Remarks to the Author):

The revised manuscript is substantially improved, and I applaud the effort of the authors in this work. My only serious reservation relates to the lack of an appropriate control for the myristoylated AIP peptide. As you point out, many publications show data lacking an appropriate control. Unfortunately, these cell membrane permeant peptides can have profound, off target actions, on myocardial electrophysiology. Please see Wu PNAS 2009 supplementary Fig 3 for an example (PMID: 19276108). Thus, the current data using AIP without a control stop short of proving a role for CaMKII in various studies throughout the manuscript.

Reviewer #3 (Remarks to the Author):

The extensive additional data and explanations are highly supportive of the original conclusions. These clarifying studies are appreciated. No further concerns.

REVIEWER COMMENTS

Reviewer #1 (Remarks to the Author):

In their revised manuscript, the authors have provided additional information and experimental data. However, a number of concerns and limitations remain.

Answer: We thank the reviewer for re-evaluating our work.

The patch clamp experiments remain in my view suboptimal, in particular the fact that late sodium current was not measured at physiological temperature. Also, late sodium current was measured at a single potential. In their rebuttal, the authors state that they "are not interested in current kinetics" and that measured late sodium current at a potential at which the maximum values was observed. However, the key point here is that one should measure the current at different potentials for the exact reason to establish at which potential the current is maximal. The current may be maximal at different potentials in various groups, and comparing the current at various potentials is therefore essential.

Answer: We took your concerns serious and have extensively answered all questions including temperature and I_{NaL} measurements in our previous revision. By doing so we could demonstrate that there is no relevant difference between room and body temperature. Therefore, we still believe that our experimental setting is appropriate.

In contrast, we agree with the reviewer that it would be of interest to provide evidence that measurement at -35 mV was well suited to investigate $Na_V1.8$ -dependent I_{NaL} . Previous studies (Yang et al, *Circ Res*, 2012, Stroud et al, *J Am Heart Assoc*, 2016) demonstrated a rightward shift in the voltage dependence of $Na_V1.8$ activation relative to $Na_V1.5$. However, the latter studies were performed either in heterologous expression systems (Yang et al, *Circ Res*, 2012) or in mouse ventricular myocytes but with artificial stimulation of I_{Na} using ATX-II (Stroud et al, *J Am Heart Assoc*, 2016), and there were substantial differences in gating behavior between these models: the positive shift in steady-state activation (compared to $Na_V1.5$) appeared to be less pronounced (about 20 mV less) in mice compared with HEK cells. However, if a similar shift was present in our model, the measurement of I_{NaL} at -35 mV would have tended to underestimate the extent of $Na_V1.8$ -dependent I_{NaL} only.

Nevertheless, to address this issue, we performed new experiments for the revised version of the manuscript that analyzed the current-voltage relationship of I_{NaL} mediated by $Na_V1.8$ in isolated ventricular myocytes from mice overexpressing CaMKII δ c. Supplementary Figure 3 shows that the overall I_{NaL} exhibits a maximal negative current at -20 mV (Fig. S3). Similar to Stroud et al, we analyzed the PF-01247324-sensitive current by subtracting the total I_{NaL} with the remaining current after exposure to $Na_V1.8$ blocker PF-01247324. Interestingly, this $Na_V1.8$ -dependent I_{NaL} exhibits a maximum negative current between -20 mV and -30 mV. Thus, I_{NaL} measurements at -35 mV, as performed in the present manuscript, are well suited to investigate the significance of $Na_V1.8$ -dependent I_{NaL} . There are several possible explanations for the lack of the rightward shift in $Na_V1.8$ activation in our model. For example, Stroud and colleagues used ATX-II to artificially stimulate I_{NaL} , and the response to ATX-II dependent stimulation could be different between $Na_V1.5$ and $Na_V1.8$.

Fig. S3: **a** Original traces of the current response between 100 and 300 ms of the respective voltage step in vehicle- and PF-01247324-treated ventricular cardiomyocytes elicited by application of the voltage step protocol shown in the inset. The box depicts the time frame between 180 and 190 ms used for analysis of I_{NaL} . **b** I-V curves (Mean data \pm SEM) of I_{NaL} in ventricular cardiomyocytes from CaMKII $\delta^{+/T}$ mice treated with vehicle or PF-01247324. I_{NaL} densities are significantly reduced in PF-01247324-treated (n=21 cells/9 mice) compared to vehicle-treated (n=13 cells/6 mice) ventricular cardiomyocytes at -30 ($p < 0.01$) and -20 mV ($p < 0.05$, mixed effects analysis with Holm-Sidak's post-hoc test). PF-01247324 sensitive I_{NaL} was calculated as the difference between the mean I_{NaL} densities of vehicle and PF-treated cardiomyocytes at each membrane potential.

We hope the reviewer accepts this experiment to resolve his/her doubts.

I do appreciate the additional work and effort by the authors. In particular, the *in vivo* assessment of arrhythmias with telemetry is interesting, although I think that the example shown in Figure 6A is actually not a ventricular arrhythmia but rather a junctional tachycardia (or supraventricular, since P waves can still be seen?).

Answer: We thank the reviewer appreciating our novel *in-vivo* experiments. Nevertheless, as scientists but also as clinical electrophysiologists we do not agree with the reviewer that our example does not show a ventricular tachycardia. The reviewer is right, that p-waves still can be seen in the ECG-trace. However, the p-waves are not in regular intervals with the QRS complexes, but the PR intervals get shorter until the p-wave disappears within the QRS. Therefore, the criteria of AV dissociation for ventricular tachycardia is positive.

However, overall the paper still provides limited novel mechanistic insight. The fact remains that the CamKII-TG mice are a model of heart failure, and it is expected that these have a pro-arrhythmic increased late sodium current driven by Nav1.8, since this has been previously demonstrated by the authors and others in other forms of heart failure. So in fact, all observations may simply be explained by the heart failure phenotype. The authors claim that their findings demonstrate evidence for a "CamKII-dependence of Nav1.8-driven late sodium current", but in fact this is only based on the observation in human HF cardiomyocytes that inhibition of both CamKII and Nav1.8 together resulted in a similar reduction in late sodium current and calcium sparks as compared to Nav1.8 inhibition alone. However, this is indirect evidence, and in my view the authors have not provided sufficient (additional) evidence for this conclusion. Co-immunoprecipitation may also reflect indirect interaction, not necessarily direct. Indeed, I would consider the findings in line with a functional "interaction" between CamKII and Nav1.8-based late sodium current, which is not novel.

As such, the insight remains incremental, and merely adds on to the previous work on this topic. The only concrete conclusion that can be drawn from the findings is that Nav1.8-based late sodium current is enhanced and pro-arrhythmic in CamKII-TG mice which is not surprising given their heart failure phenotype. While interesting and potentially relevant, the paper does not demonstrate a "new proarrhythmic CamKII downstream target".

Answer: We are a bit surprised by the persistent statement, that an interaction between CaMKII and Nav1.8-based I_{NaL} is not novel. By best of our knowledge an interaction of Nav1.8 with CaMKII was not shown before. Most importantly, inhibition of a specific Na⁺-channel isoform and the resulting positive effect on survival is completely novel and from a translational point of view unique. In the context of the general difficulty of Na⁺-channel inhibition in structural heart disease, our findings are of particular interest.

However, we already discussed this issue with the reviewer in detail in the first revision and have the feeling that different opinions will remain at this point. Nevertheless, we thank the reviewer for the detailed critical technical input on our experiments, which has clearly improved our data and thereby the manuscript. We hope the reviewer also agrees on this.

Reviewer #2 (Remarks to the Author):

The revised manuscript is substantially improved, and I applaud the effort of the authors in this work.

Answer: We thank the reviewer for appreciating our work.

My only serious reservation relates to the lack of an appropriate control for the myristoylated AIP peptide. As you point out, many publications show data lacking an appropriate control. Unfortunately, these cell membrane permeant peptides can have profound, off target actions, on myocardial electrophysiology. Please see Wu PNAS 2009 supplementary Fig 3 for an example (PMID: 19276108). Thus, the current data using AIP without a control stop short of proving a role for CaMKII in various studies throughout the manuscript.

Answer: We thank the reviewer for her/his important comment and the precise description of the publication that supports your concern. As AIP is commonly used in many projects, we were indeed surprised by this publication and have checked the literature accordingly. It has to be noted, that the effects of myristoylation on membrane potentials in the cited paper were seen in sinoatrial cells and not in ventricular cardiomyocytes. Nonetheless, we took your concern seriously.

To show that a potential myristoylation of the cell membrane may influence I_{NaL} or intracellular Ca^{2+} -handling two experiments would be possible. On the one hand, measurements of I_{NaL} and diastolic Ca^{2+} -leak in presence of myristoylation AIP and a control peptide. However, for AIP such a peptide does not exist. The paper of Wu et al cited by you used myristoylated-AC3-C which constitutes the control peptide of the CaMKII-inhibitor AC3-I. Although, AC3-C does not represent a control peptide for AIP the pure effects of membrane myristoylation on I_{NaL} or diastolic Ca^{2+} -leak should be able to detect by using this peptide. Unfortunately, AC3-C was currently hard to obtain (e.g. via Nucleics as used in the cited paper). We tried to solve this issue by a different approach.

To address your concern we measured Ca transients, I_{NaL} and diastolic Ca^{2+} -leak in cardiomyocytes from CaMKII-Knock-Out mice in the absence or presence of myristoylated AIP. To get measurable I_{NaL} and SR- Ca^{2+} -leak we treated the cardiomyocytes with 50 μ mol/L of isoproterenol. We hypothesized that if no effects are observed on I_{NaL} or diastolic Ca^{2+} -leak it can be concluded that AIP has no off target effects at least on these key variables of our manuscript.

Indeed we did not observe any effects of myristoylated AIP on neither Ca^{2+} -transient amplitude, I_{NaL} , or diastolic Ca^{2+} -leak in CaMKII knock-Out cardiomyocytes. Therefore, we believe that myristoylated AIP does not affect our measurements by off target effects.

The results of this additional experiments are presented below. We hope the reviewer accepts this approach to resolve his/her doubts at least with regards to our preparations and readouts and therefore our manuscript.

Fig A1: **a, b** I_{NaL} measured in isolated cardiomyocytes from CaMKII knock-out mice was not significantly different in cardiomyocytes treated with the CaMKII inhibitor AIP or untreated cardiomyocytes (n=5 mice, cells = 10 untreated, 7 AIP). **c, d** Ca^{2+} -spark frequency measured by confocal microscopy using the dye Fluo 4-AM did not differ between untreated cardiomyocytes and cardiomyocytes treated with AIP with or without additional treatment with isoproterenol 50 μ mol/L (n=4 mice, cells = 74 untreated, 88 AIP, 28 Iso, 33 Iso + AIP). **e, f** Ca^{2+} -transient amplitude (F/F₀) did not differ between the treatment groups (n=4 mice, cells = 24 Iso, 31 Iso + AIP) compared within paired t-test.

In summary, we hope that the reviewer agrees on publication of our manuscript in the present form. We thank you for your professionally highly qualified input on our work!

Reviewer #3 (Remarks to the Author):

The extensive additional data and explanations are highly supportive of the original conclusions. These clarifying studies are appreciated. No further concerns.

Answer: We thank the reviewer for appreciating our work guided by her/his excellent comments on the previous version of our manuscript.

REVIEWERS' COMMENTS

Reviewer #1 (Remarks to the Author):

Thank you for providing the additional data on the late sodium current, which is very helpful. I have no further comments.

Reviewer #2 (Remarks to the Author):

I am satisfied by the control studies for potential off target actions of myristoylated AIP by using CaMKII δ ^{-/-} ventricular myocytes